# Effervescence in a binary mixture with nonlinear non-reciprocal interactions

Suropriya Saha [1] ✉ & Ramin Golestanian [1,2] ✉

Non-reciprocal interactions between scalar fields that represent the concentrations of two active species are known to break the parity and time-reversal (PT) symmetries of the equilibrium state, as manifested in the emergence of travelling waves. We explore the notion of nonlinear non-reciprocity and consider a model in which the non-reciprocal interactions can depend on the local values of the scalar fields in such a way that the non-reciprocity can change sign. For generic cases where such couplings exist, we observe the emergence of spatiotemporal chaos in the steady-state. We associate this chaotic behaviour with a local restoration of PT symmetry in fluctuating spatial domains, which leads to the coexistence of oscillating densities and phase-separated droplets that are spontaneously created and annihilated. We uncover that this phenomenon, which we denote as *effervescence*, can exist as a dynamical steady-state in large parts of the parameter space in two different incarnations, as characterised by the presence or absence of an accompanying travelling wave.

Interactions between components of biological and artificial living matter are mediated in a wide variety of ways across the scales[1]: from complex behaviour patterns in humans[2], to visual perception in birds[3], hydrodynamic interactions in ensembles of cilia and flagella[4,5], information-controlled feedback in programmable active colloids[6,7], and chemical fields in catalytically active colloids[8–10] and enzymes[11,12]. These microscopic interactions quite generically break action-reaction symmetry due to non-equilibrium conditions. Reciprocity breaking has already had a far reaching impact in fields like structural mechanics, in realising meta-materials[13], and in optics, by achieving photon blockade[14]. In recent years, non-reciprocity in interactions has generated interest as an exciting new ingredient to develop minimal models for active matter systems out of thermodynamic equilibrium[12,15–20].

The scope of novel collective behaviour increases dramatically in mixed populations of active particles, which allow competition between non-reciprocity and alignment interactions in polar matter[18], or phase separation in scalar densities[15,16]. In a non-conserved polar system, the spontaneously broken symmetry is restored with the emergence of chirality, where the polarity of one species trails behind the other[18,20]. Conserved active scalar field theories for two species

with non-reciprocal interactions display travelling waves and moving patterns in the steady-state[15,16,21]. When activity, i.e., the strength of non-reciprocity, is strong enough to win over the thermodynamic forces driving the system toward bulk phase separation, the system reaches novel steady-states that break the parity and time-reversal (PT) symmetry of the bulk-separated equilibrium state. Similar phenomena are frequently encountered in open quantum systems that exchange energy with a bath and are consequently described by a non-Hermitian Hamiltonian[22,23]. So far, photonic systems have been primarily used for experimental realisations of non-Hermitian quantum mechanics[24,25]. However, classical systems have the potential to be used for similar purposes, for example, using coupled enzyme cycles in reaction networks[26].

Here, we have generalised the non-reciprocal Cahn-Hilliard (NRCH) model developed in ref. 15 for two non-reciprocally interacting conserved species with the aim to explore the pattern forming ability of scalar field theories with non-reciprocal interactions beyond travelling waves. The inspiration to look beyond coherent oscillations is abundant in the rich literature on pattern formation[27,28] and chemical turbulence[29,30]. The examples listed above, however, concern systems without explicit number conservation, whereas recent developments

[1]Max Planck Institute for Dynamics and Self-Organization (MPIDS), Göttingen, Germany. [2]Rudolf Peierls Centre for Theoretical Physics, University of Oxford, Oxford, UK. ✉e-mail: suropriya.saha@ds.mpg.de; ramin.golestanian@ds.mpg.de

in active phase separation have highlighted the significance of number conservation in pattern formation[31–34].

In this work, we have explored the notion of nonlinear non-reciprocal interaction in a system that is subject to constraints of particle-number conservation. This generalisation is important because in real physical systems, for which NRCH serves as a minimal description—such as Janus colloids[35], quorum-sensing self-propelled particles[36], or mass-conserving reaction-diffusion systems[37]—effective interactions stem from several competing effects, and, in general, depend on the number densities. In NRCH[15], non-reciprocal interaction of constant sign and strength denoted as $\alpha$ leads to a fixed phase difference between the patterns of densities of the two species, i.e., we can describe the system as being in a state where one of the species 'chases' the other. The striking consequence of nonlinear non-reciprocity is that the sign and the magnitude of non-reciprocity are no longer fixed, and can be determined by the dynamics. We include non-specific nonlinear contributions in the coefficient of non-reciprocal interaction, which amounts to having an effective density-dependent $\alpha$, and allow the binary mixture to evolve to a non-equilibrium steady-state. We find that quite generically—i.e., independent of the details of the interaction—the system fractionates into dynamic spatial domains in the steady-state. In domains with large positive effective $\alpha$, species 1 chases 2 while the roles are reversed in regions where the sign is negative. Droplet formation occurs in regions where the magnitude of the effective $\alpha$ is sufficiently low such that currents driving phase separation dominate. The oscillating densities and the nucleation of droplets occur simultaneously leading to spatiotemporal chaos evident in patterns shown in Fig. 1. The primary objective of this work is to classify the patterns that emerge across a broad range of non-reciprocity, which we term effervescence—a term that literally evokes qualities of vivacity, bubbliness, or fizz.

We will first introduce the nonlinear NRCH model, which has two main components: a bulk free energy (that describes a passive phase separating binary mixture) and a non-reciprocal interaction with nonlinear terms (that makes the system active). We will constrain the model by imposing an additional symmetry thereby reducing it to a

simple model with just two non-reciprocal couplings as free parameters. The reduction in complexity enables exact travelling wave solutions whose stability can be tested analytically. We will highlight the defining aspects of effervescence by extensive numerical analysis of the dynamics, focusing on the droplets. We then perform stability analyses—both around the travelling wave solution and the local solutions corresponding to the regions of vanishing non-reciprocity—to create a stability diagram that will be compared with a numerically obtained state diagram. We will discuss the role of tuning composition in determining the steady-state and examine the robustness of the results with regard to the form of the nonlinear terms. Finally, we discuss the experimental realisation of our work in the context of physical systems where nonlinear non-reciprocity is expected to appear naturally.

## Results

### Nonlinear NRCH model

To build our theoretical framework, we start with the dynamics of conserved fields $\phi_i$ ($i = 1, 2$) that can be written as $\dot{\phi}_i = \Gamma_i \nabla^2 \mu_i$, in terms of the scalar chemical potentials $\mu_i$ and mobilities $\Gamma_i$. At equilibrium, $\mu_i$ can be obtained from a free energy $F$ via $\mu_i \equiv \delta F / \delta \phi_i$. The free energy is chosen as $F = \int_{\mathbf{r}} f(\phi_i(\mathbf{r}, t)) + \frac{K_1}{2}(\boldsymbol{\nabla}\phi_1)^2 + \frac{K_2}{2}(\boldsymbol{\nabla}\phi_2)^2$, where $f$ is the Helmholtz free energy (per unit volume) that describes phase separation in binary systems and $K_i$ is the interfacial tension for density $i$. We now introduce non-equilibrium activity in the model by adding non-reciprocal interactions between the two species. This can be achieved by introducing an anti-symmetric coupling between the species. Reciprocal interactions are included in the expression for $f$. We can write the governing equations for the two fields as

$$\partial_t \phi_1 = \Gamma_1 \boldsymbol{\nabla}^2 \left[ \frac{\partial f}{\partial \phi_1} - \alpha(\phi_1, \phi_2)\phi_2 \right] - \Gamma_1 K_1 \boldsymbol{\nabla}^4 \phi_1, \quad (1)$$

$$\partial_t \phi_2 = \Gamma_2 \boldsymbol{\nabla}^2 \left[ \frac{\partial f}{\partial \phi_2} + \alpha(\phi_1, \phi_2)\phi_1 \right] - \Gamma_2 K_2 \boldsymbol{\nabla}^4 \phi_2, \quad (2)$$

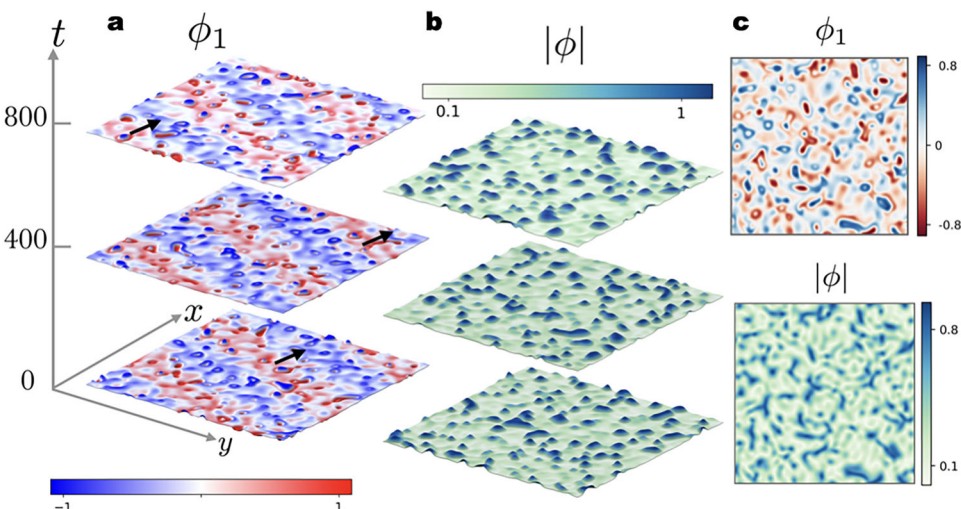

**Fig. 1 | Emergence of effervescence in the dynamical steady-states of the nonlinear NRCH model. a** We introduce nonlinear non-reciprocal interactions between conserved densities $\phi_{1,2}$. The striking feature that emerges on allowing nonspecific nonlinear couplings is the spontaneous nucleation of domains where non-reciprocity nearly vanishes and which we call `reciprocal granules'. These granules coalesce to form phase separated droplets paving the route to spatiotemporal chaos as seen in the snapshots of fields $\phi_{1,2}$ at different times. Snapshots of the density field $\phi_1$, at the times indicated and measured in the units of $\Gamma^{-1}K$, demonstrates the effervescent waves in two dimensions corresponding to $\alpha_0 = 4$

and $\alpha_1 = 5$ (see Supplementary Movie S1). The black arrows indicate the direction of the travelling wave. **b** The amplitude field $|\phi|$ corresponding to the snapshots of the wave in **a**. Note that while **a** shows a clear striped pattern, $|\phi|$ deviates from a constant value in the areas where fluctuating domains are formed. **c** For $\alpha_1$ comparable to $\alpha_0$, we observe an effervescent steady-state without an accompanying travelling wave. Snapshots of $\phi_1$ and $|\phi|$ from simulations corresponding to $\alpha_0 = 2.3$ and $\alpha_1 = 4.6$ (see Fig. S8 in the SI and Supplementary Movie S2 for the full time evolution) show an abundance of bubbles of random shapes.

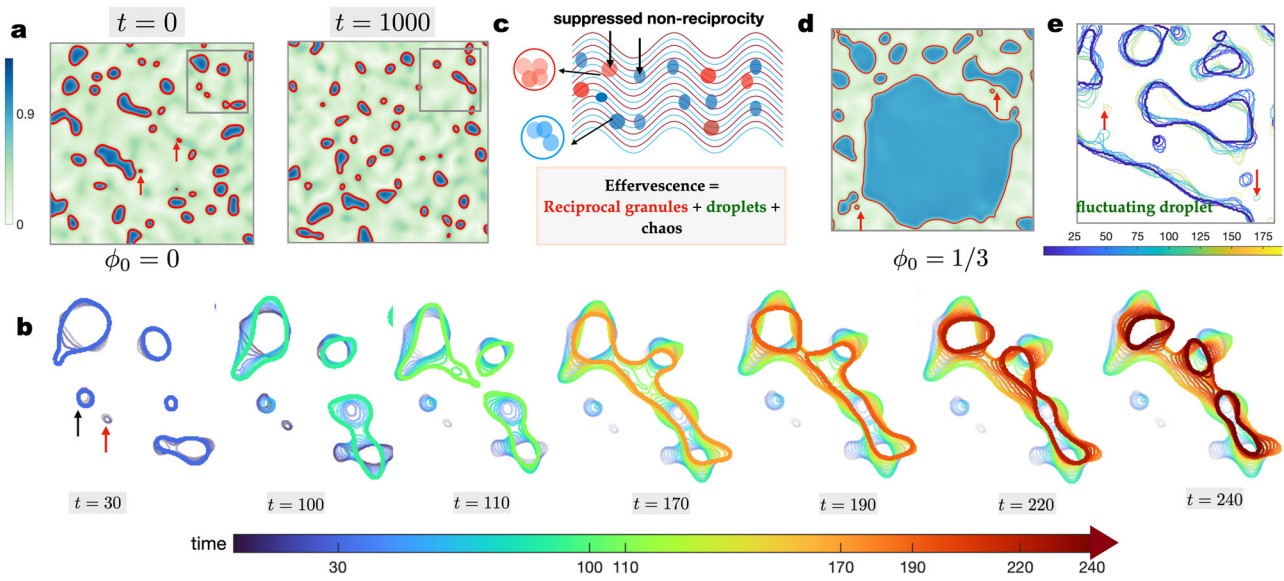

**Fig. 2 | Effervescence, reciprocal granules, and droplets. a** Snapshot of a part of the field $|\phi|$ in the steady state for $\alpha_0 = 5$, $\alpha_1 = 4$ illustrating reciprocal granules (highlighted by the red arrows) and droplets with red contours at $|\phi| = 0.8$. The configuration changes entirely in finite time, as seen in the next snapshot, emphasising the chaotic nature of the creation-annihilation processes that occur continuously in the steady state. **b** To highlight the processes involved in granule and droplet production and annihilation we now focus on the contours only. Each image in the series is produced by superimposing the contours over an interval of time between 30 and 250, in increments of 10 until the time indicated in the label, thus showing the evolution history of the granules and droplets. The final configuration is highlighted using thick lines and the preceding contours are shown in progressively lighter hues. A reciprocal granule -- so called because the strength of net non-reciprocity is suppressed within the domain -- as marked with a red arrow

disappears within $t = 10$. This is a fast process, as it occurs while the surrounding larger droplets are virtually unchanged. Droplets are formed by coalescence events to form larger droplets and dissolve either by shrinking or splitting. The droplets drift and diffuse while their interfaces strongly fluctuate. The series of events shown here are typical of the dynamics seen in the steady state. **c** The definition of *effervescence*, as realised through spatiotemporal chaos in non-reciprocal systems with number conservation showing hallmarks of both chasing dynamics and phase separation. **d** At nonzero mean composition, smaller droplets fuse to form a persistent phase-separated domain (i.e., very large droplet) with pulsating interfaces. This stable domain coexists with reciprocal granules that form and dissolve as described above. **e** Fluctuations of the interfaces in the steady state are illustrated using the same method as adapted for (**b**).

where the non-reciprocal coupling $\alpha(\phi_1, \phi_2)$ is taken to be a function of the fields, as a generalisation of the NRCH model introduced in Ref. 15. Equations (1) and (2) represent a sufficiently general dynamics that one can write down for two species while introducing active terms in the bulk part of $\mu_i$. For consistency, we choose polynomial forms for both the free energy $f$ and the coupling $\alpha$. We will show that the steady-states exhibited by Eqs. (1) and (2) can be broadly classified into the travelling waves, phase separation reported in[15], and a novel spatio-temporally chaotic state studied here, which we call effervescence. For simplicity, we set $\Gamma_1 = \Gamma_2 = \Gamma$ and $K_1 = K_2 = K$ in Eqs. (1) and (2), thus excluding the possibility of encountering a conserved-Turing instability, explored in recent work[38].

All the analytic results and a majority of the numerical results are presented for a particular choice of the free energy density $f$ in the shape of a Mexican hat[39,40] instead of the customary double-well[41],

$$f = -\frac{1}{2}|\phi|^2 + \frac{1}{4}|\phi|^4. \tag{3}$$

At equilibrium the system separates into bulk phases with $|\phi| = 1$. The free energy in Eq. (3) is invariant under rotations $\phi \rightarrow \phi \exp(-i\gamma)$, for an arbitrary phase angle $\gamma$. In the same spirit, we write the following expression for $\alpha$ including the only two terms compliant with the constraint of the imposed symmetry

$$\alpha(\phi_1, \phi_2) = \alpha_0 - \alpha_1 |\phi|^2. \tag{4}$$

Notice that by construction, $\alpha$ vanishes and reciprocity is restored at $|\phi| = \rho_0 = \sqrt{\alpha_0/\alpha_1}$. With these specific choices, Eqs. (1) and (2) can now be written equivalently as an equation for a complex field $\phi = \phi_1 + i\phi_2$

with an amplitude $|\phi| = \sqrt{\phi_1^2 + \phi_2^2}$ and a phase $\theta = \tan^{-1}\phi_2/\phi_1$, see Fig. 1a.

$$\partial_t \phi = \Gamma \mathbf{\nabla}^2 \left[ (-1 + i\alpha_0)\phi + (1 - i\alpha_1)|\phi|^2 \phi - K\mathbf{\nabla}^2 \phi \right]. \tag{5}$$

This additional symmetry is implemented simply to allow analytical calculations. We will show later that the dynamical steady-states resembling chemical turbulence observed when $|\phi|$ and $\theta$ do not relax to stationary profiles are observed for a suitable $f$ and all nonlinear forms of $\alpha$ that allow its local value of non-reciprocity to assume a wide range of values both positive and negative. Despite the similarity of Eq. (5) to the complex Landau-Ginzburg equation[29,42], the phenomenology of Eq. (5) belongs to an entirely different class. The reasons are number conservation[43], and the non-invariance of (5) under the transformation $\phi \rightarrow \phi \exp(\alpha_0 t)$ that does not allow the elimination of $\alpha_0$ through a global rotation.

A numerical solution of Eq. (5) shows that the interplay of the terms with coefficients $\alpha_0$ and $\alpha_1$ leads to the emergence of a form of spatiotemporal chaos in the steady-state, which we have termed effervescence (see Fig. 2). We observe spontaneous creation and annihilation of domains in which the strength of reciprocity is diminished. We categorise these domains into two classes. The first of these is a *reciprocal granule* – so called because it is formed when reciprocal interactions are locally restored and equilibrium-like phase separation resumes. They are spontaneously generated in abundance at the smallest possible length scale set by the surface tension. A reciprocal granule quickly disappears or grows in size via ripening or coalescence, to form larger *droplets* – thus called as they are reminiscent of the minority phase in liquid-liquid phase separation. The terminology is chosen to highlight the granularity of the droplets: the reciprocal

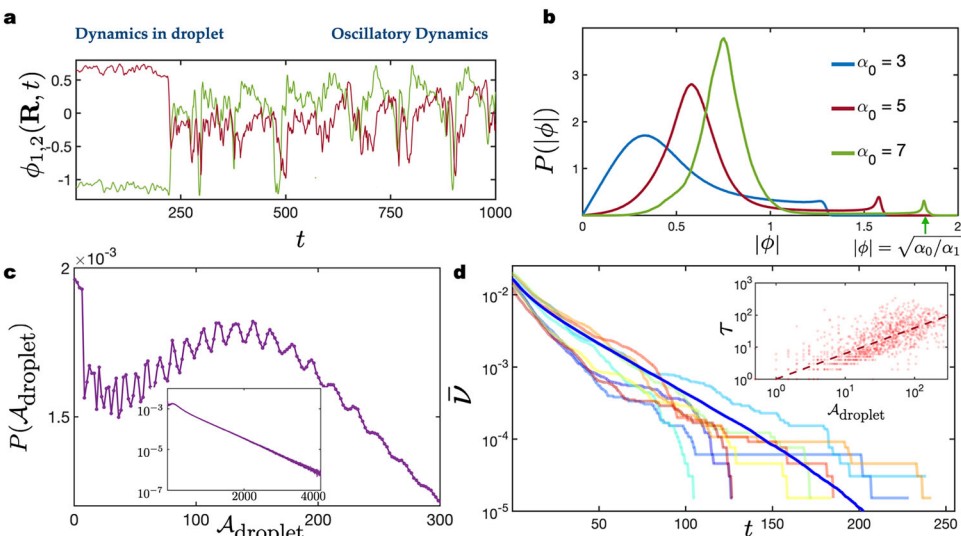

**Fig. 3 | Quantifying effervescence. a** Dynamics within a droplet is slow compared to that in regions where $\alpha$ is much larger, as can be seen from the evolution of $\phi_i$ with time at a fixed position $\mathbf{R}$. **b** Probability distribution for $|\phi|$ in the steady-state, revealing the coexistence of travelling waves (associated with the peak at $|\phi| < 1$) and droplets (linked with the peak at $|\phi| = \rho_0 = \sqrt{\alpha_0/\alpha_1}$) in which the effective non-reciprocal interaction vanishes. **c** Probability distribution for the area $\mathcal{A}_{\text{droplet}}$ (determined by calculating the area within the contours shown in Fig. 2a, b) of the droplets made of reciprocal granules (in units of $q_0^{-2}$). The peak at the origin reveals the existence of a large population of reciprocal granules. The oscillatory motif of the full distribution shows that the droplets are formed as aggregates of a number of similarly sized granules. Inset: the observed exponential decay of the droplet size distribution (as evident in the semi-log scale) verifies the existence of an underlying size selection mechanism. **d** The persistent fraction $\nu$ is plotted against time for $\alpha_0 = 5$ and $\alpha_1 = 4$. $\nu(t)$ is the fraction of space occupied by droplets of age at least $t$, implying that $\nu(0)$ is the initial average coverage of space by droplets. The bold blue line shows the average over 250 different initial configurations while the fainter lines show the time evolution for a few individual cases. The inset shows the correlation between the droplet lifespan $\tau$ and its size $\mathcal{A}_{\text{droplet}}$ (see Methods II for details) and the dashed line is a linear fit to the data.

granules appear to provide discrete units that form the constituents or building blocks of the droplets; see Fig. 2c. The small size and short lifespan of the reciprocal granules distinguish them from longer-lived, larger droplets. Structurally, droplets have pulsating interfaces and they go through shape changes (see Supplementary Movie S2 and Fig. 2) while reciprocal granules are stabilised by surface tension and hence appear predominantly disk-like in two dimensions. Droplets have a more intricate structure, as can be seen in Supplementary Fig. S1, for example. Figure 1 shows droplets enhanced in either species 1 or 2, as well as composites where a droplet enhanced in one species is encapsulated by another enhanced in the other species. The droplets are randomly dissolved at a time scale larger than the granules (see Supplementary Movie S2).

Reciprocal granules and droplets occur in conjunction with the travelling pattern (see Fig. 1a, b), or even, in its absence (see Fig. 1c). This emergent imperfect PT symmetry breaking with local restoration of reciprocal interactions produces two new states, namely an effervescent wave which is a hybrid state with droplets and a travelling pattern, shown in Fig. 1a, b, and effervescence without the travelling pattern, shown in Fig. 1c.

The novelty of the mass conserved dynamics considered in this work is that by tuning the mean composition, $\phi_0 = \langle \phi \rangle$, we gain access to a variety of dynamical behaviour. We defer a detailed discussion to later in the paper. However, we find that upon changing composition, the unsteady waves in Fig. 1a give way to stable macroscopic droplets with pulsating interfaces (see Fig. 2d) coexisting with reciprocal granules and smaller unstable droplets that undergo the same dynamics as described above.

## Quantifying effervescence

The phenomenon of effervescence is described in details in Fig. 3, with a schematic representation shown in Fig. 2c. By construction, $\alpha$ reverses sign when the local value of $|\phi|$, the modulus of the complex field $\phi = \phi_1 + i\phi_2$, increases beyond the threshold $\rho_0$ set by the non-equilibrium parameters, namely, $\alpha(\rho_0) = 0$. The change in sign manifests itself in a change in the sense of the chasing interaction, namely, species 1 chases species 2 at low densities whereas at higher densities 2 chases 1. Within spatial domains where $\alpha$ is nearly zero, the chasing stops locally and the domains grow or shrink, either by losing and gaining matter or by coalescence; see Fig. 3b. This is accompanied by spatiotemporal chaos, as shown in Fig. 3c.

For the effervescent waves, we observe a coexistence between the spontaneously selected values for $|\phi|$ corresponding to the travelling pattern and the reciprocal granules, as observed in the distribution of the order parameter shown in Fig. 3b. The peak of $P(|\phi|)$ occurs precisely at $\rho_0$ [see Fig. 3e] thus confirming that the effervescent granules correspond to a local restoration of PT symmetry.

The granular structure for the domains that restore reciprocal interactions is evident in the droplet size distribution $P(\mathcal{A}_{\text{droplet}})$ shown in Fig. 3d, where we observe a prominent peak for a fundamental reciprocal granule and an oscillatory pattern in the probability for observing larger droplets [Fig. 3d] demonstrating their granularity. A larger droplet can be created by a fusion of two or more reciprocal granules, as seen in Fig. 1b, leading to oscillations in $P(\mathcal{A}_{\text{droplet}})$ (see Fig. 3d, Supplementary Movies S1 and S2). The exponential tail implies that there is a size selection mechanism for droplet size selection and the length-scale associated with it is the average size of the largest droplets.

To associate a timescale with droplet annihilation we probe the surviving fraction $\bar{\nu}(t)$ as a function of time $t$, see Fig. 3f. By taking the mean $\bar{\nu}(t)$ over several initial conditions, we find that on average $\bar{\nu}(t)$ relaxes exponentially with a timescale $\sim 23\, \Gamma^{-1}K$, which should be compared with the time scale of the period of the accompanying wave which is $\sim 300\, \Gamma^{-1}K$. This means that the lifespan of a droplet is typically an order of magnitude smaller than the intrinsic time scale of the travelling wave.

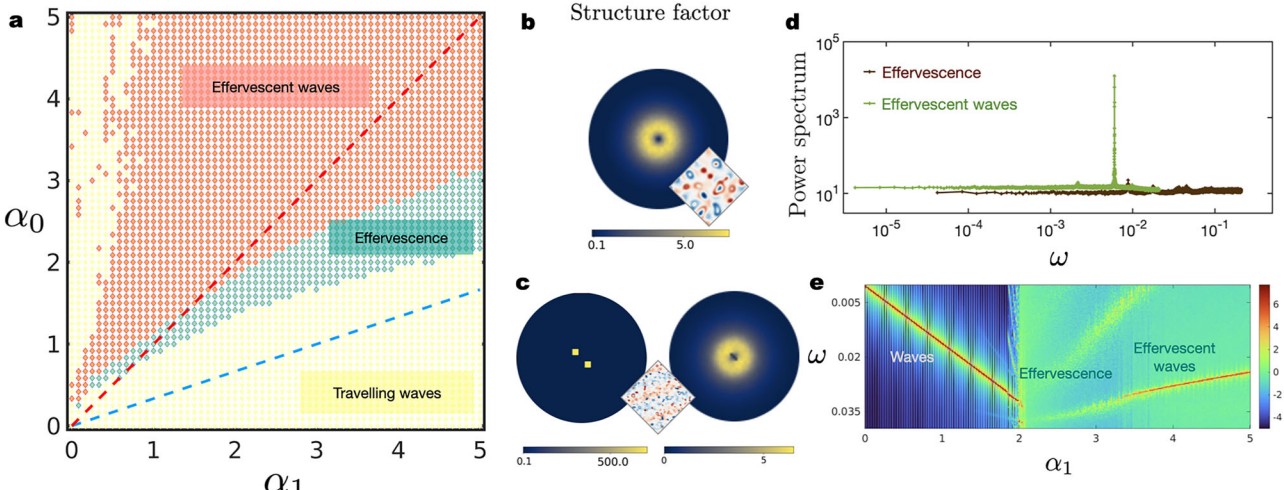

**Fig. 4 | State diagram and correlation functions. a** State diagram for the nonlinear NRCH model in two dimensions. Symbols indicate the points where the simulations were carried out and the colours correspond to the dynamical steady-state of the system. The steady-states are summarised in Fig. 1. Within the dashed red line and the y-axis, travelling waves with vanishing wavenumber are linearly unstable to small perturbations and where droplets with reciprocal interactions are stable (see discussion later in the paper) - the result is the effervescent wave. The region between the dashed blue and red lines, waves are unstable for most wavenumbers and droplets once nucleated are marginally stable - resulting in effervescence. **b** Heat-map for the structure factor $S(\boldsymbol{k})$ in the effervescent state shows fluctuations occurring at all length scales. **c** Same as in Fig. 4b but for effervescent waves - $S$ receives contributions from the waves manifesting in a pronounced peak and the isotropic fluctuations. **d** Power spectrum plotted as a function of the frequency $\omega$ for effervescent waves and effervescence for parameters $\alpha_0 = \alpha_1 = 4$, and $\alpha_0 = 2$, $\alpha_1 = 4.33$, respectively. For effervescent waves, $S(\omega)$ shows a pronounced peak in a nearly constant background of temporal fluctuations. In the effervescent state, the spectrum is nearly constant and indistinguishable from white noise. In both cases, $S(\omega)$ decays at large frequencies. **e** A heat map of $\log[S(\omega)]$ as a function of $\alpha_1$ with a fixed $\alpha_0 = 2.5$. Note that the steep peak for travelling waves disappears in the effervescence case and reappears for effervescent waves. The dispersion is linear for all waves. The black lines are added to emphasise the sharp boundaries between the dynamical steady-states.

## State diagram

The state diagram presented in Fig. 4 is constructed in a single quadrant of the $(\alpha_0, \alpha_1)$ plane, i.e. for $\alpha_{0,1} > 0$ for $\langle\phi_{1,2}\rangle = 0$, where the angular brackets denote spatial averaging. From Eqs. (1) and (2), it is clear that the simultaneous transformations $\alpha_0 \rightarrow -\alpha_0$ and $\alpha_1 \rightarrow -\alpha_1$ merely changes the sign of the fields, $\phi_{1,2} \rightarrow -\phi_{1,2}$. Therefore, it is sufficient to scan the dynamics in two adjacent quadrants only. However, we probe just the one for which both $\alpha_0$, $\alpha_1 > 0$ since for parameters with different signs, the plane wave in Eq. (8) is a stable solution and effervescence does not occur.

For $\text{sign}(\alpha_0) = \text{sign}(\alpha_1)$ and $|\alpha_1| \approx |\alpha_0|$, the steady-state is spatiotemporally chaotic, i.e., the scalar fields show non-repetitive oscillatory patterns. The order parameter used to distinguish the travelling waves from the states with spatiotemporal chaos is the probability to find that $|\phi| < 0.1$ at a given point in space, averaged over space and time. It is given by the cumulative probability $C_{0.1} = \int_0^{0.1} d|\phi| P(|\phi|)$. $C_{0.1}$ is a good descriptor for determining the boundary between the travelling wave states [denoted with yellow markers in Fig. 4a] and the effervescent states (denoted with green markers) as it jumps a few orders of magnitude upon crossing the boundary. Effervescence and effervescent waves [red markers in Fig. 4a] are distinguished from one another by the structure factor, defined as

$$S(\boldsymbol{k}) = \frac{1}{T}\int_0^T dt\, \phi_i(\boldsymbol{k},t)\phi_i(-\boldsymbol{k},t), \qquad (6)$$

where summation over repeated index $i$ is implied, and the power spectrum $S(\omega)$, defined as

$$S(\omega) = \frac{1}{A}\int d^2\boldsymbol{r}\, \phi_i(\boldsymbol{r},-\omega)\phi_i(\boldsymbol{r},\omega). \qquad (7)$$

We observe that $S(\boldsymbol{k})$ is isotropic in the effervescent state (Fig. 4b) and shows distinct peaks corresponding to the wavelength of the travelling wave in the effervescent-waves (Fig. 4c). Moreover, $S(\omega)$

exhibits a nearly constant plateau and is indistinguishable from white noise in the effervescent state, while a pronounced peak appears in addition to the nearly constant background for effervescent waves (Fig. 4d). We also show the heat-map of $\log[S(\omega)]$ as a function of $\omega$ and $\alpha_1$ for fixed $\alpha_0$ (Fig. 4e). We observe that the pronounced peak, where the maxima of $S(\omega)$ is many orders of magnitude larger than the background fluctuations for travelling waves, disappears for effervescence and reappears for effervescent waves. As seen in Fig. 4e, multiple peaks appear but all of them are comparable in magnitude to the background fluctuations. We also note that $S(\boldsymbol{k})$ is fully isotropic in the effervescent state meaning that there are no travelling waves at all, (see Movie S2).

To establish the existence of spatiotemporal chaos we calculate the Lyapunov exponent $\Omega$ in one dimension, as shown in Fig. 5c, d. We observe that $\Omega$ exhibits a distinct jump from a negative value to a positive value, as we change the values of the non-reciprocal parameters. We note that in Fig. 5c, $\Omega$ makes a discontinuous transition as the states change from travelling waves to effervescence. The transition from travelling waves to effervescent waves leaves a smoother signature in Fig. 5d in comparison. These features are also prevalent in the boundaries between the same states in the state diagrams and in the order parameter $C_{0.1}$ (see the SI for further details).

Effervescence is also the defining feature for $f$ given by the Cahn-Hilliard free energy in Eq. (24) (see Figs. S5 and S6 in the SI) for an illustration of the dynamics and a phase diagram sketched by varying the reciprocal coefficient $\chi$ and $\alpha_0$ and find spatiotemporal chaos with droplets and with or without waves.

## Stationary states

The bimodal distribution in Fig. 3e and the formation of long lived spatial domains of low non-reciprocity in Fig. 3f suggests that chaos in the nonlinear NRCH model stems from the coexistence of two different types of dynamics. In the next two sections we will demonstrate that the peak at $|\phi| < 1$ in Fig. 3f corresponds to a travelling wave, and the second peak at $|\phi| > 1$ to the formation of reciprocal granules.

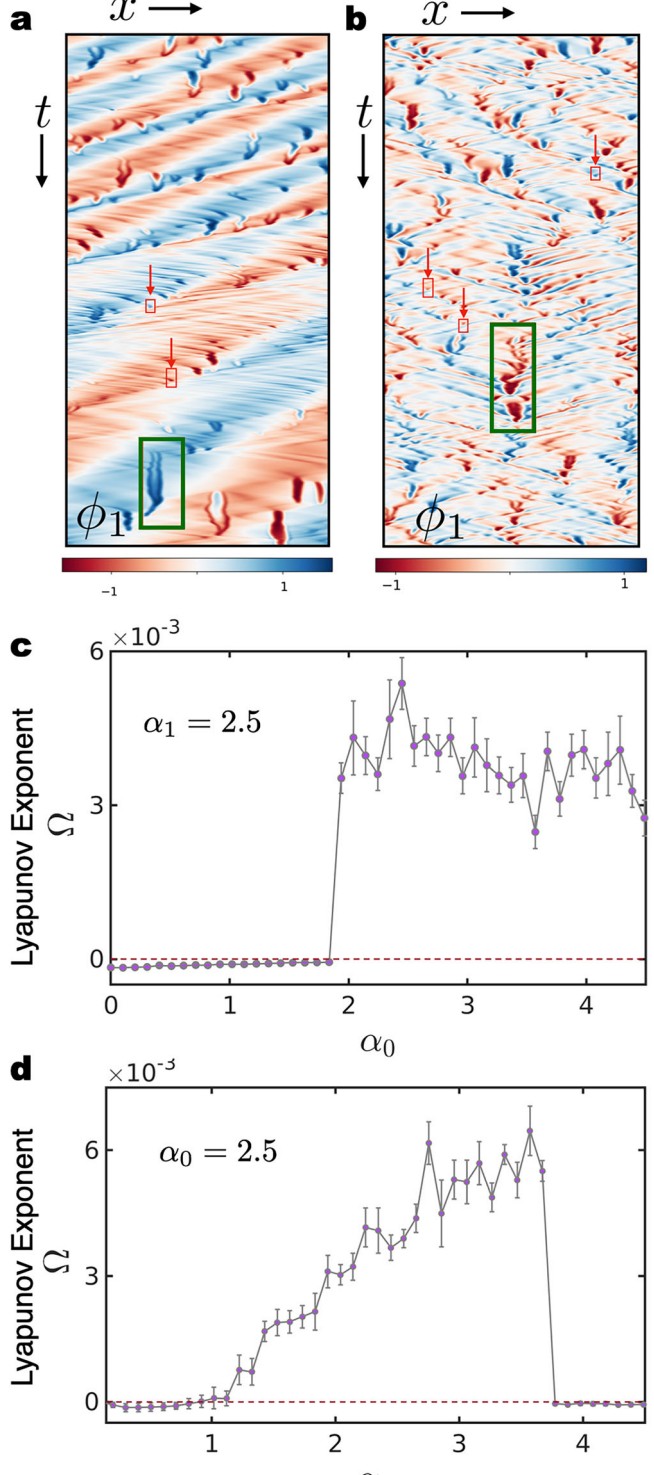

**Fig. 5 | The largest Lyapunov exponent in one dimension. a** Kymograph showing the evolution of the field $\phi_1$ in one dimensional space and time for $\alpha_0 = 4$ and $\alpha_1 = 3$. The propagating bands, and the production and dissolution of reciprocal granules (in red boxes highlighted with red arrows) droplets (in green boxes) are clearly visible. **b** Kymograph of dynamics in one spatial dimension for $\alpha_0 = \alpha_1 = 4$, showing numerous droplets appearing and dissolving. **c** The largest Lyapunov exponent $\Omega$ (for definition see Methods II) as a function of $\alpha_1$, while $\alpha_0$ is fixed at 2.5. $\Omega$ cross from a negative to a positive value corresponding to the $\alpha_1$ for which we see a transition from travelling waves to effervescence. **d** $\Omega$ calculated by varying $\alpha_1$ while $\alpha_0$ is fixed. $\Omega$ crosses from a negative to positive value when a transition occurs from travelling waves to effervescent waves and again changes sign when effervescence gives way to travelling ways.

composition, the homogeneous state is linearly unstable to perturbations irrespective of the values $\alpha_{0,1}$. A trial solution $\phi_q^w$ parameterised by a wavenumber $\boldsymbol{q}$, namely,

$$\phi_q^w(\boldsymbol{r}, t) = \rho_{\boldsymbol{q}} e^{-i(\boldsymbol{q}\cdot\boldsymbol{r}-\omega t)}, \tag{8}$$

is substituted in Eq. (5) to obtain the following expression for the amplitude $\rho_{\boldsymbol{q}}$ and a dispersion relation $\omega(q)$

$$\rho_{\boldsymbol{q}} = \sqrt{1 - \frac{q^2}{q_0^2}}, \; \forall q < q_0, \tag{9}$$

$$\omega(\boldsymbol{q}) = \Gamma q^2 \left[ \alpha_0 - \alpha_1 \left( 1 - \frac{q^2}{q_0^2} \right) \right], \tag{10}$$

where $q_0 = 1/\sqrt{K}$ [see Fig. 6c for a numerical verification of the dispersion relation Eq. (10)]. $\phi_q^w$ in Eq. (8) is defined for all values of $\alpha_0$ and $\alpha_1$ and stable for a wide spectrum of wavelengths as will be shown by a stability analysis around $\phi_q^w$. To perform the stability analysis, we insert a trial solution of the form

$$\phi_q(\boldsymbol{r}, t) = [\rho_{\boldsymbol{q}} + \delta\rho_{\boldsymbol{q}}(\boldsymbol{r}, t)] e^{i(\boldsymbol{q}\cdot\boldsymbol{r}-\omega t)}, \tag{11}$$

in Eq. (5), and derive the effective governing equation for the perturbation $\delta\rho_{\boldsymbol{q}}(\boldsymbol{k}, t)$ in Fourier space with the wavenumber $\boldsymbol{k}$, at the linear order, as has been done for the case of metachronal waves in cilia[44]. The eigenvalues of the resulting linear dynamical equations in Fourier space can then be calculated (see Methods II) and used to isolate the dominant behaviour of the system as reflected in the eigenvalue with the larger real part. Using an expansion up to quadratic order in $\boldsymbol{k}$, and a decomposition of the wavevector into the longitudinal component $k_L = \boldsymbol{k} \cdot \boldsymbol{q}/q$ and the transverse component $\boldsymbol{k}_T = \boldsymbol{k} \cdot (\boldsymbol{I} - \boldsymbol{qq}/q^2)$, we can obtain the dominant eigenvalue, which we present as

$$\lambda(\boldsymbol{k}) = iVk_L - D_L k_L^2 - D_T k_T^2, \tag{12}$$

where the advection velocity $V$, and the longitudinal and transverse diffusion coefficients $D_L$ and $D_T$, are found as

$$V(q) = 2\Gamma q(\alpha_0 - \alpha_1 + 2\alpha_1 q^2/q_0^2), \tag{13}$$

$$D_L(q) = -\Gamma\alpha_1(\alpha_0 - \alpha_1) + \frac{\Gamma\alpha_1^2 q^2(3q^2 - 5q_0^2)}{q_0^2(q_0^2 - q^2)} + \frac{\Gamma q^2(q_0^2 - 3q^2)}{q_0^2(q_0^2 - q^2)}, \tag{14}$$

$$D_T(q) = -\Gamma\alpha_1(\alpha_0 - \alpha_1) + \Gamma\alpha_1^2 q^2/q_0^2 + \Gamma q^2/q_0^2. \tag{15}$$

The travelling wave solutions in Eq. (8) are unstable in the part of the phase space where $D_L < 0$ or $D_T < 0$. First, note that for $\alpha_1 = 0$, $D_L$

Stability analysis around the travelling wave and a local stability analysis about a reciprocal granule will allow us to make predictions about their stability in the parameter space and provide a broad understanding of the state diagram in Fig. 4a.

**Travelling waves.** We start by exploring the possibility of Eq. (5) allowing travelling waves as exact solutions, which is a natural consequence of the number conservation as shown in Fig. 6b. Our choice of the bulk free-energy in Eq. (3) allows us to write down an exact dispersion relation for the travelling waves for $\langle\phi_1\rangle = \langle\phi_2\rangle = 0$. The model is invariant under global phase rotation, i.e., the transformation $\phi \rightarrow e^{i\gamma}\phi$ leaves Eq. (5) unchanged for any arbitrary constant $\gamma$. At this

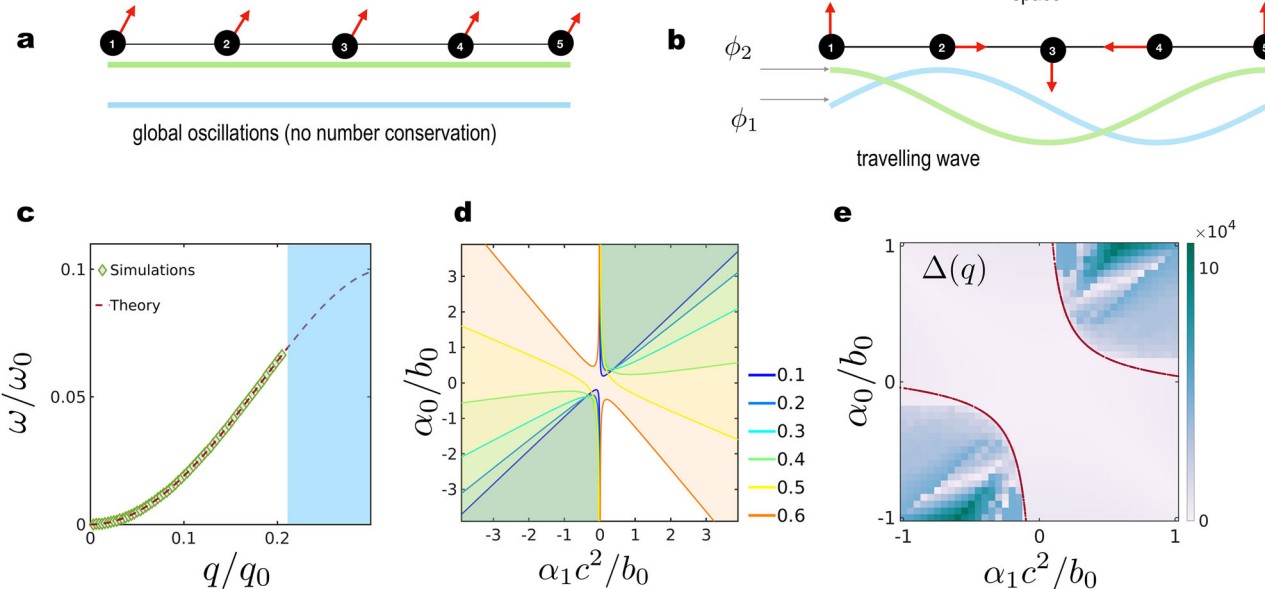

**Fig. 6 | Dispersion and stability of travelling waves in the nonlinear NRCH model. a** In a system without number conservation, a possible dynamical steady-state comprises global oscillations of uniform densities (green and blue line). **b** Number conservation calls for more complicated synchronisation patterns, resulting in a travelling wave state with $\phi_{1,2}$ varying sinuously in space and time (green and blue line); equivalent to $\theta$ rotating at all space points with constant phase difference between two points. **c** Comparison of the dispersion relation for $\alpha_0 = 1$ and $\alpha_1 = -1$ as predicted by theory and as calculated from numerical simulations. The frequency is calculated from simulations with an initial condition $\phi_q^{\mathrm{w}}(\boldsymbol{r}, 0)$ and a Fourier transform of the time series obtained at a randomly chosen fixed position in space. The blue shaded region indicates values of $q/q_0$ for which the travelling wave is unstable to linear perturbations and matches results from simulation. **d** The shaded area represents the unstable region of the phase space for travelling waves with different values of wavenumber $q/q_0$ (shown in legend) in the $(\alpha_0, \alpha_1)$ plane. **e** Theoretical predictions in Fig. 6d are checked using numerical simulations with initial conditions slightly perturbed $\phi_q^{\mathrm{w}}(\boldsymbol{r}, 0)$ [see Eq. (8)]. A heatmap of $\Delta$ [see Eq. (16)] shows a confirmation of the theoretical results. The red line is for $q = 6\pi/L$, where $L$ is the domain size (in units of $q_0^{-1}$).

reduces to the expression that holds for the conserved real Landau-Ginzburg dynamics. The stability of the waves can is controlled by a conserved Eckhaus instability which renders all plane waves with $q > q_0/\sqrt{3}$ unstable to longitudinal fluctuations. $\alpha_1$ alone can be used to tune $D_L$ to negative values at for $q > (1 - \sqrt{2/3})q_0$. For $q \to 0$, $D_L = D_T \approx -\alpha_1(\alpha_0 - \alpha_1)$, indicating that an interplay of $\alpha_0$ and $\alpha_1$ is necessary for destabilising plane waves with the largest wavenumbers. We note that the travelling wave solution has been recently used as a foundation to establish connections between NRCH, flocking active matter theories, and the Kardar-Parisi-Zhang universality class[40].

The stability diagram in the $(\alpha_0, \alpha_1)$ plane is shown in Fig. 6d, with the unstable regions corresponding to $D_L < 0$ being shaded (and the colours correspond to the wave-numbers indicated in the legend). For wavenumbers lower than a threshold value of $q_0/\sqrt{3}$, the unstable region consists of two unconnected pieces in the quadrants $\alpha_0, \alpha_1 > 0$ and $\alpha_0, \alpha_1 < 0$. Above the threshold, the two regions connect to form a single connected unstable region enclosing the origin. The Eckhaus stability criterion at equilibrium thus determines the topology of the stability diagram.

The non-conserved dynamics associated with Eq. (5) can be constructed upon the replacement of $-\nabla^2 \Gamma \to \Gamma$. For vanishing $\alpha_1$, the solution equivalent to Eq. (8) is simply $\rho_q = 1$, $\omega = \alpha_0$, as represented in Fig. 6a. If we think of the steady-state as a field of synchronised phase oscillators, for the non-conserved case we find globally phase synchronised oscillators. Number conservation rules out this globally oscillating state for the NRCH and results in travelling waves instead, as represented pictorially in Fig. 6b.

The results in Eqs. (14) and (15) are checked using numerical simulations with slightly perturbed travelling waves of a chosen wavelength as the initial condition and allowing the system to evolve for a sufficiently long time. The difference between the space averaged amplitude and the amplitude of the input wave defined as

$$\Delta(q) = \frac{1}{A} \int \mathrm{d}^2\boldsymbol{r} |\phi(\boldsymbol{r}, t)| - \rho_{\boldsymbol{q}}, \qquad (16)$$

is calculated in the $(\alpha_0, \alpha_1)$ plane to determine the stability of the travelling waves; see Fig. 6e. Here $A = 4L^2$ is the area of the system. The wavelength of the sinusoidal wave, $q$, and the time periodicity, $\omega(q)$, of the wave at a fixed position in space, are determined using Fourier transforms.

## Reciprocal granules and droplets
In order to check the stability of the reciprocal granules with $\phi \equiv \rho_0 \exp(i\theta_0)$, where $\rho_0 = \sqrt{\alpha_0/\alpha_1}$ and $\theta_0$ is an arbitrary constant phase which we set to zero without loss of generality, we linearise Eq. (5) up to linear order in deviations $\delta\phi$ and $\delta\phi^*$, retaining terms up-to order $q^2$ in the wavenumber $q$. In contrast to the stability analysis in the last section, where we tested the robustness of the travelling waves that are global solutions of Eq. (5), here our reference state exists over a finite area, similar to the concept of 'local' stability analysis[45,46]. The dynamics of the deviations can be recast as equations for $\delta\rho = (\delta\phi + \delta\phi^*)/2$ and $\delta\theta = (\delta\phi - \delta\phi^*)/(2i\rho_0)$ which are fluctuations in the modulus and the phase, respectively.

$$\partial_t \delta\rho = \Gamma\left(\frac{3\alpha_0 - \alpha_1}{\alpha_1}\right)\nabla^2 \delta\rho, \qquad (17)$$

$$\partial_t \delta\theta = \Gamma\left(\frac{\alpha_0 - \alpha_1}{\alpha_1}\right)\nabla^2\theta + 2\Gamma\sqrt{\alpha_0\alpha_1}\nabla^2\delta\rho. \qquad (18)$$

The dynamics of $\delta\rho$ is decoupled from $\delta\theta$, and it relaxes to zero via diffusive processes with a diffusion coefficient that is positive in large

parts of the parameter space in the state diagram in Fig. 4a, thus providing a justification for the peak in the distribution of $P(|\phi|)$ in Fig. 3b at $|\phi| \approx \rho_0$. Marginally stable reciprocal granules are nucleated for $3\alpha_0 > \alpha_1$, however, the $\theta$ field within them fluctuates until $\alpha_0 = \alpha_1$. For $\alpha_0 > \alpha_1$, Eq. (18) suggests that fluctuations in $\theta$ decay via diffusion too, thus stabilising the bulk phase at the reference value $\rho_0$. However, for $\alpha_0 > \alpha_1$, finite-sized stable droplets can be created with constant $|\phi|$ and a field $\theta$ inside that can be determined using if the boundary conditions at the edge of the droplet are known. Our analysis suggests that for $\alpha_0 \gg \alpha_1$, droplets are more stable in comparison to the case where $\alpha_0 \gtrsim \alpha_1$. In simulations, we explore the region of the state diagram beyond that shown in Fig. 4a and find stable droplets submerged in a travelling wave (see Fig. S7 in the SI and Supplementary Movie S3) for which $P(|\phi|)$ consists of two sharp peaks, the first at $|\phi| < 1$ and the other at $|\phi| \simeq \rho_0$.

This analysis provides an understanding of the phase diagram. The effervescent states are found where travelling waves are unstable and the droplets are at least marginally stable. In the effervescent wave state, stable droplets are tossed around in the background of a spatiotemporally chaotic state. In the effervescence state travelling waves over a large large range of wavenumbers are unstable and the droplets that are nucleated are only marginally stable.

## Effect of composition

Finally, we discuss how the steady-state dynamics changes qualitatively as we tune the average composition. So far in the paper, we have fixed the average value of the fields, $\langle \phi \rangle = 0$ since the invariance of the free energy under unitary rotations in the composition plane leads to the exact solution Eq. (8). This invariance is broken when we move away from $\langle \phi \rangle = 0$ and tune the average composition to a nonzero value $\phi_0 \equiv \langle \phi_1 \rangle + i\langle \phi_2 \rangle$. In other words, although Eq. (5) is invariant under a rotation by an arbitrary phase, the constraint $\langle \phi \rangle = \phi_0$ is not. This can also be seen by rewriting (5) in terms of deviations from $\phi_0$, which generates terms that explicitly break the rotational symmetry (see the SI for details). We will show that these terms influence the dynamics thereby enriching the phase behaviour of nonlinear NRCH.

We begin by checking the stability of the homogeneous state with average composition $\phi_0$ to small perturbations. Allowing perturbations around $\phi_0$ in Eq. (5) we obtain the following linearised equations of motion for small deviations $\delta\phi$ using the dynamical matrix $\mathcal{D}$

$$\begin{bmatrix} \delta\dot{\phi} \\ \delta\dot{\phi}^* \end{bmatrix} = \begin{bmatrix} \mathcal{D}_{11} & \mathcal{D}_{12} \\ \mathcal{D}_{21} & \mathcal{D}_{22} \end{bmatrix} \begin{bmatrix} \delta\phi \\ \delta\phi^* \end{bmatrix}, \quad (19)$$

where

$$\begin{aligned} \mathcal{D}_{11} = \mathcal{D}_{22}^* &= \Gamma q^2 \left[ (1 - i\alpha_0) - 2(1 - i\alpha_1)|\phi_0|^2 \right], \\ \mathcal{D}_{12} = \mathcal{D}_{21}^* &= -\Gamma q^2 (1 - i\alpha_1)\phi_0^2, \end{aligned} \quad (20)$$

retaining terms up to order $q^2$. Quartic terms are neglected as they serve the role of stabilising the system at larger wavenumbers. Instabilities at finite $q$ studied in [35,36,38] are ruled out in our model by the choice we have made for the interfacial tension. The eigenvalues $\lambda_\pm$ of the non-Hermitian stability matrix $\mathcal{D}$ in Eq. (19) can be complex. For the stability of the homogeneous state, real parts of $\lambda_\pm = \mathrm{tr}(\mathcal{D})/2 \mp \frac{1}{2}\sqrt{\mathrm{tr}(\mathcal{D})^2 - 4\det(\mathcal{D})}$ should be negative. Complex values for $\lambda_\pm$ imply that a slightly perturbed mixed state develops an oscillatory instability and the system generally evolves into a steady-state that carries signatures of these oscillations like the travelling wave or those summarised in Fig. 1. The determinant and the trace of the $\mathcal{D}$, given as

$$\begin{aligned} \det(\mathcal{D}) &= \Gamma^2 q^4 \left[ (1 - 2|\phi_0|^2)^2 + (\alpha_0 - 2\alpha_1|\phi_0|^2)^2 - |\phi_0|^4(1 + \alpha_1^2) \right], \\ \mathrm{tr}(\mathcal{D}) &= \Gamma q^2 \left[ 2 - 4|\phi_0|^2 \right], \end{aligned} \quad (21)$$

are functions of $|\phi_0|$ only. As $\lambda_\pm$ are determined by the trace and the determinant of $\mathcal{D}$, the stability of the mixed state is determined by $|\phi_0|$ alone. For complex values of $\lambda_\pm$, the real and imaginary parts are

$$\begin{aligned} \mathrm{Re}(\lambda) &= \Gamma q^2 \left[ 1 - 2|\phi_0|^2 \right], \\ \mathrm{Im}(\lambda) &= \Gamma q^2 \sqrt{\left[ (\alpha_0 - 2\alpha_1|\phi_0|^2)^2 - |\phi_0|^4(1 + \alpha_1^2) \right]}. \end{aligned} \quad (22)$$

For $|\phi_0| = 0$, we observe that $\mathrm{Re}(\lambda) > 0$ and $\mathrm{Im}(\lambda) \neq 0$ independently of $\alpha_{0,1}$. At this point, the homogeneous state develops oscillatory instabilities in response to small perturbations. The real and imaginary parts of the eigenvalues are plotted as functions of $|\phi_0|$. Two types of transition points arise as $|\phi_0|$ is changed while keeping other parameters constant (as shown in Fig. 7): Hopf bifurcation where the real parts of a pair of complex eigenvalues change sign, and exceptional point where $\lambda_\pm$ coalesce while the corresponding eigenvectors are parallel[22,47]. We find that upon crossing an exceptional point, the eigenvalues develop imaginary parts[15,16,18,21]. The nonlinearity of the non-reciprocal parameter being considered enhances the richness of the stability diagram as seen in Fig. 7b where multiple exceptional points arise. For $|\phi_0| < 1/\sqrt{2}$, two exceptional points appear at the following values of $|\phi_0|$:

$$|\phi_0|^2 = \frac{2\alpha_0\alpha_1}{3\alpha_1^2 - 1} \pm \frac{\alpha_0\sqrt{\alpha_1^2 + 1}}{3\alpha_1^2 - 1}. \quad (23)$$

A third possibility occurs for $|\phi_0| = 1/\sqrt{2}$ and $\alpha_0 = \alpha_1 \pm \frac{1}{2}\sqrt{1 + \alpha_1^2}$ where $\mathrm{Re}(\lambda) = \mathrm{Im}(\lambda) = 0$. Finally, a pair of real eigenvalues could both change signs signalling an instability where perturbations grow and lead to the formation of a bulk separated state indicated by the grey circles in Fig. 7.

We now look at the dynamical steady-states in numerical simulations. Moving away radially from the centre $|\phi_0| = 0$ in the composition space $(\phi_1, \phi_2)$, the currents driving the phase separation dominate over the non-reciprocal interactions. For $\alpha_0 = \alpha_1 = 4$, effervescence gives way to a predominantly phase-separated state with domains spanning the system size, and with pulsating interfaces (see Fig. 8a, b and Supplementary Movies S4 and S5).

To determine the crossover from the dynamics illustrated in Fig. 1 to that in Fig. 8 we vary $\mathrm{Re}(\phi_0)$ keeping $\mathrm{Im}(\phi_0) = 0$, and evaluate two quantities that determine the nature of the steady-state. The first is the average deviation from the homogeneous state $\delta\bar{\phi}_1 \equiv \frac{1}{A}\int d^2r \left[\phi_1 - \mathrm{Re}(\phi_0)\right]$ to identify the points where the homogeneous system is unstable. To identify the oscillatory steady-state[45] we calculate the area enclosed in $(\phi_1, \phi_2)$ space by the boundary enclosing the trajectory $(\phi_1(r, t), \phi_2(r, t))$ at a constant $r$, namely, $\sigma(r) = \frac{1}{2}\oint \phi_1 d\phi_2 - \phi_1 d\phi_2$. For steady travelling waves we find that the fields at each point in space evolve to approach a limit cycle, and $\sigma(r)$ is the area of the limit cycle in the composition plane, which probes the degree of time-reversal symmetry breaking[48]. For instance, $\sigma = 2\pi$ for $\phi_0 = 0$ as the limit cycle in this case is the unit circle, while for effervescence the trajectory is a periodic and area-filling (see Fig. S9 in the SI for an example). $\delta\bar{\phi}_1$ is non-vanishing when the homogeneous state is linearly unstable as seen in Fig. 8c, e (the threshold of instability $\mathrm{Re}(\lambda_\pm) > 0$ is marked with a grey line). $\sigma$ shows interesting deviations from predictions of the linear theory as expected in this highly nonlinear system. The dashed blue lines in Fig. 8c, d mark the points where $\sigma$ changes rapidly signalling the transition from effervescent states−with or without a wave−to bulk phase separation with pulsating interfaces. Note also that the oscillations persist even in the regions where the eigenvalues are real and can be attributed entirely to nonlinear effects. The field $\phi_1$ is shown at the points indicated with an

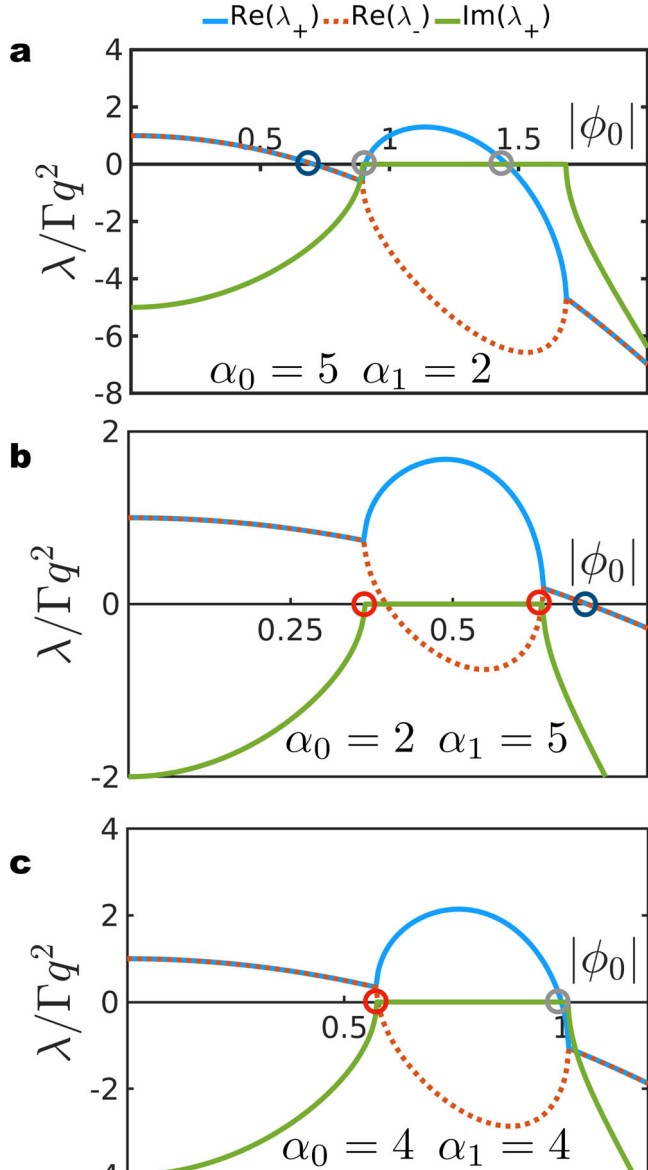

**Fig. 7 | Eigenvalues as functions of the average composition. a** The real and imaginary parts of the eigenvalue $\lambda_\pm$ determining the stability of the homogeneous state with composition $\phi_0$. $\lambda_\pm$ is plotted as a function of $|\phi_0|$ only (due to global phase invariance of the dynamics) for fixed values of $\alpha_0$ and $\alpha_1$. The emergence of an exceptional point is highlighted by the red circles. The Hopf bifurcation points are encircled in blue. **b**, **c**, Same as in panel a, but for different values of $\alpha_{0,1}$.

arrow in Fig. 8d, f depicting the qualitative changes in the dynamics. We reach similar qualitative features in one dimensional dynamics as well (see SI for further details).

## Ubiquity of effervescence

Using the special forms for $f$ and $\alpha$ (see Eqs. (3) and (4)) enables the derivation of an exact travelling wave solution Eq. (8), helps to indicate clearly that the reciprocal granules appear at $|\phi| = \rho_0$, and affords us the analytically tractable stability analysis discussed above. In this section, we illustrate that by choosing generalized polynomial forms for $\alpha$ and $f$ with reduced levels of symmetry, the general features of our results remain unchanged, i.e., most nonlinearities are sufficient to produce effervescence. In other words, the statistical properties of the dynamical steady states in nonlinear NRCH are independent of the detailed expressions of $f$ and $\alpha$. This helps to prove that the

phenomena reported here do not rely on specific fine-tuning for experimental verification, and can be observed in a wide variety of systems provided nonlinear non-reciprocity can be realised in the system. We also note also that changing the mean composition is akin to allowing a larger variety of terms in the interaction (see SI) meaning that the previous section already provides indirect proof of the ubiquity of effervescence. The form for $f$ is

$$f = -\frac{\phi_1^2}{2} + \frac{\phi_1^4}{4} - \frac{\phi_2^2}{2} + \frac{\phi_2^4}{4} + \chi\phi_1\phi_2. \tag{24}$$

At equilibrium, $f$ leads to bulk phase separation with $\phi_{1,2} = \pm 1$ and $\chi$ is the linear reciprocal coupling coefficient[41]. We use a completely general form for $\alpha$ including all possible terms in the spirit of a Landau expansion, as follows

$$\begin{aligned}\alpha = \; & \alpha_0 - (\alpha_1 + \alpha_2)\phi_1^2 - (\alpha_1 - \alpha_2)\phi_2^2 \\ & + \alpha_3\phi_1\phi_2 + \alpha_4\phi_1 + \alpha_5\phi_2.\end{aligned} \tag{25}$$

The existence of the terms in (25) is illustrated using established active matter theories, see Methods. The ingredients required to arrive at these terms are activity in a suitable form, multi-species number-conserving mixtures, and general non-linearities that allow $\alpha$ to change sign within the dynamically relevant range of densities. We note that our approach is similar to those followed in refs. 43,49 and supplements their findings. We solve Eqs. (1) and (2) with the above choices for $f$ and $\alpha$. In Fig. 9a, b, we illustrate the fields $\phi_{1,2}$ and $\alpha$, respectively, at steady-state for $\alpha_0 = 5$, $\alpha_1 = 1$, $\alpha_2 = -0.25$, $\alpha_= 0.1$, $\alpha_4 = -4.5$, and $\alpha_5 = -0.5$. The steady-state is a fluctuating travelling wave coexisting with droplets formed by phase separation dynamics.

We find additional features, such as density fluctuations travelling in the transverse direction to the travelling wave, and clustering of droplets at certain preferred phases of the wave (see Fig. 9a). The transverse waves are a particular feature of this particular choice of parameters and not generic to effervescence, as it has also been reported in simulations of linear NRCH, specifically in an active-passive mixture[49].

To further test the robustness of effervescence, we perform some simulations with random values associated with the coefficients in the non-reciprocal terms. We fix $\alpha_0$ at the constant value indicated in Fig. 9c–e, and draw $\alpha_1, ..., \alpha_5$ from a Gaussian distribution with vanishing mean and standard deviation 4 to generate random realisations of nonlinear $\alpha$ in Eq. (25). Figure 9c shows the emergent distribution of $\alpha$, $P(\alpha)$, in two dimensions for 50 realisations and the bold purple line shows the average. At steady state $\alpha$ fluctuates and evolves to a broad distribution with one peak at a positive value, another at a negative value, and significant weight at $\alpha = 0$. The phenomenon that we observe does not need sign changing $\alpha$ in the strict sense conveyed by Eq. (4) where there is a value $|\phi| = \alpha_0/\alpha_1$ where $\alpha$ vanishes. In fact, this can be achieved by any nonlinear $\alpha$, i.e., a general polynomial form of $\alpha$. Consider an example, for $\alpha = \alpha_0 + \alpha_5\phi_2$, substituting $\phi_2 = \sin(qx)$, we find that $\alpha_0$ vanishes at positions $x = -\sin^{-1}(\alpha_0/\alpha_5)q^{-1}$. This means that if the fields have the propensity to oscillate then a local change in the sign of $\alpha$ is possible with this simple nonlinearity. This argument can be extended to all allowed terms in $\alpha$.

We have run simulations in one dimension for an ensemble of $\alpha$ with 500 realisations at each selected value of $\chi$ between $-1$ and 1. The outcome at steady state is summarised in Fig. 9d. The analysis is repeated at a smaller value of $\alpha_0$ with an ensemble of size 100 (Fig. 9e) to further test the robustness of effervescence. In both sets of simulations, we find that effervescent steady states occur in the majority of cases, indicating that when a mechanism capable of driving effervescence exists through the incorporation of nonlinear terms in $\alpha$, it appears with high statistical probability.

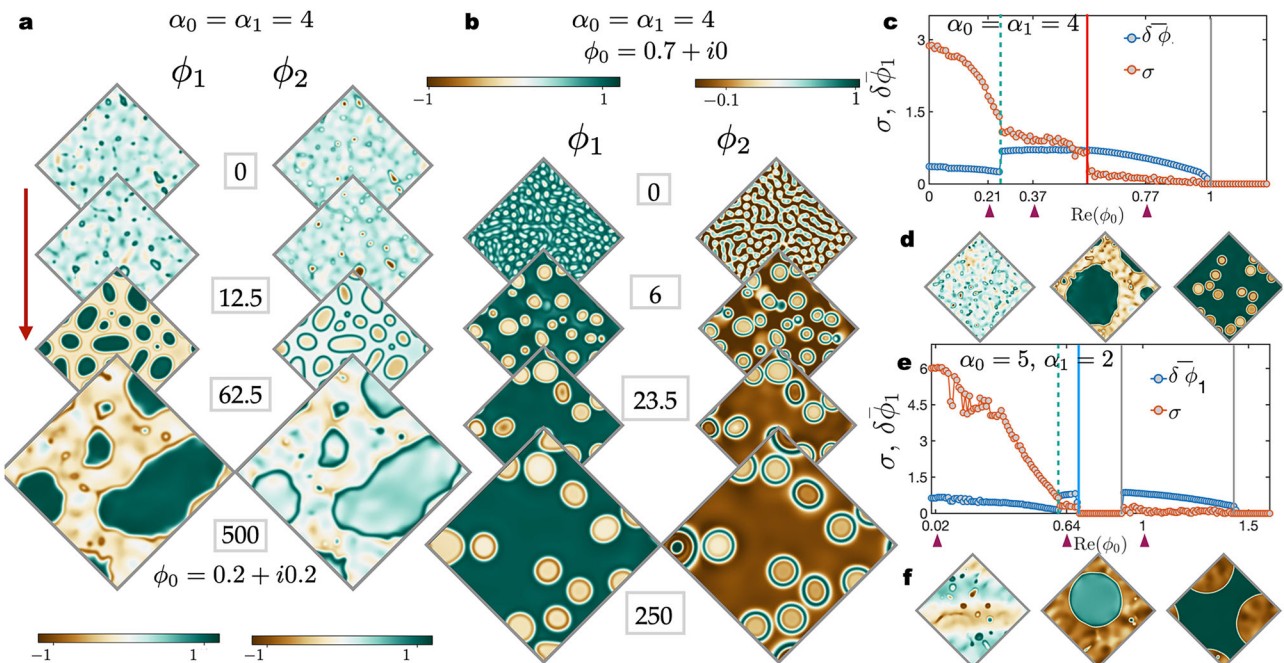

**Fig. 8 | Average composition determines steady-state dynamics. a** Snapshots depicting the evolution of the density fields in numerical simulations for $\phi_0 = 0.25 + i0.25$ and $\alpha_0 = \alpha_1 = 4$ show that effervescence assumes modified forms for $\phi_0 \neq 0$ (see Supplementary Movie S4). At steady-state we find bulk phase separation with pulsating interfaces and spontaneous droplet creation. **b** As the average concentration is changed to $\phi_0 = 0.7$, the dynamics initially resembles phase separation at equilibrium, with oscillations occurring at a later stage. Phase separation is arrested, as can be seen from the droplets at $t = 23.5$ that persist at all later times without coarsening to a steady-state. **c** Numerical verification of the linear stability analysis for $\alpha_1 = \alpha_0 = 4$. The order parameters $\sigma$ and $\delta\bar{\phi}_1$ vary as predicted by the stability analysis. Until the point marked with a dashed line, we find effervescence without travelling waves as seen in the first snapshot of (**d**). which give way to dynamics shown in panel **a**, finally giving way to that in (**b**). **e** Similar behaviour to **c** but for $\alpha_0 = 5$ and $\alpha_1 = 2$. The times are indicated next to the snapshots and measured in units of $\Gamma^{-1}K$.

## Non-reciprocal interactions in real physical systems

From a phenomenological perspective, when the symmetries and conservation laws provide conditions for non-reciprocity to be realised, then there is nothing preventing the non-reciprocal coupling to take on a nonlinear form. This means that the spirit of a Landau expansion can be applied both to the reciprocal part of the chemical potential and to its non-reciprocal part. In this sense, nonlinear terms will represent non-reciprocal interactions that require the presence of additional structure, such as three-body interactions, allostery-like effects where dimers and trimers made of one species interact differently as compared to two or three single units, or interactions mediated by other species as present in quorum-sensing systems. While the mechanistic details of these different microscopic realisations will be different, one can naturally expect them all to give rise to generic nonlinearities provided in Landau-like expansions, as we show in the Methods section below for two specific examples.

Non-reciprocal interactions emerge quite naturally in active mixtures[8–10,12,18–20,50] as well as systems with hydrodynamic interactions[4], vision-cone interactions[6,7,51], mass-conserved reaction-diffusion systems with more than one conserved density developed to model Min-protein oscillations[37,52], and mixtures of active and passive Brownian hard spheres[31]. A multi-component mixture of Janus colloids[35] is a promising model to build in non-reciprocal collective dynamics systematically, for example mixtures of active colloids coupled to reaction cycles show complex oscillations and chiral spatial patterns[50]. Active colloids interact by sensing self-generated chemical gradients and the chemical activity depends on the local concentration of the colloids in the simplest case through Michaelis-Menten kinetics as implemented in ref. 53 for one species. Mixtures of quorum-sensing self-propelled particles have reported patterns similar to what we observe in our model[36,54]. In experiments, a mixture of DNA complexes and active suspension of micro-tubules have shown nucleation of

droplets that are dynamical in nature and close in appearance to the patterns reported here[55]. Similar proposals have been put forward in the context of active gels[56], as well as active passive-mixtures[57]. Finally we note that nonlinear non-reciprocity is the norm in multi-component mixtures as observed in recent papers[43,49].

## Discussion

We have introduced a model with nonlinear non-reciprocal interactions between two species, and shown that its dynamical steady-states are characterised by a new type of spatiotemporal chaos that arises due to the imperfect breaking of PT symmetry within fluctuating spatial domains where the reciprocity in interactions is temporarily restored and phase separation dominates. This effect produces the startling phenomena of effervescence and effervescent travelling waves. Choosing the free energy in the shape of a Mexican hat[40] and enforcing rotational symmetry on the dynamics leaves us with a single contribution to the nonlinear non-reciprocal coefficient of strength $\alpha_1$ (see Eq. (4)) that competes with the linear term $\alpha_0$ first studied in refs. 15,16, thereby destabilising both the travelling waves and the bulk phase-separated state producing a hybrid state with features inherited from both states, which results in the bimodal distribution of $P(|\phi|)$ as seen in Fig. 3b.

We show numerically that effervescence requires only non-specific nonlinear interactions, which can allow the non-reciprocity to change sign dynamically in a number-conserving active scalar mixture, and is defined as a state that contains reciprocal granules and droplets undergoing chaotic dynamics. We quantify this definition in Fig. 3 where we have looked at the distribution of area for spatial domains of low non-reciprocity to find a prominent peak at the smallest possible area and an oscillatory decay due to the merging of the granules to form droplets. A statistical analysis of the steady-state dynamics of the system shows that effervescence is the dominant

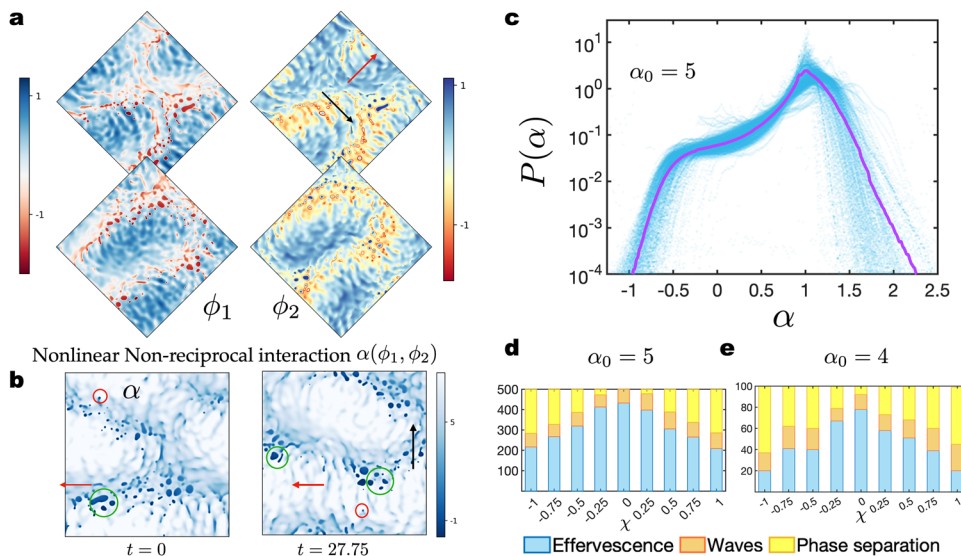

**Fig. 9 | Effervescence is ubiquitous. a** Snapshots of the fields $\phi_{1,2}$ at two separate times showing travelling waves moving in the direction of the black arrow. Fluctuations transverse to the travelling wave propagate along the red arrow. **b** The dynamics follows from the restoration of reciprocity in fluctuating spatial domains, as evident in the snapshots of the non-reciprocal coupling $\alpha$. In the darker regions $\alpha$ is close to zero meaning that free-energy-driven interactions dominate inside these regions promoting phase-separation. Larger $\alpha$ leads to chasing dynamics. The constraint of rational invariance in the composition plane is lifted for these simulations, a feature visible in the formation of the reciprocal granules and droplets preferentially at certain phases of the travelling wave. **c** $\alpha$ is a fluctuating field whose probability distribution $P$ evolves to a stationary distribution independent of its parametrization. Drawing parameters randomly from a Gaussian with variance 4 and mean zero, we find on an average a universal form for $P(\alpha)$ for steady-states that show effervescence. $P$ has significant weight at negative values, and it is bimodal with peaks at positive and negative values. **d, e** Effervescence is ubiquitous - it is the dominant steady-state when non-linear non-reciprocal interactions are allowed as seen in the statistics of observed steady states with varying reciprocal interaction coefficient $\chi$.

steady-state in generic nonlinear mass-conserving dynamics; see Fig. 9. We also observe that the chaotic solutions exhibit fluctuations that can act as a background effective white noise due to the non-reciprocal nonlinearities, similarly to the case of the Kuramoto-Sivashinsky equation[58].

Our work sheds light onto the rich and complex behaviour that can arise in minimal models of number-conserving scalar active matter systems with non-reciprocal interactions showcasing a range of fascinating dynamical states. Some motifs, such as pulsating interfaces, dividing droplets, production of bubbles, generation of noise, are similar to those seen in Turing systems[59], complex Landau-Ginzburg dynamics[29], mass-conserved reaction-diffusion systems[34] with one or more conserved fields, and active-phase separation[33]. However, there are important differences distinguishing effervescence from existing field theories, of which we discuss a few here. The generation of droplets in two dimensions bears similarities to the bubbly phase separation explored in active Model B+[33]. However, in contrast with active Model B+ where interfacial activity in the presence of stochastic forcing drives the generation of bubbles ruling them out in one dimension, we find effervescence in one dimension and in deterministic partial differential equations, thus distinguishing the two theories. Effervescence is not a form of phase turbulence—see for example[60]—where the phase can be enslaved to the amplitude[61] and the distribution of fluctuations of the amplitude about its stationary value is Gaussian, which is in stark contrast with the bimodal structure seen in Fig. 3b. Furthermore, composition change leads to a crossover from seemingly distinct classes of dynamical states (see Fig. 8), which is inaccessible in reaction-diffusion systems without mass conservation. Finally, we contrast the subtle size selection that appears in the exponential tail of the distribution of the droplets, in addition to the granular nature of the droplets, with the size regulation controlled by Rayleigh-type flux-driven instability in[62–64].

We hope that our work will pave the way for new studies of the role of non-reciprocity in colloidal systems with tunable interactions[6]

or in the field of swarm robotics[65]. We have established that effervescent steady-states will be a common feature of future experiments with non-reciprocally interacting scalar densities. Finally it would be interesting to explore the dynamics of a wet effervescent suspension—i.e., nonlinear NRCH coupled to a momentum conserving fluid—in light of several studies that show that non-equilibrium local activity can drive turbulent flows, examples of which include chaotic patterns in ciliary carpets[4] and bacterial suspensions[66], defect dynamics in[67–69], and protein propagation patterns on curved membranes[70].

## Methods
### Details of the numerical simulation
The simulations are carried out with a pseudo-spectral method that uses the Fourier basis for decomposition[15]. The time stepping ensures that the linear terms in $\phi$ are treated implicitly while the nonlinear terms are evaluated explicitly at each time step and act like source terms. Given the field configuration at a given instant of time, the field at the next instant of time is calculated. A further Fourier transform yields the fields in real space. The process is repeated until the steady-state is reached as determined by a stationary probability distribution or power spectral density. As we observe spatiotemporal chaos in the steady-state, we remove the modes with large wavenumber in the Fourier transform to avoid errors due to de-aliasing. For the time stepping we use a method that is a hybrid of the integrating factor and a two-stage Runge-Kutta method[71]. We have further checked our results in one spatial dimension using the Python-based online platform Dedalus[72], using other time stepping methods and have found that the results are robust.

### Microscopic justification of nonlinear NRCH
We illustrate using established theoretical models that the key ingredients required to arrive at the nonlinear NRCH are activity, general nonlinearities, and mixtures with at least two conserved components. In the two examples that we present here we consider the following

sources of activity · (A) self-propulsion and quorum sensing, (B) catalysis. The fields that we will work with are $\rho_a$ (where $a \in \{1, 2\}$), which represent the number density fields of self-propelled particles in (A) and catalysts densities in (B). In addition to density fields we will consider non-conserved polarisation fields in (A) and non-conserved chemical fields in (B), which will be eliminated to obtain the closed dynamics of the densities alone.

**Quorum-sensing mixtures.** We consider two species of self-propelled agents whose speeds depend on the local densities of the active particles, thereby describing phenomena, such as the formation of bio-films, etc. The conserved densities satisfy the equations of motion

$$\partial_t \rho_a = -\nabla \cdot \boldsymbol{j}_{a,\text{QS}} \equiv -\nabla \cdot (v_a \boldsymbol{n}_a), \qquad (26)$$

where the self-propulsion speeds $v_a$ are functions of $\rho_{a,b}$, thus describing a quorum-sensing active mixture. In the next step, we will eliminate the non-conserving orientation fields $\boldsymbol{n}_{a,b}$ to obtain the closed dynamics for the coupled densities only. For particles without aligning interactions, the orientation is a fast variable and can be enslaved to the density dynamics as follows[73]

$$\boldsymbol{n}_a = -\frac{1}{2D_{ra}} \nabla(v_a \rho_a). \qquad (27)$$

Identifying $v_a/2D_{ra}$ as the mobility and $\rho_a v_a$ as a non-equilibrium chemical potential we will show that a Taylor series expansion about the average values $\bar{\rho}_a$ of the fields $\rho_a = \bar{\rho}_a + \phi_a$ generates all the terms that we introduce in Eq. (25) of the main text. The expressions for the coefficients are obtained in terms of derivatives of the speeds which are expressed using a compact notation as $v_{aab}^{(a)} \equiv \frac{\partial^3 v_a(\bar{\rho}_a, \bar{\rho}_b)}{\partial \rho_a \partial \rho_a \partial \rho_b}$, etc. The terms generated are

$$
\begin{aligned}
\mu_1 \equiv \rho_1 v_1 = &\ \frac{\delta F_1}{\delta \phi_1} + \phi_2 \Big[ \bar{\rho}_1 v_2^{(1)} + \bar{\rho}_1 v_{22}^{(1)} \phi_2 + (\bar{\rho}_1 v_{12}^{(1)} + v_1^{(1)}) \phi_1 + \bar{\rho}_1 v_{222}^{(1)} \phi_2^2 \\
&+ (\bar{\rho}_1 v_{112}^{(1)} + v_{22}^{(1)}) \phi_1 \phi_2 + (\bar{\rho}_1 v_{122}^{(1)} + v_{11}^{(1)}) \phi_1^2 \Big].
\end{aligned}
$$
$$(28)$$

where the effective free energy $F_1$ is

$$
\begin{aligned}
F_1 = &\ \left( \bar{\rho}_1 v_1^{(1)} + \bar{v}^{(1)} \right) \frac{\phi_1^2}{2} + \left( \bar{\rho}_1 v_{11}^{(1)} + \bar{v}_1^{(1)} \right) \frac{\phi_1^3}{3} \\
&+ \left( \bar{\rho}_1 v_{111}^{(1)} + \bar{v}_{11}^{(1)} \right) \frac{\phi_1^4}{4}.
\end{aligned}
$$
$$(29)$$

A similar expression holds for $\rho_2 v_2$. We can write down expressions for the non-reciprocal interaction coefficients $\alpha_i$ as follows

$$
\begin{aligned}
\alpha_0 &= \bar{\rho}_1 v_2^{(1)} - \bar{\rho}_2 v_1^{(2)}, \\
\alpha_1 &= \bar{\rho}_1 v_{222}^{(1)} + \bar{\rho}_1 v_{122}^{(1)} + v_{11}^{(1)} - \bar{\rho}_2 v_{111}^{(2)} - \bar{\rho}_2 v_{112}^{(2)} - v_{22}^{(2)}, \\
\alpha_2 &= \bar{\rho}_1 v_{222}^{(1)} - \bar{\rho}_1 v_{122}^{(1)} - v_{11}^{(1)} - \bar{\rho}_2 v_{111}^{(2)} + \bar{\rho}_2 v_{112}^{(2)} + v_{22}^{(2)}, \\
\alpha_3 &= \bar{\rho}_1 v_{112}^{(1)} + v_{22}^{(1)} - \bar{\rho}_2 v_{122}^{(2)} - v_{11}^{(2)}, \\
\alpha_4 &= \bar{\rho}_1 v_{22}^{(1)} - \bar{\rho}_2 v_{11}^{(2)}, \\
\alpha_5 &= \bar{\rho}_1 v_{12}^{(1)} + v_1^{(1)} - \bar{\rho}_2 v_{12}^{(2)} - v_2^{(2)}.
\end{aligned}
$$
$$(30)$$

We note that the necessary condition for effervescence, considering just $\alpha_0$ and $\alpha_1$ is that the signs of the two should be the same, and we find that tuning the average compositions $\bar{\rho}_1$ and $\bar{\rho}_2$ is a simple way to achieve that.

**Chemically active mixtures.** We will now consider a chemically active system where the fields $\rho_a$ represent number densities of enzymes or

catalysts $E_a$ that produce other chemicals $C_i$ with number density $c_i$. The species $E_a$ do not undergo a chemical transformation themselves and follow gradients of the the species $C_i$ such that dynamics can be written as

$$\partial_t \rho_a = -\nabla \cdot \boldsymbol{j}_{a,\text{PS}} = -\nabla \cdot \left( \rho_a \sum_i \Gamma_{ai} \nabla \delta c_i \right), \qquad (31)$$

where $\Gamma_{ai}$ is a or chemotactic[74] or diffusiophoretic[75] mobility, and $c_i$ is the chemical field of the species $C_i$. The chemical fields are produced by the catalysts, and in addition, they decay at a constant rate $\tau_i$ while being involved in other chemical reactions. Consequently, the chemical concentrations obey the following flux balance conditions

$$\partial_t c_i + D_i \nabla^2 c_i = \sum_a \kappa_{ia} \rho_a - f_i(\{c_j\}), \qquad (32)$$

where $\kappa_{ia}$ is the rate at which $E_a$ produces $C_i$, $f_i$ represents the contribution of other reactions that lead to production or depletion of $C_i$, and $D_i$ is the diffusion coefficient associated with species $C_i$. We assume that the local value of $C_i$ is determined by its production and consumption, i.e., these processes dominates over its diffusion. In general the consideration of the gradients terms in Eq. (32) lead to other effects, such as non-reciprocal surface tension[35] that are not of immediate relevance in regards to the terms that we are interested in. For any nonlinear functions $f_i$ of the chemical concentrations, we set the R.H.S. of Eq. (32) to zero, formally solve for $c_i = \mathcal{F}(\phi_1, \phi_2)$, substitute it in Eq. (31), and expand the fields until the third order in deviations $\phi_a$, defined via $\rho_a = \bar{\rho}_a + \phi_a$, to obtain the non-reciprocal coefficients, as we will illustrate below with an example.

Consider the reaction where two molecules of $C_1$ combine to produce another chemical $C_2$. $C_1$ is produced through catalysis by both $E_1$ and $E_2$, as shown below

$$
\begin{aligned}
C_1 + C_1 &\rightarrow C_2, \\
S_1 + E_1 &\rightarrow C_1 + E_1, \\
S_2 + E_2 &\rightarrow C_1 + E_2.
\end{aligned}
$$
$$(33)$$

The equation can be written as

$$-r_1 c_1^2 + \kappa_1 \rho_1 + \kappa_2 \rho_2 = 0. \qquad (34)$$

The catalytic reaction rates $\kappa_i$ can be functions of densities of the $S_i$. We assume that the substrate is available in sufficient amounts and we are in the reaction limited (i.e. saturated) part of the Michaelis-Menten curve. Other consideration would be other nonlinear effects that would change the expressions that we present here. We obtain an expression for $c_1$ as follows

$$c_1 = \sqrt{\bar{\kappa} + \frac{\kappa_1 \phi_1}{r_1} + \frac{\kappa_2 \phi_2}{r_1}} \equiv \sqrt{\bar{\kappa}} + \delta c_i, \qquad (35)$$

where

$$\bar{\kappa} = \frac{\kappa_1 \bar{\rho}_1 + \kappa_2 \bar{\rho}_2}{r_1}. \qquad (36)$$

The expression for $c_i$ in Eq. (35) is valid in the regime when the fluctuations $\phi_i$ are sufficiently small such that the quantity $c_i$ representing actual densities is always positive. The densities $\rho_i$ are sensitive only to gradients $\nabla c_i$ in Eq. (31) implies that we can replace it by $\nabla \delta c_i$, a quantity that can be positive or negative depending on whether $c_i$ locally exceeds the mean value due to local production or consumption. We

can now read off the non-reciprocal coefficients as follows

$$
\begin{array}{rcl}
\alpha_0 &=& \frac{\kappa_2 \Gamma_1 r_1 - \Gamma_2 \kappa_1 r_2}{2 r_1 r_2 \sqrt{\bar{\kappa}}}, \\
\alpha_1 &=& \kappa_1^2 \frac{\Gamma_2 \kappa_1 r_2 - 3\Gamma_1 \kappa_2 r_1}{16 r_1^3 r_2 \bar{\kappa}^{5/2}}, \\
\alpha_2 &=& \kappa_2^2 \frac{\Gamma_1 \kappa_2 r_1 - 3\Gamma_2 \kappa_1 r_2}{16 r_2^3 r_1 \bar{\kappa}^{5/2}}, \\
\alpha_3 &=& 3\kappa_1 \kappa_2 \frac{\Gamma_1 \kappa_2 r_1 - \Gamma_2 \kappa_1 r_2}{16 r_1^2 r_2^2 \bar{\kappa}^{5/2}}, \\
\alpha_4 &=& \kappa_1 \frac{\Gamma_2 \kappa_1 r_2 - 2\Gamma_1 \kappa_2 r_1}{8 r_1^2 r_2 \bar{\kappa}^{3/2}}, \\
\alpha_5 &=& \kappa_2 \frac{\Gamma_1 \kappa_2 r_1 - 2\Gamma_2 \kappa_1 r_2}{8 r_2^2 r_1 \bar{\kappa}^{3/2}}.
\end{array}
\tag{37}
$$

We conclude by noting that other systems, such as mass-conserved reaction-diffusion systems and elastic networks are amenable to a similar theoretical treatment. It is very natural for nonlinear terms to appear in continuum theories that describe complex systems, and the above results exemplify the diversity of routes to nonlinear NRCH models.

## Characterising effervescence

The fields $\phi_{1,2}$ are evaluated at discrete $N$ points in one-dimensional space to obtain a $2N$ dimensional vector

$$
\boldsymbol{\phi}(t) = \left(\phi_1(h,t), \phi_1(2h,t), \ldots, \phi_1(Nh,t), \phi_2(h,t), \phi_2(2h,t), \ldots, \phi_2(Nh,t)\right)^T.
\tag{38}
$$

The Maximal Lyapunov index $\Omega$, which measures the sensitive dependence on initial conditions, is defined as follows[76]

$$
\Omega = \lim_{|\delta\boldsymbol{\phi}(0)| \to 0, T \to \infty} \frac{1}{T} \log \frac{|\delta\boldsymbol{\phi}(T)|}{|\delta\boldsymbol{\phi}(0)|},
\tag{39}
$$

where $\delta\boldsymbol{\phi}(t)$ is the difference between the two trajectories at time $t$ as a result of the difference in initial conditions $\delta\boldsymbol{\phi}(0)$. A positive value of $\Omega$ is indicative of chaos. We have calculated $\Omega$ as a function of system size (the results presented in Fig. 5c. d in the main text are free of system size effects) and as a function of $\alpha_{1,0}$ with $\alpha_{0,1} = 2.5$. A full characterisation of spatiotemporal chaos would involve a calculation of the Lyapunov spectrum which we defer to the future.

## Persistence and lifespan of droplets

We define a field $\bar{\alpha}(\boldsymbol{r}, t)$ that assumes values zero or unity where $\alpha$ is lower or higher than a threshold value $\alpha_c$ as follows

$$
\bar{\alpha}(\boldsymbol{r}, t) = \frac{1}{2} + \Theta(\alpha(\boldsymbol{r}, t) - \alpha_c).
\tag{40}
$$

The integral of $\bar{\alpha}$ over the simulation area returns the area occupied by droplets at time $t$. As the droplet at $\boldsymbol{r}$ disappears, the value of $\bar{\alpha}$ vanishes. The cumulative product $\nu(\boldsymbol{r}, n) = \Pi_{n=0}^{N} \bar{\alpha}(\boldsymbol{r}, n\Delta t)$ is unity only at those points where the droplet that existed at $t = 0$ persists at least until time $t$. Integrating this quantity over space we find the fraction of space occupied droplets that have survived until time $t$, thus defining the surviving fraction

$$
\bar{\nu}(t = N\Delta t) = \frac{1}{A} \int d^2 r \, \nu(\boldsymbol{r}, N\Delta t),
\tag{41}
$$

where $A$ is the area of the simulation box. $\nu(t)$ depends on initial conditions and is a fluctuating quantity in general. Averaging over initial conditions in two dimensions for $\alpha_0 = 5$, $\alpha_1 = 4$, we find that $\bar{\nu}(t)$ decays exponentially meaning that we can associate an average lifespan with a droplet.

A direct calculation of droplet size versus lifetime similar to the method adapted in one dimension is very challenging to implement in

two dimensions. We choose a different method using the quantity $\nu(\boldsymbol{r}, t)$ that measures the persistence of droplets. Choosing $\boldsymbol{r}$ to be the centre of an existing droplet of a size $\mathcal{A}_{\text{droplet}}$, we track $\nu(\boldsymbol{r}, t)$ as a function of $t$, and record the point $\tau$ at which it drops to zero as the lifetime associated with the droplet. $\tau$ is plotted as a function of droplet size in the inset of Fig. 2d. See SI for a discussion on the dynamics of reciprocal granules and droplets in one dimension.

## Linear stability analysis

Here, we will obtain Eqs. (12) and (15) in the main text and compare the instability due to the interplay of $(\alpha_0, \alpha_1)$ with some other known instabilities[42]. To do so, we modify Eq. (5) by replacing the surface tension $K$ with $K(1 + i\beta)$, thus introducing a non-reciprocal coupling that appears at a higher order in the gradient. The dispersion relation Eq. (10) is modified due to the non-reciprocity $\beta$ as

$$
\omega(q) = \Gamma q^2 [\alpha_0 - \alpha_1(1 - q^2) + \beta q^2],
\tag{42}
$$

while $\rho_q$ is unchanged. To investigate the stability of the plane wave solution Eq. (8) to the nonlinear NRCH equation Eq. (5), we insert the form Eq. (11) into Eq. (5), and expand the equation up to the first order in $\delta\rho_q$ in Fourier space, taking into account the definitions given in Eq. (42). We obtain

$$
\begin{bmatrix}
\delta\dot{\rho}_q(\boldsymbol{k}) \\
\delta\dot{\rho}_q^*(\boldsymbol{k})
\end{bmatrix}
=
\begin{bmatrix}
\mathcal{M}_{11} & \mathcal{M}_{12} \\
\mathcal{M}_{21} & \mathcal{M}_{22}
\end{bmatrix}
\begin{bmatrix}
\delta\rho_q(\boldsymbol{k}) \\
\delta\rho_q^*(\boldsymbol{k})
\end{bmatrix},
\tag{43}
$$

where

$$
\begin{aligned}
\mathcal{M}_{11}(\boldsymbol{q}, \boldsymbol{k}) &= -i\Gamma q^2 \left[-\alpha_0 + \alpha_1\left(1 - \frac{q^2}{q_0^2}\right)\right] - \Gamma K(1 - i\beta)(\boldsymbol{k} - \boldsymbol{q})^4 \\
&\quad + \Gamma\left[(1 - i\alpha_0) - 2(1 - i\alpha_1)\left(1 - \frac{q^2}{q_0^2}\right)\right](\boldsymbol{k} - \boldsymbol{q})^2, \\
\mathcal{M}_{12}(\boldsymbol{q}, \boldsymbol{k}) &= -\Gamma(1 - i\alpha_1)\left(1 - \frac{q^2}{q_0^2}\right)(\boldsymbol{k} - \boldsymbol{q})^2, \\
\mathcal{M}_{21}(\boldsymbol{q}, \boldsymbol{k}) &= -\Gamma(1 + i\alpha_1)\left(1 - \frac{q^2}{q_0^2}\right)(\boldsymbol{k} + \boldsymbol{q})^2, \\
\mathcal{M}_{22}(\boldsymbol{q}, \boldsymbol{k}) &= i\Gamma q^2\left[-\alpha_0 + \alpha_1\left(1 - \frac{q^2}{q_0^2}\right)\right] - \Gamma K(1 + i\beta)(\boldsymbol{k} + \boldsymbol{q})^4 \\
&\quad + \Gamma\left[(1 - i\alpha_0) - 2(1 - i\alpha_1)\left(1 - \frac{q^2}{q_0^2}\right)\right](\boldsymbol{k} + \boldsymbol{q})^2.
\end{aligned}
$$

The stability of the plane wave solution Eq. (8) is determined by the eigenvalues of the $\mathcal{M}$, which are given as $\lambda_\pm \equiv \text{tr}\,\mathcal{M}/2 \mp \frac{1}{2}\sqrt{\text{tr}\,\mathcal{M}^2 - 4\det\mathcal{M}}$. To the zeroth order in $k$, the eigenvalues are $\lambda_- = -2q^2(1 - q^2/q_0^2)$ and $\lambda_+ = 0$. For small wavelength perturbations $k \ll q$, the branch of eigenvalues $\lambda_-$ remains negative, and thus stabilising, and the stability of the travelling waves is determined by $\lambda_+$ alone. We calculate $\lambda_+$ up to quadratic order in $k^2$ to probe the advection and diffusion effects. To $O(k^2)$, and setting $\beta = 0$, we obtain the results presented in Eqs. (12) and (15).

$$
\begin{aligned}
V &= 2\Gamma q(\alpha_0 - \alpha_1 - 2\beta q^2 + 2\alpha_1 q^2), \\
D_L &= \Gamma \frac{q^2(q_0^2 - 3q^2)}{q_0^2(q_0^2 - q^2)} \\
&\quad -\Gamma\alpha_1(\alpha_0 - \alpha_1) + \Gamma\frac{\alpha_1^2 q^2(3q^2 - 5q_0^2)}{q_0^2 - q^2} \\
&\quad + 6\Gamma\beta\alpha_1 q^2/q_0^2.
\end{aligned}
\tag{44}
$$

$D_L$ can change sign as a function of the non-reciprocal parameters and wavenumber. The three contributions to the modifies Eckhaus instability are arranged in three separated lines in Eq. (44). (i) The first is a conserved version of the original Eckhaus instability, which holds at equilibrium. The instability threshold of $q > q_0/\sqrt{3}$ above which the plane waves are destroyed remains unchanged while the diffusion coefficient itself is multiplied by $q^2$ due to number conservation. (ii) The second route to an instability has no counterpart

in literature and is responsible for effervescence. (iii) The third is a conserved Benjamin-Feir-Newell instability which received contribution from the third line but also from the second and third and was explored in ref. 42. We will close the section with a comparison with the complex Landau-Ginzburg equation

$$\partial_t \phi = \Gamma \left[ (1 + i\alpha_0)\phi - (1 + i\alpha_1)|\phi|^2\phi + K\nabla^2\phi \right]. \tag{45}$$

A stability analysis similar to the one detailed in this section around travelling waves with dispersion $\omega(\boldsymbol{q}) = \Gamma(\alpha_0 - \alpha_1 + \alpha_1 q^2)$. To $O(k^2)$ we have

$$
\begin{aligned}
V' &= 2\Gamma q(\alpha_1 - \beta), \\
D'_L &= \Gamma\alpha_1\beta + \Gamma\frac{q_0^2 - 3q^2}{q_0^2 - q^2}.
\end{aligned}
\tag{46}
$$

Comparing Eqs. (44) and (46), we find the differences between the two classes of dynamics. The complex Landau-Ginzburg equations are written without the parameter $\alpha_0$ since $\alpha_0$ can be removed through the transformation $\phi(\boldsymbol{r}, t) \to \exp(i\alpha_0 t)\phi(\boldsymbol{r}, t)$. Note that $\alpha_0$ does not appear in $D'_L$ because of this symmetry. However, number conservation breaks this symmetry and elevates $\alpha_0$ to a parameter that not only produces a rich variety of patterns but also together with $\alpha_1$ participates in producing spatiotemporal chaos.

Finally, we note that the other terms in the expression for $\alpha$ in Eq. (25) cannot be treated on a similar footing to $\alpha_{0,1}$. This is because they explicitly break the phase invariance of the dynamics, i.e., terms that are multiplied by $\exp(i\boldsymbol{q} \cdot \boldsymbol{x} - \omega t)$. These terms lead to instabilities at finite wavenumber meaning that our analysis here is complete for $q, \omega \to 0$ limit.

## Data availability
The data supporting the main findings of this study are available in the paper and its Supplementary Information. Any additional data can be made available upon request.

## Code availability
The algorithms for the codes supporting the main findings of this study are available in the paper and its Supplementary Information. Any additional information concerning the code can be made available upon request.

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

## Acknowledgements

We acknowledge fruitful discussions with Jaime Agudo-Canalejo, Philip Bittihn, and Giulia Pisegna. This work has received support from the Max Planck School Matter to Life and the MaxSynBio Consortium, which are jointly funded by the Federal Ministry of Education and Research (BMBF) of Germany, and the Max Planck Society.

## Author contributions

S.S. and R.G. designed the research, conducted the research, analysed the data, and wrote the paper.

## Funding

## Competing interests

The authors have no competing interests as defined by Nature Portfolio, or other interests that might be perceived to influence the results and/or discussion reported in this paper.

## Additional information

**Peer review information** : *Nature Communications* thanks the anonymous reviewers for their contribution to the peer review of this work. A peer review file is available.

