## [Transparent Peer Review file · Nature Communications]

Effervescence in a binary mixture with nonlinear non-reciprocal interactions

Corresponding Author: Dr Suropriya Saha

Version 0:

Reviewer comments:

Reviewer #1

(Remarks to the Author)

The manuscript presents simulation results and some analytical stability analysis of a new variant of non-reciprocal Cahn-Hilliard (NRCH) systems. In particular, they present a phenomenon they term "effervescence", a form of spatiotemporal chaos. Non-reciprocal coupling has received significant attention in the soft- and condensed-matter community in the last few years. Therefore, the manuscript is timely and will be of interest for specialists in this field. However, the authors unfortunately do not formulate a clear question or open problem that they address in their work. The specific variant of NRCH system they study, which includes nonlinearity in the non-reciprocal coupling, is only motivated in a very abstract way. Therefore, it is hard to put the findings into the context of concrete physical systems and to see how the predictions from theory could be tested in future experiments. In its current form the manuscript is too technical and I cannot recommend it for publication in Nature Communications. The authors should either consider submitting to a more specialized, technical journal or formulate a clear question and motivate the equations they study from one or more concrete physical systems.

One such example might be mass-conserved reaction diffusion systems. The generic appearance of nonlinearity in the non-reciprocal couplings has been shown in Ref. [1]. A concrete physical system would be the in vitro reconstitution of the Min system of *E. coli*. It has two conserved fields, the densities of MinD and MinE, and in fact exhibits both traveling waves and spatiotemporal chaos of a sort that looks similar to "effervescence" as described in the present manuscript (see e.g. [2, 3]).

Minor issues:

- p. 3: "The peak of $P(|\varphi|)$ occurs precisely at ρ_0 [see Fig. 2a] ..." I can't find ρ_0 anywhere in Fig. 2, also Fig. 2b is the one showing $P(|\varphi|)$, so I'm confused as to what I'm supposed to look at.

- I find it a bit strange and confusing that the authors discuss results from numerical simulations before introducing the dynamical equations. This interrupts the flow of reading because one has to jump ahead to understand the meaning of the parameters that the authors refer to. I recommend that the authors first introduce the governing equations.

- Neither the term "reciprocal granules" nor "effervescence" are particularly well defined or motivated. What are the defining features of these phenomena? How is effervescence different from other types of spatiotemporal chaos? What justifies introducing new terminology?

- It is not quite clear what happens in the stability analysis of "reciprocal granules" on p. 7 and 8. These granules are spatial structures, but the authors appear to perform stability analysis around a homogeneous steady state. This is in contrast to the stability analysis of traveling waves in the previous section, which is performed around the traveling wave profile. In the granule case, it appears that the authors treat the conserved fields as effective control parameters of the local dynamics, and perform a "local stability analysis". A similar approach has been employed in previous works such as [4, 5] to understand the emergence of spatiotemporal chaos in the Min system. The authors should motivate their approach a bit more carefully and refer to previous literature to provide context.

- What does the blue shading in Fig. 5c indicate?

- It should be emphasized that the linear stability diagrams in Fig. 6 show long wavelength instability (i.e. the limit $q \rightarrow 0$). Does the system only exhibit this type of instability? Does the Hopf bifurcation always happen at $q = 0$, or are there cases where the exceptional point is at finite q ?

- I found Fig. 7 quite hard to read. Axis labels in panels a, b, e, and f are missing. Since the “phase” of the composition doesn't play a role for the linear stability and only has a minor effect on the observed patterns, it might be simpler to plot the stability diagrams together with the quantifications $\delta\phi_1$ and σ on one parameter axis (say ϕ_1) and show the two-dimensional heatmaps only as an extended data figure. Alternatively it might help to combine Figs. 6 and 7. An illustration of what the quantity σ measures might be useful.

[1] Frohoff-Hülsmann, T. & Thiele, U. Nonreciprocal Cahn-Hilliard equations emerging as one of eight universal amplitude equations. Preprint at <https://arxiv.org/abs/2301.05568> (2023).

[2] Würthner, L. et al. Bridging scales in a multiscale pattern-forming system. *Proc. Natl. Acad. Sci. U.S.A.* 119, e2206888119 (2022).

[3] Brauns, F. et al. Bulk-surface coupling identifies the mechanistic connection between Min-protein patterns in vivo and in vitro. *Nat Commun* 12, 3312 (2021).

[4] Halatek, J. & Frey, E. Rethinking pattern formation in reaction–diffusion systems. *Nature Physics* 14, 507 (2018).

[5] Denk, J. & Frey, E. Pattern-induced local symmetry breaking in active-matter systems. *Proc Natl Acad Sci USA* 117, 31623–31630 (2020).

Reviewer #2

(Remarks to the Author)

In their manuscript Saha and Golestanian introduce a novel class of non-linear pattern formation. Building on a recent series of works, they show that antisymmetric couplings between spatially coupled fields can result in the fluctuation-induced restoration of PT symmetry. This rather abstract property has a beautiful real space manifestation dubbed effervescence by the authors.

The manuscript reports a comprehensive theoretical and numerical study of a minimal model of two conserved species prone to phase separation. The number, and density, of results and comments presented in this manuscript is remarkable. The addition of non-reciprocal coupling breaking the PT symmetry typical of passive systems is thoroughly discussed in generic terms independent of the specifics of the minimal model used to conduct the quantitative analysis. The characterisation of genuine spatiotemporal chaos is also very convincing and I am not aware of any earlier results demonstrating this type of physics (in this context).

For these reasons, I do believe that this work could ultimately warrant publication in a journal of the caliber of *Nature Communications*. However, I feel that, in the present form, the manuscript is very hard to read for non-specialists of the blooming field of non-reciprocal matter. I feel that a significant effort in the presentation of the results would be required to make this manuscript accessible to the broad readership of *Nature Comm*.

1— Reporting and discussing the essential results prior to the introduction of the actual model is in general a good idea. It allows the authors to convey their main message in simple words. However, I am afraid that this attempt does not get its goal. The first discussion makes it very hard to understand what the authors are actually talking about. I had to skip the intro, understand the model and go back to the intro again to better understand it (even though I was partly familiar with this line of research).

When the authors write:

The granular structure for the domains that restore reciprocal interactions, is evident in the droplet size distribution P (Adroplet) shown in Fig. 2a-b where we observe a prominent peak for a fundamental reciprocal granule and an oscillatory pattern in the probability for the production of larger droplets [Fig. 2a].

I must say that I could not understand at all what the concept of a reciprocal granule was at this stage. Why reciprocal? In what sense? How can the PT symmetry of an effective coupling translate to a drop sized distribution?

2— I also feel it would be nice to give a simple pictorial idea of what type of simple dynamics the theory actually models when introducing Eq. 1 2. Figure 1 is very dense and refers to results established later on in the manuscript. This makes the presentation even more confusing. Referring explicitly to one, or a selection of, experiment where this type of modelling could actually apply would also help understanding the model, and give a better sense of the significance of the work.

4— As far as I understand it, the physics discussed in this article strongly rely on the $U(1)$ symmetry (breaking) of the complex density field. I therefore wonder how these results could be robust, or extended, to systems involving more than two chemical species. Could the authors comment on this point?

I understand that the authors might find these comments rather superficially and might even disagree with them as they are

subjective. But I believe that I might not be the only reader having this perception.

Minor comments along the same line

3—The long list of references in the intro rightfully credits a number of earlier results, it is however so dense and diverse that it makes it difficult to understand the actual line of research the authors aim at improving.

(The reference to classical non-Hermitian physics is also rather biased by the authors' own work. This work was definitely not foundational of the field. A number of experiments have already been reported and the synthetic review article published in Nature physics on the topic. It could provide the reader a more accurate view of the status of this field)

4—The authors write "This emergent imperfect PT symmetry breaking with local restoration of reciprocal interactions produces two new states, namely an effervescent wave which is a hybrid state with droplets and a travelling pattern, shown in Fig. 1a-b, and effervescence without the travelling pattern, shown in Fig. 1c-d. »

Fig. 1d gives a very clear picture whereas I found the sketch of Fig. 1b rather cryptic. A side by side comparison of the two states using the same representation would definitely help understanding their similarities and differences

I could write a long list of suggestions and comments along the same line, but I feel it would be more effective to first give the authors the chance to first rewrite their manuscript to make it more accessible. I can understand that this type of report could be very frustrating given the amount of work and results reported in the manuscript. But in the present form I am afraid this manuscript could appeal to a very narrow audience of specialists.

Reviewer #3

(Remarks to the Author)

In their manuscript "Effervescent waves in a binary mixture with non-reciprocal couplings", Saha et al. investigate a binary mixture of phase separating fields with non-reciprocal couplings. They introduce nonlinear inter-species couplings to the system, and demonstrate that such cross coupling can give rise to spatiotemporal chaos, which they termed "effervescence". They then characterize the various states observed in such system, and rationalize their transitions by using linear stability analysis.

Nonreciprocal active systems have recently attracted much attention due to the rich dynamics invoked by nonreciprocity. Particularly, nonreciprocity is known to be able to induce a PT symmetry breaking and lead to traveling states. In this manuscript, the authors generalize the nonreciprocal couplings, which are usually linear in previous studies, to a nonlinear form, and find various new states at such settings. The simulations and analytics are well executed, and the results are interesting and scientifically solid. However, I have a few concerns that the authors should clarify before I can recommend for publication.

1. Whereas nonreciprocal interactions are not uncommon in open systems, I am concerned whether the specific nonlinear nonreciprocal couplings, which change signs when varying concentrations, exist in real systems. If so, the authors should list those systems. If not, then I am afraid the significance of this manuscript might be reduced considerably since changing sign of nonreciprocal coupling is the basis of this manuscript.

2. The "effervescence" is supposed to be the major selling point of this paper. But I fail to spot the unique/nontrivial properties of such state. Is it merely a spatiotemporal chaos manifested in nonreciprocal systems? If not, how is different from the spatiotemporal chaos exhibited in other nonlinear systems, e.g. the Kuramoto-Sivashinsky system? To what extent is it different from the micro-phase separation found in ABP (e.g. PRL 125/178004/2020, PRX 8/031080/2018), which also contains fluctuating droplets? Are there some properties that are specific to the "effervescence" here, yet unidentified in previous studies? Highlighting the unique properties of the effervescent states, I believe, can make the manuscript more compelling.

3. Since "effervescence" is the main point here, it will be useful to provide some results on the characteristic length and time scales of such state, and argue the parameters that control these scales.

4. If I understand correctly, it is claimed in the manuscript that the interplay between linear and nonlinear non-reciprocal interactions is crucial to the emergence of spatiotemporal chaos. Yet, spatiotemporal chaos has been found in systems with

merely linear nonreciprocal coupling, see e.g. arxiv:2306.08868. Can the authors comment on this? And what are the differences between the chaos exhibited in the two systems?

5. In the last paragraph on page 7, the authors wrote:

"However, a globally constant θ as a solution of Eq. (20) is ruled out by number conservation as argued in the previous section (also see Fig. 5b), suggesting that for $\alpha_0 > \alpha_1$ finite-sized stable droplets can be created with constant $|\phi|$ and a field θ inside that can be determined using if the boundary conditions at the edge of the droplet is known."

Can the authors elaborate on the emergence of "finite-sized" and "stable" droplets out of this instability? In other words, how can tell that this instability (Eq. 20) leads to "finite-sized" and "stable" droplets? What determines the size of the droplets?

6. In Fig. 3e, it seems that the red line in the right panel can be extended all the way to $\alpha_1=2$, which means that the Effervescence state also contains propagating modes, right? Also, I see another branch with higher frequencies. What does this branch represent?

7. In Fig. 2a, how are droplets identified from the field values?

8. The authors claimed that:

"Effervescence gives rise to spatiotemporal chaos, and the emergence of an effective noise from the deterministic nonlinear dynamics, due to nonlinear non-reciprocal interactions."

Can the authors explain in an intuitive picture how effervescence can give rise to spatiotemporal chaos?

Minor points:

1. "The stability of the waves can is" -> "The stability of the waves can be".

2. "In the effervescence state travelling waves over a large large range ..." -> "In the effervescence state travelling waves over a large range ...".

Reviewer #4

(Remarks to the Author)

In the manuscript "Effervescent waves in a binary mixture with non-reciprocal couplings" by Suropriya Saha and Ramin Golestanian, the

authors explore conserved field models, describing e.g. mixtures of two particle species, where nonreciprocal interactions appear at linear and nonlinear order, allowing for spatially strongly heterogeneous interactions. The paper reports the formation of two spontaneously formed dynamical steady states, called effervescence and effervescent traveling waves, respectively, the first characterized by spatiotemporal chaos, the second by a combination of spatiotemporal chaos and regular traveling waves.

The authors analyze the model by numerical and analytical methods and are able to predict the rough characteristic of the state diagram using a linear stability analysis. They describe the emergence of the exotic phases as a complex interplay between the formation of traveling waves and spontaneously forming and dissolving droplets of fluctuating shape. For a special case of composition of the two components of the mixture, they deduce the exact dispersion relation of the traveling waves and establish their stability. The authors also identify the droplets as local areas of restoration of reciprocity, which are marginally stable.

The paper gives a nice account of the rich and complex behaviour that can arise in minimal models of active matter systems with non-reciprocal interactions. In particular, a new type of chemical spatiotemporal chaos is observed. It is argued to arise from spatial heterogeneities in the non-reciprocal couplings that change sign as a function of the amplitude of the scalar fields or "species".

It is also pointed out that this phenomenology can be understood as an instance of imperfect breaking of PT symmetry, involving fluctuating domains in space where the symmetry is temporarily restored. All these results make the paper a rich and valuable contribution to its specific research field and of likely interest to a specialised audience of experts, who can cope with the paper's rather technical style. Yet, the paper does not penetrate more deeply into the physical mechanisms behind these observations and, for example, describe or explain the process of droplet annihilation in detail. It also does not attempt to offer an explanation for the presence of spatiotemporal chaos in the background of the droplets. The analysis in the case of arbitrary compositions does not go beyond linear stability of homogenous configurations, but the authors claim, on this basis, the existence of particular forms of inhomogeneous stationary solutions. This does not seem to be justified by further qualified arguments. Also with respect to other aspects of the phenomenology, one might have hoped for a more thorough analysis and clearer presentation, as detailed in the comments below. A more severe issue might however be the question who is actually the targeted audience of the paper. The authors clearly attempt to put their work into a broader context and to describe it verbally, in particular in the first part of the paper. Yet, this description remains somewhat vague, while, at the same time, abstract and technical. I'm afraid that most non-experts will almost certainly not be able to understand either the verbal characterisation or the following technical discussion, nor will they be able to appreciate the possible interdisciplinary pertinence and applications of the work (if any), while the small group of initiated experts who do would most probably have preferred a more extensive classical physics-article style of the exposition of the topic.

In summary, I think it is fair to say that the authors have discovered a rich and interesting phenomenology but have not fully

explained or analysed it, nor have they provided convincing arguments for why it should be of interest to an interdisciplinary readership or catered their presentation to it. Altogether, I therefore conclude that the paper is, in its present form, not suitable for the chosen journal and its broad interdisciplinary readership, but would be better redirected to and rephrased for classical physics journals, such as the Physical Review or similar.

More specific comments:

1. In the introduction and throughout the paper, the authors use the terms non-reciprocal interactions, activity, and PT symmetry breaking synonymously and should motivate and explain this. They could also mention Refs. [T. Frohoff-Hülsmann et al., Physical review. E 103 4-1 (2020)], [T. Frohoff-Hülsmann et al., arXiv:2301.05568 (2023).] which have investigated the phenomenology of field equations with nonreciprocal dynamics and especially the nonreciprocal Cahn-Hilliard model in great depth.
2. It seems to me that neither Ref. [15] nor Ref. [16] report purely temporal oscillations, as stated in the manuscript.
3. The statement: "The transition to traveling patterns occurs upon running the parameters of the model such that an exceptional point is crossed" is incorrect because all cited references [15,18,21] also mention the possibility of a transition to a traveling pattern state via an oscillatory instability (within the terminology of the authors called Hopf bifurcation).
4. The definition of exceptional point transitions seems slightly incorrect, because it neither mentions the important scenario of an EP transition from a static state of broken continuous symmetry (not fully mixed) to a dynamical state, nor is an eigenvalue with nonzero imaginary part a general feature of an exceptional point.
5. On page 2 the authors state that "An emergent feature of the non-reciprocal interactions implemented at the linear level in a mixture of two species is the spontaneous breaking of space-translation, time-translation, and polar symmetries". It should be made clear that only spontaneous breaking of time-translation and polar symmetries are exclusive features of nonreciprocal mixtures.
6. In the verbal discussion, the variables and parameters are not generally understandable without first reading the more mathematical exposition further below. For example, "alpha" should be better defined together with the model equations. Also, the description in the first paragraph of page 3 is not very clear. Should not the explicit form of alpha, given in Eq. (4), be introduced here? What is the motivation for this particular choice? In general, it would seem advisable to adopt more of an extensive and explicit classical physics-article style, e.g., by introducing the model earlier in the text and discussing the meaning of the parameters afterwards less cryptically.
7. A clear distinction between the terms droplet and granule would be helpful, e.g, Fig. 1b suggests that a droplet is formed by the pairing of two reciprocal granules, while on page 3 it is stated that a droplet "is created by the fusion of two or more reciprocal granules".
8. The authors state on page 3 that "domains grow and shrink by losing or gaining matter". Could the precise mechanism be mentioned?
9. Similarly, it might help to provide clear definitions of the term droplet and A_{droplet} , used to obtain the numerical data presented in Fig. 2?
10. In Fig. 2a, the peak of $P(|\phi|)$ at ρ_0 is not apparent. Maybe the authors meant to refer to Fig. 2b? If so, also here the density ρ_0 should be indicated.
11. In Eqs. (1) and (2), the functional-derivative notation (as in the inlined formula for μ_i above these equations) would seem more appropriate.
12. The authors should distinguish the terms spatiotemporal chaos and effective noise. How would the effective noise relate to an additive Gaussian white noise, as typically used in stochastic versions of field models of the type analysed in their work?
13. Wouldn't $\Sigma_1 \neq \Sigma_2$ already for the "reciprocal" case $\alpha=0$ mean a violation of the fluctuation dissipation relation?
14. On page 4, it is stated that the form of the free energy in Eq. (3) leads to microphase separation, while the form chosen in Eq. (6) leads to bulk phase separation. Could the authors please resolve this apparent discrepancy?
15. In Fig. 3e it is not visible by eye that the peak in the effervescent wave phase actually fully disappears at some specific value of α_1 , as stated. The placement of the right black line therefore appears rather arbitrary.
16. It is not clear for which composition the numerical state diagram in Fig. 3 was computed.
17. The term bifurcation seems to be used in a somewhat generalised/overloaded (and maybe not acceptable) way. Isn't a bifurcation point always related to the loss of stability of a fixed point? The authors also use this term for the splitting of the real parts of the eigenvalues of the stability matrix, although none of the real parts changes its sign.
18. From Fig. a,b there seems no strong connection between the exceptional point indicated by the red line and the type of steady state the system adopts. Hence, the linear stability analysis does not seem to be really informative.
19. What's the meaning of the grey circles in Fig. 6?
20. The authors should define the meaning of \leftrightarrow
21. On page 7, the authors claim that "for $\alpha_0 \gg \alpha_1$, we should find stable droplets submerged in a traveling wave". The basis for this is not clear.

Version 1:

Reviewer comments:

Reviewer #1

(Remarks to the Author)

The authors have significantly improved the structure of the manuscript. It is now written in a style that is more accessible to a broad audience. However, much of the content is still quite technical. Nonetheless, I think the manuscript might be suitable

for Nat Comms, given the significant recent interest in mass-conserved, non-reciprocally coupled systems. Before I can recommend the manuscript for publication, the authors should address the following remaining concerns.

1) It is not clear what microscopic dynamics would lead to a changing sign of the non-reciprocal coupling in a real physical system. This sign change is the crucial ingredient in the model that gives rise to the described phenomena of “effervescence”. At the end of the new section on “real physical systems” the authors appeal to the generic nonlinearity of non-reciprocal coupling that arises from coarse-graining microscopic models. However, a changing sign is a much stronger requirement, and I’m not aware of a case where this arises out of a microscopic model that might have an experimental realization. Overall, I did not find the section on potential experimental systems very convincing, since this key aspect is not addressed.

Related to this issue is that some of the phenomena described by the authors, such as spatio-temporal chaos and transversally traveling fluctuations (see Fig. 8a) do, in fact, not require any nonlinearity of the non-reciprocal coupling at all (see Ref. [56]). This should be mentioned in the manuscript and the authors should disambiguate more clearly between phenomena that require the sign-change in the non-reciprocal coupling and those that don’t.

2) I still find some of the terminology introduced by the authors unnecessary and not sufficiently well motivated and defined. There is still no precise explanation of what a “reciprocal granule” is and what makes it “reciprocal”. On page 3 they are described as droplets on “the smallest possible length scales”. What sets these length scales? Why the plural, are there multiple such scales? Further the authors write “Their small size and short lifespan distinguishes them from longer-lived larger droplets.” Can this be turned into a quantitative criterion? What makes the “granules” “reciprocal” isn’t really clear either. Are there “reciprocal droplets”, too?

In the section “Quantifying effervescence.” the authors state on the one hand that the non-reciprocal interactions change sign at $|\phi| = \rho_0$, implying that non-reciprocal coupling is small for $|\phi| \sim \rho_0$. They then write “Within spatial domains where $|\phi|$ is nearly zero, the chasing stops locally [...]”. Why this is the case remains unclear. It also seems at odds the previous statement, which suggests that the chasing dynamics stops near ρ_0 which is not close to zero. This point needs to be clarified.

Since the term “reciprocal granules” is still unclear, so is the definition of “effervescence” given in Fig. 2. For instance, I was wondering whether the authors would some of the chaotic phases of the Gray-Scott model “effervescent” (Pearson, Science 261(5118), 189–192, 1993, see <https://pmneila.github.io/jsxp/grayscott/> for an interactive in-browser simulation). The phases reminiscent of “effervescence” (called “Chaos” and “Chaos and Holes” in the interactive simulation) are classified as phase turbulence by Pearson. This suggests that effervescence might simply be a form of phase turbulence, which calls into question the need for a new term.

Minor issues

- Fig. 4 suggests that “effervescence” also occurs in 1D. This should be stated clearly. Could the authors mark granules vs droplets in the kymographs in Figs. 4a and 4b?
- On page 5 a criterion is described to distinguish TWs from spatio-temporal chaos based on the fluctuation amplitude being small ($|\phi| < 0.1$). At this point in the manuscript, it is not clear that TWs in the model generally have such small amplitude. This should be clarified.
- On page 8, in the context of local stability analysis, the authors write “a globally constant ρ as a solution of Eq. (18) is ruled out by number conservation as argued in the previous section (also see Fig. 5b)”. The use of the word “global” in the context of the local stability analysis is a bit confusing since the latter only regards a small subsystem of the whole. Indeed, on the local level, number conservation need not hold, since mass can be redistributed in space.
- Fig. 2a is not very clear. The elements shown should be labeled.
- In Fig. 2b, it might be useful to show the contour at ρ_0 where the non-reciprocal coupling actually changes sign. A stronger contrast color would also help.
- What do the different colored lines in Fig. 2f show?
- The labels in the inset of Fig. 2d are too small.
- I found the oscillatory features in the distribution of droplet areas Fig. 2d very striking, in particular since they do not appear to have a constant period, but instead the period increases with the droplet size. Given the chaotic nature of the dynamics, it is remarkable that such a feature is found after thresholding of a continuous field. Does it depend on the precise value of the threshold? Does this observation depend on domain size?
- Many figure references in the text are broken (number missing).
- Page 6: “In the next two sections we will demonstrate that the peak at $|\phi| < 1$ in Fig. 2f” should probably be 2e instead.

(Remarks on code availability)

Reviewer #2

(Remarks to the Author)

I am very impressed by the efforts the authors have devoted to improving the quality of their manuscript. They have convincingly addressed all my questions and comments. I am happy to recommend their revised manuscript for publication.

(Remarks on code availability)

Reviewer #3

(Remarks to the Author)

I appreciate the authors for the thorough response and clearance to my comments. The authors have added considerable new results to the manuscript. I am convinced that the results in the manuscript are scientifically solid, technically clear, and self-contained. However, I agree with the other referees that the manuscript, even after revision, is a bit too technical for Nat. Commun. Moreover, I am a bit concerned about the potential scientific impact of the manuscript in relevant fields. Specifically, most results presented in the manuscript are based on the assumption that nonreciprocity or cross-coupling is composition-dependent and can change sign. Although the ubiquity of nonreciprocal interaction is broadly accepted, the existence of the specific form of nonreciprocity (i.e. it can change sign) presented in this paper is questionable. If so, then this can significantly reduce the impact of the manuscript. If not, the authors should list systems that exhibit such form of nonreciprocity or, provide other evidences/information to strengthen the motivation of this work. It is only after a clear resolution to this issue, that I can recommend publication of the manuscript on Nat. Commun.

(Remarks on code availability)

Version 2:

Reviewer comments:

Reviewer #1

(Remarks to the Author)

The authors have significantly improved the reasoning behind the terminology they introduce and their explanation of the observed phenomena. I can now recommend the manuscript for publication.

I still would like to make a remark on the sign-changing nonreciprocity: In the added section "Ubiquity of effervescence", it is shown that the specific form of the nonlinearity is unimportant for effervescence to emerge, as long as it allows α to change sign locally in some way, thereby leading to regions where reciprocal coupling is restored. I appreciate the extra work the authors put in to make this point. However, I'm still skeptical that this sign change of the coupling is something that is generic or ubiquitous in experimental systems (or even models with explicit microscopic interactions). The issue with the author's appeal to Landau expansion is that such an expansion assumes small values of the fields. For the sign change to happen, the field has to become sufficiently large and therefore might leave the range of validity of the expansion. Imagine for instance a scenario where the "true" coupling decreases and asymptotes to zero, but never changes sign, e.g. $\alpha = \alpha_0/(1+|\phi|)$. In this case, an expansion for small ϕ would lead to a spurious sign change of α . I therefore think that arguing for "ubiquity" on the grounds of Landau expansions is somewhat problematic.

The authors might want to stress that the "ubiquity of effervescence" is still conditional on the sign-changing nature of the non-reciprocal coupling, even if it is independent of how exactly the sign change comes about.

The sentence "We have established that effervescent steady-states that we expect to be the defining feature of future experiments with non-reciprocally interacting scalar densities." in the last paragraph seems to be grammatically broken. In the light of my concerns above, I also recommend that the authors tone down their claim a bit to say that effervescent steady-states are "a common feature" rather than "the defining feature" of non-reciprocally interacting scalar densities.

Typos:

Page 11 under Eq. 25, $\alpha_0=0.1$ should probably be $\alpha_3 = 0.1$

Page 12: "Consider an example ... we find that α_0 vanishes ..." should probably be α without the subscript

Reviewer #3

(Remarks to the Author)

I believe the current focus of the review, based on the comments from referees 1&3, is whether or not it is meaningful to introduce the change of sign of interspecies interaction, which, apparently is important to the results presented in the manuscript.

I agree with the arguments raised by the authors that such feature can emerge from a simple Landau-like expansion since there's no obvious reason to forbid this. However, there's also no reason why this should appear in any system, and that's why the referees thought it is necessary to clarify the microscopic dynamics that can lead to such feature or, alternatively, list one or two real systems that exhibit such behavior.

Furthermore, I strongly disagree with the statement that 'expecting this requirement as a condition for publication of our work is equivalent to not allowing Landau theory of superconductivity to be published until the BCS theory is developed for Cooper pair formation through electron-phonon interactions'. In fact, it is the difference between the Landau theory and the work presented here that raises my concern about the impact of this manuscript. The Landau theory proposes a GENERIC framework that captures the most important properties of phase transition, and is applicable to diverse types of systems and phenomena. The results presented in this manuscript, relies on the fact that the inter-species interaction must be able to change sign which, although not forbidden by the physical law, could be a very limiting case that only appears in very few systems. If not, the authors should clarify this.

So, to conclude, I still have mixed feelings about this manuscript. On the one hand, I am convinced that the results presented here is scientifically solid and interesting from the theoretical point of view. On the other hand, I am still concerned about its limitation of impact due to the lack of clarification on the changing sign.

Version 3:

Reviewer comments:

Reviewer #1

(Remarks to the Author)

In their responses, the authors have addressed my comments and those of the second referee. In the responses, they stress that a sign change in the nonreciprocal coupling is necessary for effervescence. However, this crucial aspect is easily lost in the long manuscript, where it is only mentioned relatively late and might easily be missed by the reader. I think the authors should clearly state this key requirement at prominent places in the manuscript, including in the abstract and the "Concluding remarks".

For example, I suggest that in the abstract they expand "... model in which the non-reciprocal interactions can depend on the local values of the scalar fields" by "in such a way that the nonreciprocity can change sign". The same caveat should be included in the sentence "We have established that effervescent steady-states will be a common feature of future experiments with non-reciprocally interacting scalar densities." in the "Concluding remarks" section.

In the section "Ubiquity of effervescence" the authors write "any nonlinearity is sufficient to produce effervescence." This contradicts their own statement that a sign change is required. There are many nonlinearities that do not change the sign, such as the term $\alpha = 1/(1 + |\phi|^2)$ I suggested in my previous report.

In the same section, below Eq. (25), the sentence "The ingredients required to arrive at these terms are activity in a suitable form, multi-species number-conserving mixtures, and general non-linearities" should be amended by adding "that allow α to change sign within the dynamically relevant range of densities".

Similarly, in the "Concluding remarks", the authors write "We show numerically that effervescence only requires non-specific nonlinear interactions." Again, I this needs to be amended to emphasize the requirement of a dynamic sign change due to the nonlinearity, which is a specific constraint on the nonlinear interactions that will produce effervescence.

In their response, the authors write "As a result our model has the same level of rigour as the passive model B as the fields never leave the regime of 'smallness'." I disagree with this statement, since the radius of convergence of the Taylor expansion of a given α might be smaller than the bounds on the fields due to the effective free energy. For example, if the "true" nonreciprocity is $\alpha = 1/(1 + 4|\phi|^2)$, then the Taylor expansion will fail for $|\phi| > 1/2$. In fact, it will produce a spurious sign change exactly at $|\phi| = 1/2$.

Note that denominators of the form $(1 + |\phi|^n)$ appear generically in models for (bio-)chemical reactions, e.g. in Hill functions and Michaelis-Menten kinetics. The above arguments are therefore not a technical detail but point to an important limitation of the polynomial expansion approach employed by the authors.

My understanding of the second referee's comments is that he is concerned exactly about the same thing when he writes "The results presented in this manuscript, relies on the fact that the inter-species interaction must be able to change sign which, although not forbidden by the physical law, could be a very limiting case that only appears in very few system."

In fact, if effervescence were so ubiquitous, shouldn't it already have been observed in nature or in experiments. Maybe dynamic sign-changes in the non-reciprocal coupling are indeed rare.

For the reasons listed above I think the authors need to generally tone down the claims about "ubiquity of effervescence" and emphasize the key requirement of sign changes in the nonreciprocity.

I appreciate that the authors added a section on "Microscopic justification of nonlinear NRCH" in the Methods. For the second example provided there, "Chemically active mixtures", it is not clear to me whether the example can actually produce a sign change. In fact, the polynomial expansion provided by the authors will fail when the $\phi_i < -\bar{\rho}_i$. The authors should explicitly show that the sign change occurs for the full expression for c_1 Eq. (35).

I recommend this manuscript for publication once the above issues have been addressed.

(Remarks on code availability)

Reviewer #3

(Remarks to the Author)

The authors have made significant improvements to the manuscript. Particularly, they have added considerable discussions on microscopic models that can give rise to the form of nonreciprocal interaction required by effervescence. I am happy to recommend the manuscript for publication.

(Remarks on code availability)

Response to Referees: Effervescent waves in a binary mixture with non-reciprocal couplings

Suropriya Saha^{1,*} and Ramin Golestanian^{1,2,†}

¹*Max Planck Institute for Dynamics and Self-Organization (MPIDS), D-37077 Göttingen, Germany*

²*Rudolf Peierls Centre for Theoretical Physics, University of Oxford, Oxford OX1 3PU, United Kingdom*

(Dated: April 29, 2024)

I. SUMMARY OF CHANGES IN THE MANUSCRIPT

We have thoroughly revised the manuscript taking into account the feedback from all four referees. We have added new results that establish the generality of the phenomenology presented in this manuscript. Here is a summary of the changes that we have made in the manuscript.

- We have changed the title of the paper to ‘Effervescence in a binary mixture with nonlinear non-reciprocal interactions’ which we think suits the work better.
- We have revised the text extensively. The revised parts are highlighted in blue. We have revised the text to present the theoretical model first before describing effervescence.
- We have redefined the parameters such that for positive α , species 1 chases species 2.
- Fig. 1 in the previous version of the manuscript has been split into Figs. 1 and 2. In Fig. 1, we now show snapshots from simulations to show the two main classes of dynamical behaviour.
- In Fig. 2, we now explain the terms ‘reciprocal granules’ and droplets. In Fig. 2, we have also added panels that show snapshots of a part of the field $|\phi|$ to highlight the droplet dynamics that occurs at different time scales. We have also added a figure that uncovers the timescale governing droplet dissolution and coalescence events.
- We have changed Fig. 7 to include results only for simulations in two dimensions and presented the phase diagram along a line to simplify the presentation.
- We have added two more sections to the paper after the discussion on the role of composition in tuning the steady-state dynamics. The first of these ‘ubiquity of effervescence’ demonstrates that the steady-states reported here should be observed generally for any form of nonlinearity. We have added a section called ‘Non-reciprocal interactions in real physical systems’ where we have listed systems where we expect to find effervescence.
- We have added two new sections - ‘Persistence and lifespan of droplets’, and ‘Nonlinear NRCH with non-zero ϕ_0 .’ to the methods section to supplement to discussion of effervescence in one dimension, and to touch on some technical aspects of dynamics at non-zero average composition respectively.

II. UBIQUITY OF EFFERVESCENCE

We have now added a section to the manuscript demonstrating that the results are unchanged if we change the form of the potential. We include here some results that further substantiate our claims. We aim to illustrate that the features of effervescence reported here are quite general using the standard Cahn-Hilliard free energy

$$f = -\frac{\phi_1^2}{2} + \frac{\phi_1^4}{4} - \frac{\phi_2^2}{2} + \frac{\phi_2^4}{4} + \chi\phi_1\phi_2, \quad (1)$$

where χ is the coefficient of reciprocal interaction and a general form for α

$$\alpha = \alpha_0 - (\alpha_1 + \alpha_2)\phi_1^2 - (\alpha_1 - \alpha_2)\phi_2^2 + \alpha_3\phi_1\phi_2 + \alpha_4\phi_1 + \alpha_5\phi_2. \quad (2)$$

* [suopriya.saha@ds.mpg.de](mailto:suropriya.saha@ds.mpg.de)

† ramin.golestanian@ds.mpg.de

FIG. 1. **a**, α is a fluctuating field whose probability distribution P evolves to a stationary distribution independent of its parametrization. Drawing parameters randomly from a Gaussian with variance 5 and mean zero, we find on an average a universal form for $P(\alpha)$ for steady-states that show effervescence. P has significant weight at negative values, and it is bimodal with peaks at positive and negative values. **b**, Effervescence is ubiquitous - it is the dominant steady-state when non-linear non-reciprocal interactions are allowed as seen in the statistics of observed steady states with varying reciprocal interaction coefficient χ .

We choose the values $\alpha_0 = 5$ and $\alpha_0 = 4$ for the two sets of simulations and draw the other coefficients from a Gaussian distribution with zero mean and variance 5. We run simulations in one and two dimensions for different realisations of α and random mixed initial conditions at vanishing average composition. We run 500 different simulations, first setting $\alpha_0 = 5$ in one dimension and vary χ between -1 and 1 . Next we run 100 different simulations, setting $\alpha_0 = 4$ in one dimension varying χ between -1 and 1 . We also run about 50 simulations in two dimensions. From these simulations, we then calculate $P(\alpha)$ and use the distribution to determine whether the system goes into a state with effervescence (with or without a travelling wave), phase-separation or a travelling wave. The results are summarised in Fig. 1. Panel (a) shows in faint blue markers, $P(\alpha)$ in two dimension for 50 realisations of α . The purple line shows the average $P(\alpha)$ for the effervescent steady-states. Panel (b) shows the fraction of dynamical steady-states as a function of χ for $\alpha_0 = 5$. At $\alpha_0 = 4$, the percentage of phase separation increases slightly. However, Fig. 1 shows that effervescence is truly ubiquitous, and is likely to be abundantly accessible in experiments, with visible signatures to look for.

We note that we have decided not to include these results in the main paper, as they overlap with the work that involving other authors in our group, and should be published after that particular paper is in the preprint server. We just thought we should reveal them to the referees as justification and in support of our statement on the robustness of our results.

III. RESPONSE TO REFEREE 1

Referee: The manuscript presents simulation results and some analytical stability analysis of a new variant of non-reciprocal Cahn-Hilliard (NRCH) systems. In particular, they present a phenomenon they term "effervescence", a form of spatiotemporal chaos. Non-reciprocal coupling has received significant attention in the soft- and condensed-matter community in the last few years. Therefore, the manuscript is timely and will be of interest for specialists in this field. However, the authors unfortunately do not formulate a clear question or open problem that they address in their work. The specific variant of NRCH system they study, which includes nonlinearity in the non-reciprocal coupling, is only motivated in a very abstract way. Therefore, it is hard to put the findings into the context of concrete physical systems and to see how the predictions from theory could be tested in future experiments. In its current form the manuscript is too technical and I cannot recommend it for publication in Nature Communications. The authors should either consider submitting to a more specialized, technical journal or formulate a clear question and motivate the equations they study from one or more concrete physical systems.

Response: We would like thank the referee for their positive evaluation of our work and very helpful comments and suggestions. We have revised the manuscript to take care of the two main points of criticism raised by the referee. We have posed the following motivation at the beginning of the paper: ‘In NRCH [1], non-reciprocal interaction of constant sign and strength leads to a fixed phase difference between the patterns of densities of the two species, i.e. one of the species ‘chases’ the other. The striking consequence of nonlinear non-reciprocity is that the sign and the magnitude of non-reciprocity is no longer fixed but is determined by the dynamics. We include non-specific nonlinear contributions in the coefficient of non-reciprocal interaction and allow the binary mixture to evolve to a non-equilibrium steady-state. We find that quite generically, i.e. independent of the details of the interaction, the system fractionates into dynamic spatial domains in the steady-state.’ Indeed, nonlinear couplings are rather the norm than the exception in this class of theories, as others have also now argued. We have added a section that summarises possible experimental realization scenarios, including Janus colloids and quorum sensing particles.

Referee: One such example might be mass-conserved reaction diffusion systems. The generic appearance of non-linearity in the non-reciprocal couplings has been shown in Ref. [2]. A concrete physical system would be the in vitro reconstitution of the Min system of *E. coli*. It has two conserved fields, the densities of MinD and MinE, and in fact exhibits both traveling waves and spatiotemporal chaos of a sort that looks similar to “effervescence” as described in the present manuscript (see e.g. [3], [4]).

Response: We thank the referee for suggesting the useful references and we have included them in the main text particularly where we review a list of systems including recent experimental findings for which NRCH with nonlinear coupling should serve as a sound framework.

We agree that the MinE-MinD system presented in [4] is indeed a good candidate where effervescence might be observed because it involves at least two conserved densities and non-linear interactions [5]. The authors describe the spatiotemporal chaos they find in [4] as defect mediated turbulence which is qualitatively different from effervescence. There are some interesting similarities with the images presented in our work and in Fig 1 of [4] however establishing meaningful connections would need further work.

Referee: p.3: “The peak of $P(|\phi|)$ occurs precisely at ρ_0 [see Fig. 2a] ...” I can’t find ρ_0 anywhere in Fig. 2, also Fig. 2b is the one showing $P(|\phi|)$, so I’m confused as to what I’m supposed to look at.

Response: In the current manuscript we have defined ρ_0 right after defining the non-reciprocal interaction coupling α . We have pointed to the peak in the figure with the value of ρ_0 in terms of $\alpha_{0,1}$.

Referee: I find it a bit strange and confusing that the authors discuss results from numerical simulations before introducing the dynamical equations. This interrupts the flow of reading because one has to jump ahead to understand the meaning of the parameters that the authors refer to. I recommend that the authors first introduce the governing equations.

Response: We have changed the structure of the manuscript as instructed by the referee, first introducing the dynamical equations. We then proceed to impose the additional symmetry of invariance under rotations in the composition plane and introduce the parameters $\alpha_{0,1}$ in α . We hope that these definitions will make it easy to follow the content of Fig. 1.

Referee: Neither the term reciprocal granules nor effervescence are particularly well defined or motivated. What

are the defining features of these phenomena?

Response: We have now addressed this concern and clearly defined both terms. To characterise the patterns in a comprehensive manner, it is best to separate the part of the phenomenon that is connected to phase separation into two types of processes, separating the generation of reciprocal granules which are spontaneously created and annihilated from larger-sized longer-lived droplets that are formed by coalescence of two or more reciprocal granules. Crucially, our observations are supported by numerical evidence - the peak in the distribution of areas of droplets shows a sharp peak at the smallest possible length scale due to the reciprocal granules and an oscillatory motif at higher areas revealing the coalescence events.

Upon changing composition, we find that the travelling waves give way to bulk phase separation with pulsating interfaces. In this phase completely stable droplets coexist with reciprocal granules (see Supplemental movie S4,S5). It is thus crucial to define droplets separately from reciprocal granules as we do find stable droplets as seen in the Extended Data figure.

Finally, with this evidence, effervescence is defined as the coexistence of reciprocal granules and droplets in a background of chaos.

Referee: How is effervescence different from other types of spatiotemporal chaos?

Response: In the revision, we have clearly defined the term effervescence, which corresponds to a state where chaos coexists with size-selected reciprocal granules and droplets. Chemical turbulence in complex Ginzburg-Landau has been characterised by defect turbulence consisting of spiral defects or phase turbulence, which are both observed at large length scales. Spiral-defect turbulence is well-documented in many forms of reaction-diffusion systems including the min protein system, the multicomponent version of which also includes conserved densities.

The only form of spatiotemporal chaos (that we are aware of) which has features somewhat similar to effervescence is the bubbly phase in Active Model B+. However, the bubbles in that model are macroscopic. That model describes active phase separation for a single species. In a single component scalar field theory activity can be introduced only via fourth order gradient terms leading to a negative Laplace pressure that produces droplets in the presence of noise and in two dimensions. However both these features, noise and negative interfacial tension are absent in our theory. We consider only a deterministic PDEs only in our paper and a recent paper [6] has shown that negative Laplace pressure does not arise in linear NRCH without cross couplings in the interfacial part of the free energy, .

Referee- What justifies introducing new terminology?

As we have argued in response to the previous question, the phenomenology we report is completely new, and the new terminology helps to dis-entangle it from superficially similar reports in the literature. In other words, we believe that we have discovered a new type of spatiotemporal chaos that bears signatures of phase separation, stemming from imperfect PT symmetry breaking. So, the name stresses on the novelty of our work. Moreover, the new terminology also helps with the presentation of our results.

Referee: It is not quite clear what happens in the stability analysis of “reciprocal granules” on p. 7 and 8. These granules are spatial structures, but the authors appear to perform stability analysis around a homogeneous steady state. This is in contrast to the stability analysis of traveling waves in the previous section, which is performed around the traveling wave profile. In the granule case, it appears that the authors treat the conserved fields as effective control parameters of the local dynamics, and perform a “local stability analysis”. A similar approach has been employed in previous works such as [4, 5] to understand the emergence of spatiotemporal chaos in the Min system. The authors should motivate their approach a bit more carefully and refer to previous literature to provide context.

Response: The referee is correct in thinking that we have carried out a local stability analysis about a locally homogeneous state. We thank the referee for the comment and we have added relevant citations and the following comment in the beginning of the section ‘Reciprocal granules and droplets’ – In contrast to the stability analysis in the last section, where we tested the robustness of the travelling waves that are global solutions of Eq. (5), here our reference state exists over a finite area, similar to the concept of ‘local’ stability analysis introduced in [7, 8].

Referee: What does the blue shading in Fig. 5c indicate?

Response: It indicates the wave-number beyond which the traveling waves are unstable. We have added a line to the figure caption clarifying this point.

Referee: It should be emphasized that the linear stability diagrams in Fig. 6 show long wavelength instability (i.e. the limit $q \rightarrow 0$). Does the system only exhibit this type of instability? Does the Hopf bifurcation always happen at $q = 0$, or are there cases where the exceptional point is at finite q ?

Response: We have now clarified and further emphasised this point in the section where we talk about the effect of tuning composition adding a statement after Eq. 20.

A Hopf Bifurcation can occur at higher order in gradients upon including terms that resemble a non-reciprocal surface tension as shown in [9, 10] or when the two species have different values of interfacial tension [11]. In our case addition of terms at higher order in q simply stabilises the system, i.e. we include activity only in the bulk interaction terms thus precluding the possibility of a short wavelength oscillatory instability [10].

Referee: I found Fig. 7 quite hard to read. Axis labels in panels a, b, e, and f are missing. Since the “phase” of the composition doesn’t play a role for the linear stability and only has a minor effect on the observed patterns, it might be simpler to plot the stability diagrams together with the quantifications $\delta\bar{\phi}_1$ and σ on one parameter axis (say ϕ_1) and show the two-dimensional heatmaps only as an extended data figure. Alternatively it might help to combine Figs. 6 and 7. An illustration of what the quantity σ measures might be useful.

Although the phase of the composition does not play a role in determining the stability or the dynamics, it plays a crucial conceptual role that we have now highlighted in the main text. Upon tuning composition away from $\phi_0 = 0$, Eq. (4) loses its resemblance to the conserved complex Landau-Ginzburg equations. Tuning to nonzero ϕ_0 is similar to allowing a broader class of non-reciprocal and free-energy-driven interactions between the two species further supporting our claim that effervescence is a ubiquitous phenomenon.

We have followed the advice of the referee and split the old figure 7 into a figure 7 and a new figure S3 in the Supplement. In Figure 7 we have discussed the dynamics in two dimension explaining the changes in the order parameters on crossing the boundaries predicted from stability analysis. We have kept figure 6 (old figure 6) showing plots of the eigen-frequencies, as we think that Figure 7 is quite busy already. Finally, we have elaborated on our discussion of σ and added a figure in the extended Data Fig. E6 to illustrate the concept.

IV. RESPONSE TO REFEREE 2

Referee: In their manuscript Saha and Golestanian introduce a novel class of non-linear pattern formation. Building on a recent series of works, they show that antisymmetric couplings between spatially coupled fields can result in the fluctuation-induced restoration of PT symmetry. This rather abstract property has a beautiful real space manifestation dubbed effervescence by the authors. The manuscript report a comprehensive theoretical and numerical study of a minimal model of two conserved species prone to phase separation. The number, and density, of results and comments presented in this manuscript is remarkable. The addition of non reciprocal coupling breaking the PT symmetry typical of passive systems is thoroughly discussed in generic terms independent of the specifics of the minimal model used to conduct the quantitative analysis. The characterisation of genuine spatiotemporal chaos is also very convincing and I am not aware of any earlier results demonstrating this type of physics (in this context). For these reasons, I do believe that this work could ultimately warrant publication in a journal of the caliber of Nature Communications. However, I feel that, in the present form, the manuscript is very hard to read for non-specialists of the blooming field of non-reciprocal matter. I feel that a significant effort in the presentation for the results would be required to make this manuscript accessible to the broad readership of Nature Comm.

Response: We would like thank the referee for their positive evaluation of our work and very helpful comments and suggestions.

Referee: Reporting and discussing the essential results prior to the introduction of the actual model is in general a good idea. It allows the authors to convey their main message in simple words. However, I am afraid that this attempt does not get its goal. The first discussion makes it very hard to understand what the authors are actually talking about. I had to skip the intro, understand the model and go back to the intro again to better understand it (even though I was partly familiar with this line of research). When the authors write: The granular structure for the domains that restore reciprocal interactions, is evident in the droplet size distribution $P(A_{droplet})$ shown in Fig. 2a-b where we observe a prominent peak for a fundamental reciprocal granule and an oscillatory pattern in the probability for the production of larger droplets [Fig. 2a].

Response: We have followed the recommendation of the referee and changed the introduction in the following ways to make sure that the main findings are communicated as clearly as possible. We have added a section at the beginning of the manuscript to introduce the theoretical model and define the parameters that describe the system before showing results from the simulation. We have introduced the $U(1)$ symmetric potential in the manuscript in a manner that we hope will avoid confusion about the generality of the model.

Referee: I must say that I could not understand at all what the concept of a reciprocal granule was at this stage. Why reciprocal? In what sense? How can the PT symmetry of an effective coupling translate is a drop sized distribution?

Response: We have rewritten the manuscript substantially to present the notion of reciprocal granules more clearly and highlight their significance. To do so, we have first defined the value ρ_0 of $|\phi|$ at which $\alpha(|\phi|)$ vanishes resulting in a restoration of reciprocity locally. In domains with $\phi \approx \rho_0$ the interactions are driven by gradients of the chemical potential derived from a free energy. We then proceed to talk about the dynamics observed and explain how we observed short lived small-scale structures that exhibit granularity, hence our motivation to coin the term ‘reciprocal granules’.

Referee: I also feel it would be nice to give a simple pictorial idea of what type of simple dynamics the theory actually models when introducing Eq. 1 2. Figure 1 is very dense and refers to results established later on in the manuscript. This makes the presentation even more confusing. Referring explicitly to one, or a selection of, experiment where this type of modelling could actually apply would also help understanding the model, and give a better sense of the significance of the work.

Response: In the revision, we have made a number of changes to illustrate the idea with more clarity and de-clutter the figure as instructed by the referee. We have changed figure 1 to only illustrate the main dynamical steady-states in two dimensions. We have removed the pictorial representation of ρ and θ . We have also removed density profiles in one dimension. The schematic illustrating what we mean by reciprocal granules and effervescence is now moved to figure 2 where we also present quantitative results relating to the dynamics of the granules/droplets, droplet size distribution, and lifespan. We have added a list of concrete physical systems where we expect to see signatures of effervescence before the concluding remarks.

Referee: As far as I understand it, the physics discussed in this article strongly rely on the $U(1)$ symmetry (breaking) of the complex density field. I therefore wonder how these results could be robust, or extended, to systems involving more than two chemical species. Could the authors comment on this point? I understand that the authors might find these comments rather superficially and might even disagree with them as they are subjective. But I believe that I might not be the only reader having this perception.

Response: We appreciate the referee's (non-superficial) concern. Indeed, effervescence would have had less significance, had it appeared solely for $U(1)$ -symmetric field theories. We demonstrate in the current version of the manuscript that effervescence can be observed for other choices of the free energy, and indeed for other choices of the non-reciprocal terms. We have now emphasised that the $U(1)$ -symmetric free energy and α provide a simple choice for which analytical progress is possible.

To show that ubiquity of effervescence, we solve the NRCH model with 500 different realisations of α . The distribution $P(\alpha)$ for values of α sampled by the system at steady-state is startlingly similar for all effervescent states. In Fig. 1a we show $P(\alpha)$ for all 500 realisations of α in faint lines, and we plot the average distribution as the (bold) magenta line. Note that for $\chi = 0$, we find only travelling waves and effervescent states, whereas with finite χ , bulk phase separated states are found as seen in Fig. 1b.

In the previous version of the manuscript we included results only for $U(1)$ symmetric field theories because the theory is simpler to tackle technically. The analytic grasp that the $U(1)$ symmetric field theory provides is unmatched by any that we know of.

Referee: The long list of references in the intro rightfully credits a number of earlier results, it is however so dense and diverse that it makes it difficult to understand the actual line of research the authors aim at improving. (The reference to classical non-Hermitian physics is also rather biased by the authors' own work. This work was definitely not foundational of the field. A number of experiments have already been reported and the synthetic review article published in Nature physics on the topic. It could provide the reader a more accurate view of the status of this field)

Response: We have shortened the introduction focusing on active matter systems. We have moved the sentences related to active turbulence to the conclusion. We have added a reference to the Nature Physics review article that the referee has suggested [12].

Referee: The authors write "This emergent imperfect PT symmetry breaking with local restoration of reciprocal interactions produces two new states, namely an effervescent wave which is a hybrid state with droplets and a travelling pattern, shown in Fig. 1a-b, and effervescence without the travelling pattern, shown in Fig. 1c-d. Fig. 1d gives a very clear picture whereas I found the sketch of Fig. 1b rather cryptic. A side by side comparison of the two states using the same representation would definitely help understanding their similarities and differences

Response: We have added a figure in the extended data section, see Fig. E1, that brings out these features.

Referee: I could write a long list of suggestions and comments along the same line, but I feel it would be more effective to first give the authors the chance to first rewrite their manuscript to make it more accessible. I can understand that this type of report could be very frustrating given the amount of work and results reported in the manuscript. But in the present form I am afraid this manuscript could appeal to a very narrow audience of specialists.

Response: We are encouraged by the referee's positive comments on our work. We and hope that they find the generality of the results to be unambiguously established, and the presentation of the manuscript improved.

V. RESPONSE TO REFEREE 3

Referee: In their manuscript “Effervescent waves in a binary mixture with non-reciprocal couplings”, Saha et al. investigate a binary mixture of phase separating fields with non-reciprocal couplings. They introduce nonlinear inter-species couplings to the system, and demonstrate that such cross coupling can give rise to spatiotemporal chaos, which they termed “effervescence”. They then characterize the various states observed in such system, and rationalize their transitions by using linear stability analysis. Nonreciprocal active systems have recently attracted much attention due to the rich dynamics invoked by nonreciprocity. Particularly, nonreciprocity is known to be able to induce a PT symmetry breaking and lead to traveling states. In this manuscript, the authors generalize the nonreciprocal couplings, which are usually linear in previous studies, to a nonlinear form, and find various new states at such settings. The simulations and analytics are well executed, and the results are interesting and scientifically solid. However, I have a few concerns that the authors should clarify before I can recommend for publication.

Response: We would like thank the referee for their positive evaluation of our work and very helpful comments and suggestions.

Referee: Whereas nonreciprocal interactions are not uncommon in open systems, I am concerned whether the specific nonlinear nonreciprocal couplings, which change signs when varying concentrations, exist in real systems. If so, the authors should list those systems. If not, then I am afraid the significance of this manuscript might be reduced considerably since changing sign of nonreciprocal coupling is the basis of this manuscript.

Response: This is a very important question. To demonstrate the generality of the conclusion, we have performed additional computations (see above) by essentially generating random forms of the non-reciprocal couplings. We conclude that *any* nonlinearity is likely to produce effervescence. The specific form of α in the previous version was essential as it allowed us to make analytical progress basing our calculations entirely on the $U(1)$ symmetric free energy f and α .

The simplest form for α with the constraint that it is a function of ϕ with well behaved derivatives is $\alpha = \alpha_0 - \alpha_1 |\phi|^2$. This particular form can *only* change sign if $\text{sign } \alpha_0 = \text{sign } \alpha_1$. We now introduce the most general form for α in the manuscript and show numerically that the resulting steady states are similar to those that we observe using the $U(1)$ symmetric f . This means that any nonlinear coupling should produce the effect that we see here. We have listed a number of microscopic models that could reduce to the minimal NRCH with non-reciprocal couplings generally being nonlinear.

Referee: The “effervescence” is supposed to be the major selling point of this paper. But I fail to spot the unique/nontrivial properties of such state. Is it merely a spatiotemporal chaos manifested in nonreciprocal systems? If not, how is different from the spatiotemporal chaos exhibited in other nonlinear systems, e.g. the Kuramoto-Sivashinsky system? To what extent is it different from the micro-phase separation found in ABP (e.g. PRL 125/178004/2020, PRX 8/031080/2018), which also contains fluctuating droplets? Are there some properties that are specific to the “effervescence” here, yet unidentified in previous studies? Highlighting the unique properties of the effervescent states, I believe, can make the manuscript more compelling.

Response: This question has already been touched upon in responses to some of the other points. Here, we present a comparison on the phenomenology observed in our system and other systems studied so far in the literature.

We will first compare our system to the microphase separation found in ABP type systems, particularly ABP+. The most important feature of the ABP+ system is that it is active due to terms introduced in the dynamics that cannot be derived from a chemical potential and are fourth order in gradients. This can be immediately contrasted to nonlinear NRCH where we introduce terms in the chemical potential which appear at second order in gradients in the dynamics of the fields.

The papers that the referee mentions show that the effect of active terms in ABP+ is to produce a negative Laplace pressure that leads to the proliferation of droplets, which scale with system size, in *two* dimensions and in the presence of *stochastic terms* in the evolution. Effervescence occurs in one dimension as well as two dimensions, and for a *deterministic* system, while the droplets exhibit granularity, containing units whose size remains finite in the course of the dynamics.

A preprint authored by one of us [6] shows that cross couplings in the interfacial tension is essential to find evidence of negative Laplace pressure. In this paper we linked the stationary steady-states of a class of nonlinear NRCH to the stationary-states of a passive system with an effective free energy. Most importantly, interfacial tension of the type we consider here does not produce negative Laplace pressure upon integrating across the interface of a droplet in two dimensions. Our preliminary calculations show that this is likely to be true for a class of nonlinear NRCH for which

we can define an effective free energy.

The connection to the Kuramoto Sivashinsky equation is through the observation that in both nonlinear NRCH and the KS equation an effective white noise is generated due to non-linearities.

Referee: Since "effervescence" is the main point here, it will be useful to provide some results on the characteristic length and time scales of such state, and argue the parameters that control these scales.

Response: Since $\alpha_{0,1}$ appear at the same order in gradients, they feature in a linear combination in the only characteristic frequency that can be constructed using the parameters of the system. We can associate a time scale with the angular frequency of the travelling wave. Our analysis shows that the dependence varies quadratically with the wavelength.

Effervescence involves a number of scales. The droplets appear and disappear stochastically, as can be seen from the new results that has been added to the revised manuscript to highlight this aspect. In Fig. 2f, we have plotted the persistence fraction of droplets averaged over the simulation box as a function of time. For each initial configuration we observe several merging events after which the persistence fraction drops, as visible in the faint line in the figure. Upon averaging over about 200 initial conditions, we find a nearly exponential decay from which we can extract a characteristic timescale for the dissolution of the droplets. Comparing this timescale to the time period of the wave, we find that droplet dynamics is an order of magnitude faster than the wave motion, which is consistent with the plateau in the power spectrum.

Furthermore, we have looked at the correlation between droplet size and lifespan in one dimension. The droplet sizes and lifetimes bear a rather simple correlation. A larger droplet survives longer than a smaller one. We added a plot to show the correlation between them. We have also plotted the fraction of droplets that survive after a time t in the revised figure 3 c.

Referee: If I understand correctly, it is claimed in the manuscript that the interplay between linear and nonlinear non-reciprocal interactions is crucial to the emergence of spatiotemporal chaos. Yet, spatiotemporal chaos has been found in systems with merely linear nonreciprocal coupling, see e.g. arxiv:2306.08868. Can the authors comment on this? And what are the differences between the chaos exhibited in the two systems?

Response: This particular paper has reported spatiotemporal chaos for active-passive mixture with no-flux boundary conditions in a finite box. While it is as yet unclear what determines the wavelength of the travelling wave, our analysis predicts that for constant α the waves are always stable below the threshold set by the Eckhaus criterion. We can only speculate that perhaps this type of chaos could originate from the travelling wave unable to 'settle' down into regular oscillations due to the constraint of box size. We have observed similar behaviour within circular domains (unpublished work) but they always turned out to be transient. It difficult to comment on the nature of these solutions without discussions with the authors. We also note that the model in this preprint is different in the selection of f . We find it plausible that NRCH can likely accommodate chaotic solutions under different conditions. This paper definitely presents an interesting combination of boundary condition and choice of f . We further emphasize that we do not claim that effervescence is the only form of chaos in these systems.

Referee: In the last paragraph on page 7, the authors wrote: "However, a globally constant θ as a solution of Eq. (20) is ruled out by number conservation as argued in the previous section (also see Fig. 5b), suggesting that for $\alpha_0 > \alpha_1$ finite-sized stable droplets can be created with constant $|\phi|$ and a field θ inside that can be determined using if the boundary conditions at the edge of the droplet is known." Can the authors elaborate on the emergence of "finite-sized" and "stable" droplets out of this instability? In other words, how can tell that this instability (Eq. 20) leads to "finite-sized" and "stable" droplets? What determines the size of the droplets?

Response: We have qualified our approach better in the current version of the manuscript. We state - 'In contrast to the stability analysis in the last section, where we tested the robustness of the travelling waves that are global solutions of the NRCH equation, here our reference state exists over a finite area, similar to the concept of 'local' stability analysis introduced in [7, 8].' This means that the droplet size is not necessary in the analysis. Moreover, it implies that we cannot predict a stable size from this type of analysis.

Referee: In Fig. 3e, it seems that the red line in the right panel can be extended all the way to $\alpha_0 = \alpha_1$, which means that the Effervescence state also contains propagating modes, right? Also, I see another branch with higher frequencies. What does this branch represent?

Response: The power spectrum of the effervescent state (without the travelling wave) does have multiple peaks.

However, one should note that the intensity of the peak for the effervescent wave is orders of magnitude higher than those in the case of effervescence, which can be considered as background fluctuations. This is clear from the S2 where we plot the peak of the power spectrum $S(\omega)$ as a function of α_1 keeping α_0 constant. Also important is to consider the equal time structure factor $S(\mathbf{k})$ (see Fig. 3d) for the two versions of effervescence and notice that no wavelength is selected in effervescence. The time averaged spatial fluctuations are isotropic. We have added a comment to this effect in the main text.

Referee: In Fig. 2a, how are droplets identified from the field values?

Response: We have added a series of snapshots of the field ϕ highlighting the various events that occur in effervescence. The blue contour is drawn at $\phi = 0.8$ to identify the droplet. In general we identify a droplet by looking at the histogram $P(|\phi|)$ and picking a value that is intermediate between the two peaks seen in Fig. 2e. For effervescent states the two peaks meet in which case we use a method of trial and error to fix a value that involves checking that the threshold chosen does not change the distribution qualitatively. A more technical description of the method of trial and error is that we use Matlab to identify a single contour that defines the landscape of $|\phi|$ and we check for consistency between the threshold so obtained with that from $P(|\phi|)$ to decide the threshold.

Referee: The authors claimed that: "Effervescence gives rise to spatiotemporal chaos, and the emergence of an effective noise from the deterministic nonlinear dynamics, due to nonlinear non-reciprocal interactions." Can the authors explain in an intuitive picture how effervescence can give rise to spatiotemporal chaos?

Response: We have changed the presentation of the manuscript to clarify the description of effervescence. We characterize the chaotic dynamics by measuring the Lyapunov exponent and probe various related characteristics such as the power spectrum of the fluctuations. We have also added snapshots that demonstrate how the creation and annihilation of the droplets take place in the system. Taken together, we hope that these can convey an accurate picture of the spatiotemporal chaos that can be observe in our system in the effervescent state.

VI. RESPONSE TO REFEREE 4

Referee: In the manuscript “Effervescent waves in a binary mixture with non-reciprocal couplings” by Suropriya Saha and Ramin Golestanian, the authors explore conserved field models, describing e.g. mixtures of two particle species, where nonreciprocal interactions appear at linear and nonlinear order, allowing for spatially strongly heterogeneous interactions. The paper reports the formation of two spontaneously formed dynamical steady states, called effervescence and effervescent traveling waves, respectively, the first characterized by spatiotemporal chaos, the second by a combination of spatiotemporal chaos and regular traveling waves. The authors analyze the model by numerical and analytical methods and are able to predict the rough characteristic of the state diagram using a linear stability analysis. They describe the emergence of the exotic phases as a complex interplay between the formation of traveling waves and spontaneously forming and dissolving droplets of fluctuating shape. For a special case of composition of the two components of the mixture, they deduce the exact dispersion relation of the traveling waves and establish their stability. The authors also identify the droplets as local areas of restoration of reciprocity, which are marginally stable.

Response: We would like thank the referee for their positive evaluation of our work and very helpful comments and suggestions.

Referee: The paper gives a nice account of the rich and complex behaviour that can arise in minimal models of active matter systems with non-reciprocal interactions. In particular, a new type of chemical spatiotemporal chaos is observed. It is argued to arise from spatial heterogeneities in the non-reciprocal couplings that change sign as a function of the amplitude of the scalar fields or “species”. It is also pointed out that this phenomenology can be understood as an instance of imperfect breaking of PT symmetry, involving fluctuating domains in space where the symmetry is temporarily restored. All these results make the paper a rich and valuable contribution to its specific research field and of likely interest to a specialised audience of experts, who can cope with the paper’s rather technical style.

Response: To make the manuscript more accessible to the general readership of the journal, we have introduced major changes both in content and in style. In particular, we have revised the introduction thoroughly developing the material at a slower pace. We have also added a list of physical systems where effervescence is likely to be found. Moreover, we have shown that effervescence is likely to appear in the most generic variant of two species NRCH that can be written down.

Referee: Yet, the paper does not penetrate more deeply into the physical mechanisms behind these observations and, for example, describe or explain the process of droplet annihilation in detail.

Response: In the revision, we have added additional material in this regard, including investigation into the timescales associated with droplets. We probe the droplet dynamics by defining and calculating, from numerical data, the persistence fraction of the droplets as a function of time and by averaging over many initial conditions. Our analysis shows that droplets vanish with a timescale one order of magnitude smaller as compared to the time period of the accompanying travelling wave. We find that although droplets vanish with a selected timescale, the average behaviour does not capture events where a droplet persists longer than the picture taken on an average would predict.

We have also included a plot of droplet sizes and droplet lifetimes to show that they bear a linear correlation, a rather strikingly simple result in a complex system. We have added these results including the distribution of the droplet size and droplet lifespan in the methods section.

Referee: It also does not attempt to offer an explanation for the presence of spatiotemporal chaos in the background of the droplets.

Response: We have changed the presentation of the manuscript to clarify the description of effervescence. We characterize the chaotic dynamics by measuring the Lyapunov exponent and probe various related characteristics such as the power spectrum of the fluctuations. We have also added snapshots that demonstrate how the creation and annihilation of the droplets take place in the system. Taken together, we hope that these can convey an accurate picture of the spatiotemporal chaos that can be observe in our system in the effervescent state.

Referee: The analysis in the case of arbitrary compositions does not go beyond linear stability of homogeneous configurations, but the authors claim, on this basis, the existence of particular forms of inhomogeneous stationary solutions. This does not seem to be justified by further qualified arguments. Also with respect to other aspects of the phenomenology, one might have hoped for a more thorough analysis and clearer presentation, as detailed in the

comments below.

Response: The analysis for arbitrary composition is an important part of the analysis because we are exploring number conserving systems where mean density is an important tuning parameter that determines phase composition and the number of phases. This is in contrast to classic reaction-diffusion or complex Landau-Ginzburg type models without conserved fields. In [1] it has been shown that composition can be tuned to cross a line of Hopf bifurcation, accompanied by two dimensional moving patterns. Here, we find a consistent change from effervescence with steady states where phase separation dominates producing pulsating interfaces.

We claim the existence of continuously varying chaotic stationary-states on the basis of extensive simulations in both one and two dimensions, as summarised in Fig. 7 of the main text and S3 of the supplement. The linear stability analysis and the results from direct numerical simulations are consistent with each other. The points where the mixed state becomes unstable matches exactly with the results of the linear stability analysis. We find that the exceptional point and the Hopf bifurcation points coincide with rapid qualitative changes in dynamics. To highlight this we have added a new figure 8 which now contains results from simulations in two dimensions only allowing us to present these aspects with clarity. Results from simulations in one dimension are now in the Methods section.

We have also added a section in the Methods sections where we have highlighted that dynamics at non-zero average composition can be mapped to a more general form for the nonlinear non-reciprocal interaction with vanishing composition. The complexity of the dynamics at nonzero ϕ_0 makes it manifestly clear why analytical results are difficult to obtain in this regime. As we have emphasised in the text, $\phi_0 = 0$ is a special point where analytical results are possible.

Referee: A more severe issue might however be the question who is actually the targeted audience of the paper. The authors clearly attempt to put their work into a broader context and to describe it verbally, in particular in the first part of the paper. Yet, this description remains somewhat vague, while, at the same time, abstract and technical.

Response: We are encouraged by the referee's judgement that the manuscript contains narratives that are addressed to the general reader and that this will lower the barrier for its readership. It is, of course, important to broaden the scope while preserving the level of rigour that is necessary for the presentation. We are surprised, however, that the referee has chosen to label this aspect of our work as being technical, and suggest that it goes against our attempts to make the work accessible to the general readers. It is indeed standard practice in journals such as nature communications to strike a balance between generalist narratives, use of visual aid to its maximum capacity, and supplementing it with highest level of technical analysis and rigour. We believe this the most appropriate style for writing for the journal. We also believe that this is exactly how we have written the manuscript. Of course, any such complex article can always be improved and we are grateful to all four referees for their many suggestions for improvement, which we have fully implemented in this revision.

In particular, we have added two new sections to the paper to emphasise the generality of our results. The first is a section on the ubiquity of effervescence where we show with an example that qualitatively similar results are obtained for Cahn-Hilliard dynamics with arbitrary nonlinear interactions. While we have put ample emphasis on the point of generality, we have provided justification for the need to start with a simple form for the free energy and the nonlinear non-reciprocity.

Referee: I'm afraid that most non-experts will almost certainly not be able to understand either the verbal characterisation or the following technical discussion, nor will they be able to appreciate the possible interdisciplinary pertinence and applications of the work (if any), while the small group of initiated experts who do would most probably have preferred a more extensive classical physics-article style of the exposition of the topic.

Response: We do not understand how the referee can draw such a conclusion with such level of certainty. To some extent one can argue that any paper published in a journal that is aimed at general readership will suffer by this in part. The aim of high visibility journals is to bring novel and important results to the attention of as many scientists as possible. Of course, only the specialists in the field will be able to understand all the technical aspects. We fail to see how the arguments provided by the referee would support a judgement that our results are not novel enough to warrant such exposure.

We hope that the revised manuscript can convince the reader and the referee that our paper brings to light a new aspect of non-reciprocity in scalar mixtures hitherto unexplored. We have also shown with numerical evidence and theoretical arguments the ways in which the nonlinearity can be 'constructed' to see the behaviour we have highlighted in our work.

Referee: In summary, I think it is fair to say that the authors have discovered a rich and interesting phenomenology but have not fully explained or analysed it, nor have they provided convincing arguments for why it should be of

interest to an interdisciplinary readership or catered their presentation to it. Altogether, I therefore conclude that the paper is, in its present form, not suitable for the chosen journal and its broad interdisciplinary readership, but would be better redirected to and rephrased for classical physics journals, such as the Physical Review or similar.

Response: We respectfully disagree with the referee and hope that they will be persuaded to change their view concerning publication of our work in Nature Communications, where we believe the manuscript belongs.

Referee: 1- In the introduction and throughout the paper, the authors use the terms non-reciprocal interactions, activity, and PT symmetry breaking synonymously and should motivate and explain this. They could also mention Refs. [T. Frohoff-Hülsmann et al., Physical review. E 103 4-1 (2020)], [T. Frohoff-Hülsmann et al., arXiv:2301.05568 (2023).] which have investigated the phenomenology of field equations with nonreciprocal dynamics and especially the nonreciprocal Cahn-Hilliard model in great depth.

Response: As we specify in the introduction, in this paper non-reciprocity is the only source of activity in the system. In the rest of the paper the word non-reciprocal is synonymous with active. We agree that PT symmetry breaking and non-reciprocity are not synonymous, however we find no instance where we allude to them interchangeably in the past version of the manuscript. Following the referee's suggestion we have added the Ref. [T. Frohoff-Hülsmann et al., Physical review. E 103 4-1 (2020)]. The second reference was already added in the previous version of the manuscript and we have updated it as it has been published in PRL.

Referee: 2. It seems to me that neither Ref. [15] nor Ref. [16] report purely temporal oscillations, as stated in the manuscript.

Response: We have removed the mention of temporal oscillations. It was a typographical error as we explain how in number conserving systems purely temporal oscillations are not possible in the section 'Travelling waves'.

Referee: 3. The statement: "The transition to traveling patterns occurs upon running the parameters of the model such that an exceptional point is crossed" is incorrect because all cited references [15,18,21] also mention the possibility of a transition to a traveling pattern state via an oscillatory instability (within the terminology of the authors called Hopf bifurcation).

Response: We have removed this sentence and we discuss the linear instabilities, through crossing an exceptional point and the Hopf bifurcation, only in the section 'Effect of composition'.

Referee: 4. The definition of exceptional point transitions seems slightly incorrect, because it neither mentions the important scenario of an EP transition from a static state of broken continuous symmetry (not fully mixed) to a dynamical state, nor is an eigenvalue with nonzero imaginary part a general feature of an exceptional point.

Response: We agree with the referee that coalescence of two eigenvalues and eigenvectors is the correct definition of an exceptional point and this is precisely how we define them. The referee is also correct in stressing that nonzero imaginary part is not a general feature of an exceptional point. We have modified the relevant sentence to avoid the misconception. We have already stressed that in the introduction that crossing an exceptional point leads to dynamical states - 'When activity, i.e. the strength of non-reciprocity, is strong enough to win over the thermodynamic forces driving the system towards bulk phase separation, the system reaches novel steady-states that break the parity and time-reversal (PT) symmetry of the bulk-separated equilibrium state.'

Referee: 5. On page 2 the authors state that "An emergent feature of the non-reciprocal interactions implemented at the linear level in a mixture of two species is the spontaneous breaking of space-translation, time-translation, and polar symmetries". It should be made clear that only spontaneous breaking of time-translation and polar symmetries are exclusive features of nonreciprocal mixtures.

Response: This particular sentence changed during the revisions we made and does not appear in the manuscript anymore. In principle we agree with the referee on this point.

Referee: 6. In the verbal discussion, the variables and parameters are not generally understandable without first reading the more mathematical exposition further below. For example, "alpha" should be better defined together with the model equations. Also, the description in the first paragraph of page 3 is not very clear. Should not the explicit form of alpha, given in Eq. (4), be introduced here? What is the motivation for this particular choice? In

general, it would seem advisable to adopt more of an extensive and explicit classical physics-article style, e.g., by introducing the model earlier in the text and discussing the meaning of the parameters afterwards less cryptically.

Response: We believe that these issues are now adequately taken care of as summarised in the response to the referee in the earlier part of this response.

Referee: 7. A clear distinction between the terms droplet and granule would be helpful, e.g, Fig. 1b suggests that a droplet is formed by the pairing of two reciprocal granules, while on page 3 it is stated that a droplet ‘is created by the fusion of two or more reciprocal granules’.

Response: We have added precise definitions of both in the paragraph just before the section ‘Quantifying effervescence’. We say - ‘ We notice two types of phase separated domains in the system. *Reciprocal granules* are spontaneously generated in copious amounts at the smallest possible length-scales. They either disappear or coalesce to form droplets. Their small size and short lifespan distinguishes them from longer-lived larger droplets. ’

We have removed a part of the schematic from Fig. 1b as it did not do justice to the complex processes involved in coalescence of these highly dynamic granules and droplets. We have instead included snapshots of the field $|\phi|$ from simulation in Fig. 2 to sketch the various events occurring over different time scales.

Referee: 8- The authors state on page 3 that “domains grow and shrink by losing or gaining matter”. Could the precise mechanism be mentioned?

Response: As can be seen in the videos droplets grow in size while remaining at roughly the same position. The mechanism could be diffusive processes akin to Oswald ripening.

Referee: 9- Similarly, it might help to provide clear definitions of the term droplet and $\mathcal{A}_{droplet}$, used to obtain the numerical data presented in Fig. 2?

Response: $\mathcal{A}_{droplet}$ is the area enclosed by the contours depicted in Fig. 2b determined from the field $|\phi|$. The blue contour is drawn at $\phi = 0.8$ to identify the droplet. In general we identify a droplet by looking at the histogram $P(|\phi|)$ and picking a value that is intermediate between the two peaks seen in Fig. 2e. For effervescent states the two peaks meet in which case we use a method of trial and error to fix a value that involves checking that the threshold chosen does not change the distribution qualitatively. A more technical description of the method of trial and error is that we use Matlab to identify a single contour that defines the landscape of $|\phi|$ and we check for consistency between the threshold so obtained with that from $P(|\phi|)$ to decide the threshold. .

Referee: 10. In Fig. 2a, the peak of $P(|\phi|)$ at ρ_0 is not apparent. Maybe the authors meant to refer to Fig. 2b? If so, also here the density ρ_0 should be indicated.

Response: We have indicated the peak in Fig. 2e as suggested.

Referee: 11. In Eqs. (1) and (2), the functional-derivative notation (as in the inlined formula for μ_i above these equations) would seem more appropriate.

Response: Functional derivative is correct for the inline formula as it includes the bulk part of the free energy f and the contribution from the interface. In Eqs. (1) and (2) a partial derivative notation is correct as f is a function of ϕ_i only and not gradients of the fields.

Referee: 12. The authors should distinguish the terms spatiotemporal chaos and effective noise. How would the effective noise relate to an additive Gaussian white noise, as typically used in stochastic versions of field models of the type analysed in their work?

Response: We do not think that we have used the terms ‘spatiotemporal chaos’ and ‘effective noise’ interchangeably. Added noise is an entirely different concept as it introduces coupling to a bath. Our system is deterministic.

Referee: 13. Wouldn’t $\Sigma_1 \neq \Sigma_2$ already for the “reciprocal” case $\alpha = 0$ mean a violation of the fluctuation dissipation relation?

Response: We are not sure what the referee is alluding to here. We cannot find this reference in our manuscript.

14. On page 4, it is stated that the form of the free energy in Eq. (3) leads to microphase separation, while the form chosen in Eq. (6) leads to bulk phase separation. Could the authors please resolve this apparent discrepancy?

Response: In one dimension the $U(1)$ symmetric free energy leads to stable microphase separation in contrast to two dimensions. We have removed this sentence from the paper because we have only talk about dynamics in two dimension in the main paper.

Referee: 15. In Fig. 3e it is not visible by eye that the peak in the effervescent wave phase actually fully disappears at some specific value of α_1 , as stated. The placement of the right black line therefore appears rather arbitrary.

Response: We have removed the black line as it was mainly to guide the eye. The old version of the paper had an error as we did not specify that the heat-map in Fig. 3e is of $\log[S(\omega)]$ and not $S(\omega)$ which means that the peaks in $S(\omega)$ are comparable to the background fluctuations while in the effervescent wave the peak of $S(\omega)$ is at least an order of magnitude or more higher than the background fluctuations while for travelling waves the peaks is 5 orders higher. We have added a discussion in the section ‘State diagram’.

Referee: 16. It is not clear for which composition the numerical state diagram in Fig. 3 was computed.

Response: For $\langle \phi_{1,2} \rangle = 0$. We have added this missing piece of information in the beginning of the section and ‘State diagram’.

Referee: 17. The term bifurcation seems to be used in a somewhat generalised/overloaded (and maybe not acceptable) way. Isn't a bifurcation point always related to the loss of stability of a fixed point? The authors also use this term for the splitting of the real parts of the eigenvalues of the stability matrix, although none of the real parts changes its sign.

Response: We have replaced the word ‘bifurcation’ with the word ‘transition’ explicitly stating what we mean mathematically.

Referee: 18. From Fig. a,b there seems no strong connection between the exceptional point indicated by the red line and the type of steady state the system adopts. Hence, the linear stability analysis does not seem to be really informative.

Response: We have stressed in the manuscript that indeed there is a connection between the linear system crossing an exceptional point and qualitative changes in the dynamics as we mention in response to the referee’s query regarding arbitrary composition.

Referee: 19. What’s the meaning of the grey circles in Fig. 6?

Response: Those are the points where one or both of the two real eigenvalues change sign and thus stability. We have added the sentence - ‘Finally, a pair of real eigenvalues could both change signs signalling an instability where perturbations grow and lead to the formation of a bulk separated state indicated by the grey circles in Fig. 7’ - to clarify this point.

Referee: 20. The authors should define the meaning of $\langle \rangle$

Response: We have added a definition in the beginning of the section ‘State diagram.’.

Referee: 21. On page 7, the authors claim that “for $\alpha_0 \gg \alpha_1$, we should find stable droplets submerged in a traveling wave”. The basis for this is not clear.

Response: This statement is based on numerical results which as summarised in Fig. E4 and supplemental video S3. We have expanded the sentence as - ‘Our analysis suggests that for $\alpha_0 \gg \alpha_1$, droplets are more stable in comparison to the case where $\alpha_0 \gtrsim \alpha_1$. In simulations, we explore the region of the state diagram beyond that shown in Fig. ??a and find stable droplets submerged in a travelling wave (see Extended Data Fig. ?? and Movie S3 in the supplement)

for which $P(|\phi|)$ consists of two sharp peaks, the first at $|\phi| < 1$ and the other at $|\phi| \simeq \rho_0$.’

-
- [1] S. Saha, J. Agudo-Canalejo, and R. Golestanian, Scalar active mixtures: The nonreciprocal cahn-hilliard model, *Phys. Rev. X* **10**, 041009 (2020).
 - [2] T. Frohoff-Hülsmann and U. Thiele, Nonreciprocal cahn-hilliard model emerges as a universal amplitude equation, *Phys. Rev. Lett.* **131**, 107201 (2023).
 - [3] L. Würthner, F. Brauns, G. Pawlik, J. Halatek, J. Kerssemakers, C. Dekker, and E. Frey, Bridging scales in a multiscale pattern-forming system, *Proceedings of the National Academy of Sciences* **119**, e2206888119 (2022).
 - [4] F. Brauns, G. Pawlik, J. Halatek, J. Kerssemakers, E. Frey, and C. Dekker, Bulk-surface coupling identifies the mechanistic connection between min-protein patterns in vivo and in vitro, *Nature Communications* **12**, 3312 (2021).
 - [5] F. Brauns and M. C. Marchetti, Non-reciprocal pattern formation of conserved fields (2023), [arXiv:2306.08868](https://arxiv.org/abs/2306.08868) [nlin.PS].
 - [6] S. Saha, Phase coexistence in the non-reciprocal cahn-hilliard model (2024), [arXiv:2402.10057](https://arxiv.org/abs/2402.10057) [cond-mat.soft].
 - [7] J. Halatek and E. Frey, Rethinking pattern formation in reaction–diffusion systems, *Nature Physics* **14**, 507 (2018).
 - [8] J. Denk and E. Frey, Pattern-induced local symmetry breaking in active-matter systems, *Proceedings of the National Academy of Sciences* **117**, 31623 (2020).
 - [9] G. Tucci, R. Golestanian, and S. Saha, Nonreciprocal collective dynamics in a mixture of phoretic janus colloids (2024), [arXiv:2402.09279](https://arxiv.org/abs/2402.09279) [cond-mat.soft].
 - [10] Y. Duan, J. Agudo-Canalejo, R. Golestanian, and B. Mahault, Dynamical pattern formation without self-attraction in quorum-sensing active matter: The interplay between nonreciprocity and motility, *Phys. Rev. Lett.* **131**, 148301 (2023).
 - [11] T. Frohoff-Hülsmann and U. Thiele, Localized states in coupled Cahn–Hilliard equations, *IMA Journal of Applied Mathematics* **86**, 924 (2021), <https://academic.oup.com/imamat/article-pdf/86/5/924/40744935/hxab026.pdf>.
 - [12] K. Ding, C. Fang, and G. Ma, Non-hermitian topology and exceptional-point geometries, *Nature Reviews Physics* **4**, 745 (2022).

Response to Referees: Effervescent waves in a binary mixture with non-reciprocal couplings

Suropriya Saha^{1,*} and Ramin Golestanian^{1,2,†}

¹Max Planck Institute for Dynamics and Self-Organization (MPIDS), D-37077 Göttingen, Germany

²Rudolf Peierls Centre for Theoretical Physics, University of Oxford, Oxford OX1 3PU, United Kingdom

(Dated: October 10, 2024)

I. SUMMARY OF CHANGES IN THE MANUSCRIPT

We have thoroughly revised the manuscript taking into account the feedback from all four referees. We have added new results that establish the generality of the phenomenology presented in this manuscript. Here is a summary of the changes that we have made in the manuscript.

- We have split the old Fig. 2 into two figures. New Fig. 2 now illustrates the definition of reciprocal granules and droplets, and spells out our definition for effervescence. New Fig. 3 contains results from quantitative analysis of the steady states that we observe.
- We have revised the text extensively. The revised parts are highlighted in blue. We have extended the section where we describe the phenomenon of effervescence and introduced the concept of reciprocal granules and droplets. We have added new results to quantify effervescence showing that reciprocal granules are indeed short lived compared to droplets.
- We have extended the section on 'Ubiquity of effervescence' emphasising that the change of sign for the effective coefficient of non-reciprocity is an emergent behaviour that occurs for a wide range of general nonlinear non-reciprocal interactions.
- We have rewritten the conclusion section reiterating the properties of effervescence, stating the generality of the results and drawing comparisons between nonlinear NRCH and other field theories to explain that effervescence does not fit into any other category of dynamics explored in the literature so far.
- We have modified Fig. 5 to show reciprocal granules and droplets in one dimension.
- We have removed material from the Methods section that is not needed for immediate understanding of the main text and compiled them in a Supplement, including the previous 'extended data' figures.

II. RESPONSE TO REFEREE 1

Referee: The authors have significantly improved the structure of the manuscript. It is now written in a style that is more accessible to a broad audience. However, much of the content is still quite technical. Nonetheless, I think the manuscript might be suitable for Nat Comms, given the significant recent interest in mass-conserved, non-reciprocally coupled systems. Before I can recommend the manuscript for publication, the authors should address the following remaining concerns.

Response: We would like thank the referee for their positive evaluation of our work and very helpful comments and suggestions. We have revised the manuscript to address the remaining concerns of the referee.

Referee: It is not clear what microscopic dynamics would lead to a changing sign of the non-reciprocal coupling in a real physical system. This sign change is the crucial ingredient in the model that gives rise to the described phenomena of "effervescence". At the end of the new section on "real physical systems" the authors appeal to the generic nonlinearity of non-reciprocal coupling that arises from coarse-graining microscopic models. However, a

* [suopriya.saha@ds.mpg.de](mailto:suropriya.saha@ds.mpg.de)

† ramin.golestanian@ds.mpg.de

changing sign is a much stronger requirement, and I'm not aware of a case where this arises out of a microscopic model that might have an experimental realization. Overall, I did not find the section on potential experimental systems very convincing, since this key aspect is not addressed.

Response: From a phenomenological perspective, when the symmetries and conservation laws provide conditions for non-reciprocity to be realised, then there is nothing preventing the non-reciprocal coupling to take on a nonlinear form. This means that the spirit of a Landau expansion can be applied both to the reciprocal part of the chemical potential and to its non-reciprocal part. In this sense, nonlinear terms will represent non-reciprocal interactions that require the presence of additional structure such as three-body interactions, allosteric-like effects where dimers and trimers made of one species interact differently as compared to two or three single units, or interactions mediated by other species as present in quorum-sensing systems. While the mechanistic details of these different microscopic realizations will be different, one can naturally expect them all to give rise to generic nonlinearities provided in Landau-like expansions.

We would like to emphasize that committing to a specific microscopic picture to build a phenomenological theory is not a natural strategy. To make the argument clear, we would like to point out that expecting this requirement as a condition for publication of our work is equivalent to not allowing Landau theory of superconductivity to be published until the BCS theory is developed for Cooper pair formation through electron-phonon interactions. In this hypothetical scenario, Landau could have argued that his phenomenological theory provides an overarching umbrella for the to-be-discovered BCS theory of superconductivity and all other forms of superconductivity which could lead to condensation of other quasi-particles. Our work is the first to introduce this phenomenology to the field of non-reciprocal active matter, and we have subsequently supplemented it with a great deal of additional work to convince the referees. These additions, as have been required by the referees, have made the manuscript much longer and more technical than originally designed by us, and this has led the referees to complain about the presentation now being too technical while simultaneously asking for more work to be done to convince them. We do not believe that this is justified.

Nevertheless, we have performed additional calculations by considering a more general polynomial form for the effective non-reciprocity coupling and used random coefficients to demonstrate the robustness of the phenomenon of Effervescence. The results are now added to the manuscript as the new Fig. 9 and in the section 'Ubiquity of Effervescence'. We would also like to highlight that the composition dependent phenomenon reported in our earlier versions of the manuscript can be written in the more general polynomial form as expanding around a non-zero composition will generate all those terms. Therefore, effervescence occurring for nonzero composition also provides indirect proof of its generality.

Referee: Related to this issue is that some of the phenomena described by the authors, such as spatio-temporal chaos and transversally traveling fluctuations (see Fig. 8a) do, in fact, not require any nonlinearity of the non-reciprocal coupling at all (see Ref. [56]). This should be mentioned in the manuscript and the authors should disambiguate more clearly between phenomena that require the sign-change in the non-reciprocal coupling and those that don't.

Response: Transverse fluctuations are not a generic feature of the phenomenon that we report here. They appear occasionally when we allow arbitrary nonlinear couplings between the fields. We have added a sentence in the section 'Ubiquity of effervescence' to stress the fact. We have also listed the defining features of effervescence in the conclusion.

This particular paper has reported spatiotemporal chaos for active-passive mixture with no-flux boundary conditions in a finite box, whereas the spatiotemporal chaos we report occurs with periodic boundary conditions. Our analysis predicts that for constant α the waves are always stable below the threshold set by the Eckhaus criterion. We can only speculate that perhaps the type of chaos reported in this paper could originate from the travelling wave unable to 'settle' down into regular oscillations due to the constraint of box size. We have observed behaviour similar to that in the paper within circular and square domains (unpublished work) but they always turned out to be transient. It difficult to comment on the nature of these solutions without discussions with the authors.

Spatiotemporal chaos can indeed arise in other systems with two coupled fields, the most celebrated being the Kuramoto-Sivashinsky equation. We mention this in the conclusion. Moreover, in the current thoroughly version of the conclusion, we have drawn parallels between our findings and phenomenon reported for active phase separating systems and reaction diffusion systems with or without number conservation.

I still find some of the terminology introduced by the authors unnecessary and not sufficiently well motivated and defined. There is still no precise explanation of what a "reciprocal granule" is and what makes it "reciprocal".

Response: A reciprocal granule is formed when reciprocity is restored in localised spatial domains and phase

separation resumes leading to the formation of a small granule that either disappears or leads to the formation of larger droplets via coalescence or growth. This is highlighted in the new Fig. 2, and Movies S1-S4. These structures are called granules as they are the small structures that provide a discrete unit for the formation of larger droplets, as clearly demonstrated from the quantization of the size of the droplets.

Droplets have fluctuating interfaces, as seen in Fig. S1 in the Supplement, where we have presented a few examples, while reciprocal granules are stabilised by surface tension and are disk-like in two dimensions. Droplets can have a more intricate structure where one droplet is encapsulated by the other. We have added these arguments in the section before ‘Quantifying effervescence’ in the main text and we have presented a statistical analysis of shapes of the reciprocal granules and the droplets to clarify the physical properties they represent in the SI.

Also, as we have mentioned in the text that the hallmark of these systems is that the mean composition is a tuning variable due to mass conservation. Tuning the mean composition to nonzero values stabilises phase separation dynamics and produces persistent bulk phase-separated droplets with fluctuating interfaces. In these states reciprocal granules and unstable finite-sized droplets are still continually created and annihilated. Thus there are important differences between droplets and reciprocal granules which we have now emphasised in the main text as well as in the new Fig. 2.

Response: On page 3 they are described as droplets on “the smallest possible length scales”. What sets these length scales? Why the plural, are there multiple such scales?

There is only one length scale set by the surface tension. We have made the necessary correction.

Further the authors write “Their small size and short lifespan distinguishes them from longer-lived larger droplets.” Can this be turned into a quantitative criterion?

Response: In the last revision we added in the Methods section some calculations that related the lifespan of the droplet to its size for simulations in one dimension. We reported that the droplet lifetime was proportional to its size (see Fig. S2). Over the variation of about three decades of droplet size, the lifespan of a droplet varies similarly, see Fig. S2 in the SI.

A similar calculation for droplets in two dimension would be extremely challenging as we would have to consider three dimensional data (2 space + 1 time). To circumvent the problem we have used the same strategy we used before to associate a timescale with droplet dissolution. We have calculated a ‘local’ persistent fraction ν to associate a time scale with a droplet of a particular size. The procedure is discussed in the Method and the result is shown in the inset of Fig. 3d of the main text.

Referee: What makes the “granules” “reciprocal” isn’t really clear either. Are there “reciprocal droplets”, too?

Response: As it is widely known in the literature of liquid-liquid phase separation, droplets are the minority phase submerged in a majority phase and they are formed by reciprocal or passive interactions. In the nonlinear NRCH a droplet is formed by a fusion of several reciprocal granules, which are formed when reciprocity in interactions is reinstated locally. Thus droplets are always reciprocal in nature, being formed by passive phase separation.

Referee: In the section ‘Quantifying effervescence’ the authors state on the one hand that the non-reciprocal interactions change sign at $|\phi| = \rho_0$, implying that non-reciprocal coupling is small for $|\phi| \rho_0$. They then write “Within spatial domains where $|\phi|$ is nearly zero, the chasing stops locally”.

There is a typographical error in this sentence. $|\phi|$ should be replaced by α . This has now been corrected.

Referee: Why this is the case remains unclear. It also seems at odds the previous statement, which suggests that the chasing dynamics stops near ρ_0 which is not close to zero. This point needs to be clarified.

Response: Chasing stops as seen in Fig. 3a where we show the dynamics within a droplet where α is close to zero.

Referee: Since the term “reciprocal granules” is still unclear, so is the definition of “effervescence” given in Fig. 2.

Response: We have now highlighted more explicitly our definition of effervescence.

Referee: For instance, I was wondering whether the authors would some of the chaotic phases of the Gray-Scott

model “effervescent” (Pearson, *Science* 261(5118), 189–192, 1993, see <https://pmneila.github.io/jsexp/grayscott/> for an interactive in-browser simulation). The phases reminiscent of “effervescence” (called “Chaos” and “Chaos and Holes” in the interactive simulation) are classified as phase turbulence by Pearson. This suggests that effervescence might simply be a form of phase turbulence, which calls into question the need for a new term.

The Gray-Scott model as studied in [1] is indeed a beautiful model and we have added the reference in the conclusion. We thank the referee for pointing this paper out. A key difference between the holes reported in this paper and the droplets we observe is that they are self-replicating meaning that they are of similar sizes with stringent size selection emerging from the model. As we report in the new Fig. 3, the droplets have a comparatively wider range of sizes and shapes, as seen in the distribution of aspect ratios shown in Fig. S1 of the Supplement. The paper [1] draws comparisons with mitotic divisions as the holes replicate, while we think that Oswald ripening is indicated in the coalescence dynamics that we observe, meaning that larger structures grow at the expense of the smaller ones until the droplet size becomes comparable to the wavelength of the travelling wave; see Fig. 3c of the main text and Fig. S1.

Referring to the work of [2], we note that for the complex Landau-Ginzburg equations, they describe phase turbulence as the type of dynamics where the amplitude can be enslaved to the phase thereby obtaining a description in terms of the latter only. The distribution of the amplitude fluctuations around the stationary value is reported to be Gaussian. This can be contrasted to the similar plot in Fig. 3b where the amplitude distribution is bimodal and thus qualitatively different from that reported in [2]. This leads us to the conclusion that effervescence is not a form of phase turbulence. We are not aware of any instances in the literature where a bimodal distribution of the amplitude (similar to what we show in Fig. 3b) has been reported.

We have added two new paragraphs in the conclusion where we list the defining features of effervescence and we contrast it with relevant models familiar to us and those brought to our notice by the referees.

Referee: Fig. 4 suggests that “effervescence” also occurs in 1D. This should be stated clearly. Could the authors mark granules vs droplets in the kymographs in Figs. 4a and 4b?

Indeed this is an interesting point that deserves to be mentioned. In the main text, we planned to discuss dynamics in two dimensions only. We note however that in contrast with active Model B+ where interfacial activity drives the generation of bubbles in the system meaning that the phenomenon is not possible in one dimension, we find effervescence in one dimension as well, distinguishing it from the known theories. Reciprocal granules, enclosed in red boxes in Fig. 5 appear in one dimension as well. The statistics of lifespan and size, as seen in Fig. S1, are different from two dimensions because Oswald ripening is anomalously slow in one dimension. We have stated this in the conclusion where we compare nonlinear NRCH with similar field theories.

Referee: On page 5 a criterion is described to distinguish TWs from spatio-temporal chaos based on the fluctuation amplitude being small ($|\phi| < 0.1$). At this point in the manuscript, it is not clear that TWs in the model generally have such small amplitude. This should be clarified.

Indeed, the referee is correct in thinking that the travelling waves do not have such small amplitudes when we initiate the simulations from random conditions. In the effervescent phase, the amplitude distribution broadens and while one of its peaks still occurs at relatively small values, for example around 0.5 in Fig. 3b, the distribution $P(|\phi|)$ acquires relatively small but nonzero values close to vanishing $|\phi|$, i.e. at values $|\phi| < 0.1$ in the effervescent states. This can be contrasted with the travelling waves where $P(|\phi|)$ is sharply peaks at $1 - Kq^2$, where q is the wavenumber associated with the sinusoidal travelling wave and orders of magnitude smaller for $|\phi| < 0.1$.

This is yet another advantage of the simple model in Eq. (5) of the main text, as it is numerically very easy to distinguish travelling waves from the effervescence states.

Referee: On page 8, in the context of local stability analysis, the authors write “a globally constant rho as a solution of Eq. (18) is ruled out by number conservation as argued in the previous section (also see Fig. 5b)”. The use of the word “global” in the context of the local stability analysis is a bit confusing since the latter only regards a small subsystem of the whole. Indeed, on the local level, number conservation need not hold, since mass can be redistributed in space.

We have revised the sentence to address this point.

Referee: Fig. 2a is not very clear. The elements shown should be labeled. In Fig. 2b, it might be useful to show the contour at ρ_0 where the non-reciprocal coupling actually changes sign. A

stronger contrast color would also help.
 What do the different colored lines in Fig. 2f show?
 The labels in the inset of Fig. 2d are too small.

We have taken care of these remarks by splitting the old figure 2 into two figures - new figures 2 and 3. We thank the referee for pointing these out. We think that the splitting also streamlines the narrative.

The lines in old Fig. 2f correspond to different initial configurations and the bold line is the average over several initial configurations. We have included this sub-figure in new Fig. 3 and added a line to clarify this point.

Referee: I found the oscillatory features in the distribution of droplet areas Fig. 2d very striking, in particular since they do not appear to have a constant period, but instead the period increases with the droplet size. Given the chaotic nature of the dynamics, it is remarkable that such a feature is found after thresholding of a continuous field. Does it depend on the precise value of the threshold? Does this observation depend on domain size?

We thank the referee for the positive comments and we agree that the result now presented in the new Fig. 3c is very striking. It was indeed one of the strongest motivators for us to propose the new terminology for the two types of spatial domains created by phase separation, namely reciprocal granules and droplets. The prominent peak in the distribution points to the existence of reciprocal granules and the tail of the distribution hints at a subtle size selection of the droplet.

In general we identify a droplet by looking at the histogram $P(|\phi|)$ and selecting a value that is intermediate between the two peaks seen in Fig. 2e. For effervescence states the two peaks meet in which case we fix a value by checking that the threshold chosen does not change the distribution qualitatively. The qualitative features so obtained are completely independent of the domain size. Results reported here are for $N = 2048$. We used $N = 256, 512$ to benchmark the choice of thresholds and we find qualitatively similar results. A large system size was chosen to increase data collection rate and minimise errors due to droplets located at the boundaries.

As the referee observes, the periodicity in oscillations increases with droplet size as coalescence is most effective at small sizes.

Referee: Many figure references in the text are broken (number missing).

We have resolved this issue.

Referee: Page 6: “In the next two sections we will demonstrate that the peak at $-\phi = 1$ in Fig. 2f” should probably be 2e instead.

We have fixed this error.

III. RESPONSE TO REFEREE 2

Referee: I am very impressed by the efforts the authors have devoted to improving the quality of their manuscript. They have convincingly addressed all my questions and comments. I am happy to recommend their revised manuscript for publication.

Response: We would like to thank the referee for a positive evaluation of our work and for the recommendation to publish.

IV. RESPONSE TO REFEREE 3

Referee: I appreciate the authors for the thorough response and clearance to my comments. The authors have added considerable new results to the manuscript. I am convinced that the results in the manuscript are scientifically solid, technically clear, and self-contained.

Response: We would like to thank the referee for a positive evaluation of our work. We hope that we have addressed the remaining concerns of the referee.

Referee: However, I agree with the other referees that the manuscript, even after revision, is a bit too technical for Nat. Commun. Moreover, I am a bit concerned about the potential scientific impact of the manuscript in relevant fields. Specifically, most results presented in the manuscript are based on the assumption that nonreciprocity or cross-coupling is composition-dependent and can change sign. Although the ubiquity of nonreciprocal interaction is broadly accepted, the existence of the specific form of nonreciprocity (i.e. it can change sign) presented in this paper is questionable. If so, then this can significantly reduce the impact of the manuscript. If not, the authors should list systems that exhibit such form of nonreciprocity or, provide other evidences/information to strengthen the motivation of this work. It is only after a clear resolution to this issue, that I can recommend publication of the manuscript on Nat. Commun.

Response: From a phenomenological perspective, when the symmetries and conservation laws provide conditions for non-reciprocity to be realised, then there is nothing preventing the non-reciprocal coupling to take on a nonlinear form. This means that the spirit of a Landau expansion can be applied both to the reciprocal part of the chemical potential and to its non-reciprocal part. In this sense, nonlinear terms will represent non-reciprocal interactions that require the presence of additional structure such as three-body interactions, allosteric-like effects where dimers and trimers made of one species interact differently as compared to two or three single units, or interactions mediated by other species as present in quorum-sensing systems. While the mechanistic details of these different microscopic realizations will be different, one can naturally expect them all to give rise to generic nonlinearities provided in Landau-like expansions.

We would like to emphasize that committing to a specific microscopic picture to build a phenomenological theory is not a natural strategy. To make the argument clear, we would like to point out that expecting this requirement as a condition for publication of our work is equivalent to not allowing Landau theory of superconductivity to be published until the BCS theory is developed for Cooper pair formation through electron-phonon interactions. In this hypothetical scenario, Landau could have argued that his phenomenological theory provides an overarching umbrella for the to-be-discovered BCS theory of superconductivity and all other forms of superconductivity which could lead to condensation of other quasi-particles. Our work is the first to introduce this phenomenology to the field of non-reciprocal active matter, and we have subsequently supplemented it with a great deal of additional work to convince the referees. These additions, as have been required by the referees, have made the manuscript much longer and more technical than originally designed by us, and this has led the referees to complain about the presentation now being too technical while simultaneously asking for more work to be done to convince them. We do not believe that this is justified.

Nevertheless, we have performed additional calculations by considering a more general polynomial form for the effective non-reciprocity coupling and used random coefficients to demonstrate the robustness of the phenomenon of Effervescence. The results are now added to the manuscript as the new Fig. 9 and in the section 'Ubiquity of Effervescence'. We would also like to highlight that the composition dependent phenomenon reported in our earlier versions of the manuscript can be written in the more general polynomial form as expanding around a non-zero composition will generate all those terms. Therefore, effervescence occurring for nonzero composition also provides indirect proof of its generality.

-
- [1] J. E. Pearson, Complex patterns in a simple system, *Science* **261**, 189 (1993).
 [2] P. Manneville and H. Chaté, Phase turbulence in the two-dimensional complex ginzburg-landau equation, *Physica D: Nonlinear Phenomena* **96**, 30 (1996).

Response to Referees: Effervescent waves in a binary mixture with non-reciprocal couplings

Suropriya Saha^{1,*} and Ramin Golestanian^{1,2,†}

¹Max Planck Institute for Dynamics and Self-Organization (MPIDS), D-37077 Göttingen, Germany

²Rudolf Peierls Centre for Theoretical Physics, University of Oxford, Oxford OX1 3PU, United Kingdom

(Dated: April 7, 2025)

I. SUMMARY OF CHANGES IN THE MANUSCRIPT

We have thoroughly revised the manuscript taking into account the feedback from both referees. We have added new results that establish the generality of the phenomenology presented in this manuscript. Here is a summary of the changes that we have made in the manuscript.

- In the Methods we have added a new section titled ‘Microscopic derivation of nonlinear NRCH’ where we have presented two systems where we expect our model to be relevant.
- We have adjusted the wording of our claims to follow the suggestion of Referee 1.
- We have added sentences to direct the attention of the reader to the microscopic justification of our theoretical framework.

II. RESPONSE TO REFEREE 1

Referee: The authors have significantly improved the reasoning behind the terminology they introduce and their explanation of the observed phenomena. I can now recommend the manuscript for publication.

Response: We are delighted to learn of the referee’s positive evaluation of our work.

Referee: I still would like to make a remark on the sign-changing nonreciprocity: In the added section “Ubiquity of effervescence”, it is shown that the specific form of the nonlinearity is unimportant for effervescence to emerge, as long as it allows α to change sign locally in some way, thereby leading to regions where reciprocal coupling is restored. I appreciate the extra work the authors put in to make this point.

Response: We thank the referee for appreciation of the crucial point on which the generality of our results hinges. The ‘sign change’ of non-reciprocity is indeed local and can occur for a generic nonlinear coupling.

Referee: However, I’m still skeptical that this sign change of the coupling is something that is generic or ubiquitous in experimental systems (or even models with explicit microscopic interactions).

Response: To address this concern, we have now incorporated two microscopic models for non-reciprocal interactions, which illustrate that it is possible to consider mixtures of active particles with any generic nonlinearity to obtain the non-reciprocal couplings as we have written down on general grounds.

Referee: The issue with the author’s appeal to Landau expansion is that such an expansion assumes small values of the fields. For the sign change to happen, the field has to become sufficiently large and therefore might leave the range of validity of the expansion. Imagine for instance a scenario where the “true” coupling decreases and asymptotes to zero, but never changes sign, e.g. $\alpha = \alpha_0/(1 + |\phi|)$. In this case, an expansion for small ϕ would lead to a spurious sign change of α . I therefore think that arguing for “ubiquity” on the grounds of Landau expansions is somewhat problematic.

Response: We have a number of points to present in response to this query. A Landau expansion, like any other

* suropriya.saha@ds.mpg.de

† ramin.golestanian@ds.mpg.de

theoretical paradigm has its usefulness and limitations. It is the foundation of a large number of papers in theoretical physics and a natural framework for our work where the broad objective is to explore the role of nonlinear non-reciprocity in a general setup. We are consistent in our construction of the model consisting of two inputs - the free energy density f and non-reciprocity α . In our case consistency is rather straightforward as it reduces to retaining terms until the third order in the fields in both functions, which we do.

We will now address the issues specific to our model. For reasons that we partly understand [1], the absolute values of the fields are bound by a Maxwell construction of an effective energy function. In the phase invariant model (Eq. 5 of the main text) we show explicitly that the amplitude is bounded by unity (the minima of f) and the angular frequency of the wave is determined by $\alpha_{0,1}$. The role of non-reciprocity is to produce phase shifts between the densities leading to motion or spatio-temporal chaos. As a result our model has the same level of rigour as the passive model B as the fields never leave the regime of ‘smallness’.

In the example suggested by the referee we agree that an expansion in small $|\phi|$ would lead to a spurious sign change and indeed this would be an error. This is where the statistical analysis is useful as it frees the model from constraints of a specific choice. In our statistical analysis of the general model, a majority, but not all random choices lead to effervescence. When we say that effervescence is ‘ubiquitous’, we mean in a statistical sense as we explain in the paper.

Finally, it is a quirk of the phase invariant model in Eq. 5 that the local sign change is conditional. Substituting $\alpha_1 = -\alpha_0$ in Eq. 4, we find that the non-reciprocity $\alpha = \alpha_0(1 + |\phi|^2)$ is always positive and effervescence does not occur in the steady state. In this regime, the travelling waves are stable and consistent with the linear stability analysis. For $\alpha_0 \sim \alpha_1$, α changes sign for $|\phi|$ close to unity (value determined by f), and we observe effervescence. However the model is quite elegant as it allows analytical calculations and easy numerical analysis.

Referee: The authors might want to stress that the “ubiquity of effervescence” is still conditional on the sign-changing nature of the non-reciprocal coupling, even if it is independent of how exactly the sign change comes about.

Response: We have added a line in the section “Ubiquity of effervescence” to stress this aspect.

Referee: The sentence “We have established that effervescent steady-states that we expect to be the defining feature of future experiments with non-reciprocally interacting scalar densities.” in the last paragraph seems to be grammatically broken. In the light of my concerns above, I also recommend that the authors tone down their claim a bit to say that effervescent steady-states are “a common feature” rather than “the defining feature” of non-reciprocally interacting scalar densities.

Response: We have modified the statement according to the recommendation made by the referee.

III. RESPONSE TO REFEREE 3

Referee: I believe the current focus of the review, based on the comments from referees 1 and 3, is whether or not it is meaningful to introduce the change of sign of inter-species interaction, which, apparently is important to the results presented in the manuscript. I agree with the arguments raised by the authors that such feature can emerge from a simple Landau-like expansion since there’s no obvious reason to forbid this. However, there’s also no reason why this should appear in any system, and that’s why the referees thought it is necessary to clarify the microscopic dynamics that can lead to such feature or, alternatively, list one or two real systems that exhibit such behavior.

Response: We illustrate using established theoretical models that the key ingredients required to arrive at the nonlinear NRCH are activity in a suitable form and general nonlinearities in mixtures with at least two conserved components. We now present two examples in the Methods section, namely, a system of self-propelled particles with quorum sensing and a system with catalytic activity, for which we calculate the parameters of the Landau expansion. The scalar fields represent the number density fields for the self-propelled particles in the first case and the catalysts in the second case. In addition to density fields, we have considered non-conserved polarisation fields and non-conserved chemical fields, respectively, which we eliminate to obtain the closed form dynamics of the density fields.

We conclude by noting that this approach supplements the work in many other papers. Other systems such as mass-conserved reaction-diffusion systems, elastic networks are amenable to a similar theoretical treatment. Nonlinearities are the rule rather than the exception in complex systems and it is our aim to highlight the diversity of routes

to nonlinear NRCH models.

Referee: Furthermore, I strongly disagree with the statement that 'expecting this requirement as a condition for publication of our work is equivalent to not allowing Landau theory of superconductivity to be published until the BCS theory is developed for Cooper pair formation through electron-phonon interactions'. In fact, it is the difference between the Landau theory and the work presented here that raises my concern about the impact of this manuscript. The Landau theory proposes a GENERIC framework that captures the most important properties of phase transition, and is applicable to diverse types of systems and phenomena. The results presented in this manuscript, relies on the fact that the inter-species interaction must be able to change sign which, although not forbidden by the physical law, could be a very limiting case that only appears in very few systems. If not, the authors should clarify this.

Response: Before addressing the main point of the referee here, we would like to point out that we have now rephrased some sentences in the conclusion of the main paper at the suggestion of Referee 1. We believe that non-reciprocal systems present a class of active matter for which we need to modify and generalize some of the concepts in comparison with how they are implemented for passive systems, as we discuss below.

In Landau theory, as applied specifically to a binary passive mixture, the linear term in the free energy changes sign as a function of temperature at which the free energy is concave at the mean composition leading to bulk phase separation.

This scenario changes in non-reciprocal systems where the sign of the active parameter itself is unimportant. For example the sign of α_0 simply determines whether 1 chases 2 or vice versa. This is true in polar non-reciprocal systems as well [2].

Now we discuss the nonlinear coefficients. The change of sign that we talk about here is a local change not a global one - a requirement that is qualitatively different from the necessity that interactions in passive phase separation should be attractive to drive phase separation. Among the five coefficients that we introduce in Eq. 5, there is no need to fix the sign of the coefficients α_{3-5} . As long as they are non-zero, effervescence will occur as shown in the results of Fig. 9 of the main paper. For example, if $\alpha = \alpha_0(1 + \alpha_1\phi_1)$, the non-reciprocal coefficient 'changes sign' at points in space where $\phi_1 = -1/\alpha_1$. This requirement is far less stringent than the requirement that particles become more attractive. These terms are different from $\alpha_{1,2}$ which are associated with terms that do not allow for his mechanism to be materialized.

There is an important difference between the phase-invariant theory that we presented in the first submission and the general theory that we present in the current version of the manuscript. The complex conserved Landau Ginzburg dynamics in Eq. 5 is more constrained than Eq. 25 which requires more tuning to observe effervescence. We used the terms 'sign changing' multiple times in the first submission but that description though accurate is perhaps misleading because - firstly, the change is local, and secondly, we have uncovered that the correct description of the phenomenon is through figure 9c where the distribution of the coefficient of non-reciprocity evolves to a form where it picks up both signs - a picture strikingly different from equilibrium. In fact Referee 1 is now convinced of this aspect and remarks - 'In the added section "Ubiquity of effervescence", it is shown that the specific form of the nonlinearity is unimportant for effervescence to emerge, as long as it allows α to change sign locally in some way, thereby leading to regions where reciprocal coupling is restored. I appreciate the extra work the authors put in to make this point.'

In the last few revisions we avoided mentioning any concrete microscopic model because of the generality our results. We expect nonlinear NRCH to be important based on very basic arguments that we outline in the response to the previous query. Perhaps, we did not emphasize this point enough in the last two revisions.

Referee: So, to conclude, I still have mixed feelings about this manuscript. On the one hand, I am convinced that the results presented here is scientifically solid and interesting from the theoretical point of view. On the other hand, I am still concerned about its limitation of impact due to the lack of clarification on the changing sign.

Response: We would to thank the referee for a positive evaluation of our results, and hope that they will return to

their initial positive verdict on our work.

-
- [1] S. Saha, Phase coexistence in the non-reciprocal cahn-hilliard model (2024), arXiv:2402.10057 [cond-mat.soft].
[2] M. Fruchart, R. Hanai, P. B. Littlewood, and V. Vitelli, Non-reciprocal phase transitions, *Nature* **592**, 363 (2021).

Response to Referees: Effervescent waves in a binary mixture with non-reciprocal couplings

Suropriya Saha^{1,*} and Ramin Golestanian^{1,2,†}

¹Max Planck Institute for Dynamics and Self-Organization (MPIDS), D-37077 Göttingen, Germany

²Rudolf Peierls Centre for Theoretical Physics, University of Oxford, Oxford OX1 3PU, United Kingdom

(Dated: May 28, 2025)

I. SUMMARY OF CHANGES IN THE MANUSCRIPT

We thank all Referees for their recommendations in the reviewing process and for the thought provoking dialogue. We have incorporated all the changes suggested by Referee 1. Here is a synopsis of the changes that we have made in this round of revisions.

- We have incorporated the changes suggested by the Referee in the abstract, summary and conclusion emphasizing the aspect of sign change.
- We have made some changes in the Methods section to clarify some aspects of the microscopic justification nonlinear NRCH.

II. RESPONSE TO REFEREE 1

Referee: In their responses, the authors have addressed my comments and those of the second referee. In the responses, they stress that a sign change in the nonreciprocal coupling is necessary for effervescence. However, this crucial aspect is easily lost in the long manuscript, where it is only mentioned relatively late and might easily be missed by the reader. I think the authors should clearly state this key requirement at prominent places in the manuscript, including in the abstract and the “Concluding remarks”.

For example, I suggest that in the abstract they expand “... model in which the non-reciprocal interactions can depend on the local values of the scalar fields” by “in such a way that the nonreciprocity can change sign”. The same caveat should be included in the sentence “We have established that effervescent steady-states will be a common feature of future experiments with non-reciprocally interacting scalar densities.” in the “Concluding remarks” section. In the section “Ubiquity of effervescence” the authors write “any nonlinearity is sufficient to produce effervescence.” This contradicts their own statement that a sign change is required. There are many nonlinearities that do not change the sign, such as the term $\alpha = 1/(1 + |\phi|^2)$ I suggested in my previous report.

In the same section, below Eq. (25), the sentence “The ingredients required to arrive at these terms are activity in a suitable form, multi-species number-conserving mixtures, and general non-linearities” should be amended by adding “that allow α to change sign within the dynamically relevant range of densities”.

Similarly, in the “Concluding remarks”, the authors write “We show numerically that effervescence only requires non-specific nonlinear interactions.” Again, I this needs to be amended to emphasize the requirement of a dynamic sign change due to the nonlinearity, which is a specific constraint on the nonlinear interactions that will produce effervescence.

Response: We thank the Referee for the suggestions. We have taken their advice and modified the relevant parts of the manuscript in a suitable way. These are marked in blue. We have also emphasized further that effervescence does not occur for $\text{sign}(\alpha_0) = -\text{sign}(\alpha_1)$ in the section ‘Steady states’ of the main text.

Referee: In their response, the authors write “As a result our model has the same level of rigour as the passive model B as the fields never leave the regime of ‘smallness’.” I disagree with this statement, since the radius of convergence of the Taylor expansion of a given alpha might be smaller than the bounds on the fields due to the effective free energy. For example, if the “true” nonreciprocity is $\alpha = 1/(1 + 4|\phi|^2)$, then the Taylor expansion will

* suropriya.saha@ds.mpg.de

† ramin.golestanian@ds.mpg.de

fail for $|\phi| > 1/2$. In fact, it will produce a spurious sign change exactly at $|\phi| = 1/2$.

Note that denominators of the form $(1 + |\phi|^n)$ appear generically in models for (bio-)chemical reactions, e.g. in Hill functions and Michaelis-Menten kinetics. The above arguments are therefore not a technical detail but point to an important limitation of the polynomial expansion approach employed by the authors.

My understanding of the second referee’s comments is that he is concerned exactly about the same thing when he writes “The results presented in this manuscript, relies on the fact that the inter-species interaction must be able to change sign which, although not forbidden by the physical law, could be a very limiting case that only appears in very few system.”

In fact, if effervescence were so ubiquitous, shouldn’t it already have been observed in nature or in experiments. Maybe dynamic sign-changes in the non-reciprocal coupling are indeed rare.

For the reasons listed above I think the authors need to generally tone down the claims about “ubiquity of effervescence” and emphasize the key requirement of sign changes in the non-reciprocity.

Response: We thank the Referee for their thoughtful summary and appreciate the important points they have raised. We agree that some of these valid concerns arise from the simplicity of our model, particularly the assumption of a polynomial expansion truncated at third order in the fields. In real physical systems, such an expansion may indeed exceed its range of validity. In future work, we aim to address these concerns more directly by numerically solving the full set of equations presented in the Methods section, which describe microscopic realizations of nonlinear NRCH, without relying on a Taylor expansion around reference states. Within this more general framework—where the free energy f is treated, for instance, as a Flory-Huggins-type functional, and active contributions are included—we expect to resolve the conceptual issues highlighted.

With respect to experiments, we respectfully disagree with the Referee. While it would indeed be ideal to identify a system in which the coexistence of droplets and chaos can be unambiguously attributed to nonlinear NRCH, we note that few theoretical models offer immediate experimental relevance. This has been the case even for frameworks such as MIPS [1], polar non-reciprocal mixtures [2], Janus colloids [3], and odd elasticity [4].

Referee: I appreciate that the authors added a section on “Microscopic justification of nonlinear NRCH” in the Methods. For the second example provided there, “Chemically active mixtures”, it is not clear to me whether the example can actually produce a sign change. In fact, the polynomial expansion provided by the authors will fail when the $\phi_i < -\bar{\rho}_i$. The authors should explicitly show that the sign change occurs for the full expression for c_1 Eq. (35).

Response: We thank the referee for the question. The field c_i represent actual densities and not deviations from a constant value and indeed cannot change sign. the quantity that enters that dynamics of ρ_i , the densities of the scalar fields, is $\delta c_i \equiv c_i - \bar{c}_i$. The deviations can be of either sign, depending on whether they chemicals are being locally consumed or produced.

The other point relevant to the query of the referee is that in the approximation of ‘local chemical equilibrium’ made in the derivation of Eq. 35, holds for small deviations in ϕ_i such that c_i is always a positive quantity. The full dynamics of c_i and ρ_i implemented with appropriate boundary conditions does not encounter this limitation. As stated in the response to the query above, a rigorous connection to the phenomenon of effervescence and the models presented above will need to be developed within the context of a specific experimental realization of the system, and that is not compatible with the context and philosophy of the present work.

Referee: I recommend this manuscript for publication once the above issues have been addressed.

Response: We thank the Referee for their positive recommendation and we hope that we have now addressed all remaining issues.

III. RESPONSE TO REFEREE 3

Referee: The authors have made significant improvements to the manuscript. Particularly, they have added considerable discussions on microscopic models that can give rise to the form of nonreciprocal interaction required by effervescence. I am happy to recommend the manuscript for publication.

Response: We thank the Referee for their positive recommendation.

- [1] M. E. Cates and J. Tailleur, Motility-induced phase separation, *Annu. Rev. Condens. Matter Phys.* **6**, 219 (2015).
- [2] M. Fruchart, R. Hanai, P. B. Littlewood, and V. Vitelli, Non-reciprocal phase transitions, *Nature* **592**, 363 (2021).
- [3] R. Golestanian, T. B. Liverpool, and A. Ajdari, Propulsion of a molecular machine by asymmetric distribution of reaction products, *Phys. Rev. Lett.* **94**, 220801 (2005).
- [4] C. Scheibner, A. Souslov, D. Banerjee, P. Surówka, W. T. M. Irvine, and V. Vitelli, Odd elasticity, *Nature Physics* **16**, 475 (2020).